

# 1 Climate change and Northern Hemisphere lake and river

# 2 ice phenology

Andrew M. W. Newton[1] and Donal Mullan[1]
[1]Geography, School of Natural and Built Environment, Queen's University Belfast, Belfast, BT7 1NN,
UK.
*Correspondence to*: Andrew M. W. Newton (amwnewton@gmail.com)
**Abstract.** At high latitudes and altitudes one of the main controls on hydrological and biogeochemical
processes is the breakup and freezeup of lake and river ice. This study uses ~2600 time series from
across 644 Northern Hemisphere lakes and river to explore historical patterns in lake and river ice
phenology across four time periods (1931-1960, 1961-1990, 1991-2005, and 1931-2005). These time
series show later breakup dates by 0.6 days per decade from 1931-2005 across North America and
Europe, with trends closely correlating with temperature. Freezeup trends are more spatiotemporally
complex with those in Europe negligible compared to later freezeup trends for North America. For the
most recent time period (1991-2005) high magnitude trends towards later freezeup that are considerably
larger than in other time periods are observed. Freezeup trends show a more limited correlation with
climate and this is likely because freezeup is not guaranteed to occur simply by temperatures dropping
below 0 °C. Across the Northern Hemisphere the length of the open water season is shown to have
increased through time, with the magnitude at its largest in the most recent time period. These results
provide an important contribution that can be used to help understand how ice phenology patterns may
change in the future with an expected rise in global mean air temperatures. Observations of an
acceleration in warming trends through time shows the importance of non-linear responses to climate
forcings. This will be crucial because it is probable that lake and river ice phenology changes, brought
about by rising air temperatures, may in turn begin to feedback into the climate system. Thus,
understanding historical changes, causes, and consequences is required to fully unravel the potential
implications of future ice phenology change.



Keywords: Lake ice, River ice, Ice phenology, Climate change

## 1. Introduction

One of the main controls on hydrological and biogeochemical processes at high latitudes is the freezeup and breakup of lake and river ice (Bengtsson, 2011; Rees et al., 2008; Stottlemyer and Toczydlowski, 1999). Ice phenology is governed by the geographical setting (heat exchange, wind, precipitation, latitude, and altitude) and the morphometry and heat storage capacity of the water body (Jeffries and Morris, 2007; Korhonen, 2006; Leppäranta, 2015; Livingstone and Adrian, 2009; Weyhenmeyer et al., 2004; Williams, 1965; Williams and Stefan, 2006). Though preceding surface air temperatures provide a seasonal energy flux that is well correlated with breakup/freezeup (Assel and Robertson, 1995; Brown and Duguay, 2010; Jeffries and Morris, 2007; Livingstone, 1997; Palecki and Barry, 1986), cycles of temperature linked to large-scale climatic indices have also occasionally been observed to impact ice phenology (Livingstone, 2000a).

The majority of lakes and rivers that seasonally freeze are in the Northern Hemisphere and most research has tended to focus on breakup/freezeup dates, ice season length and ice thickness (Duguay et al., 2003; Prowse et al., 2011). As acknowledged by the IPCC (2013), an assessment of changes in broader ice phenology is complicated by, among several factors, the tendency to consider only local areas. Although trends vary, there is a proclivity for breakup/freezeup records to lean toward shorter ice seasons that are correlated with temperature trends (Table 1). Changes in ice breakup/freezeup dates, therefore, provide an additional data source for investigating climate patterns (Assel et al., 2003). Whilst the current literature supports observations of a warming climate, the full spatiotemporal variation seen in smaller case studies has not been transferred to hemispheric scale. This is important because over the next century temperature rise is expected to continue across the Arctic, where lakes and rivers subjected to freeze and thaw cycles are predominantly located (Collins et al., 2013). Understanding historical patterns and changes in lake and river ice phenology is required to confidently project future evolution and climate system feedbacks (Brown and Duguay, 2011; Emilson et al., 2018). In the last century the





number of ice phenology observations have increased markedly due to their importance for energy and
water balances (Rouse et al., 2003; Weyhenmeyer et al., 2011) and infrastructure such as ice roads
(Mullan et al., 2017). This paper explores the hemispheric spatiotemporal trends in ice phenology by
investigating an extensive database containing ~2600 individual time series from 644 Northern
Hemisphere study sites. This database is used to explore the spatiotemporal variability of lake and river
ice breakup/freezeup dates from 1931-2005. Observed changes are then compared with climate records
and atmospheric/oceanic modes of variability to understand their respective roles in driving the
observed ice phenology patterns.

| Region | Reference | Time Period | Key Observations |
|---|---|---|---|
| North America | Assel and Robertson (1995) | 1851-1993 | - Breakup dates have become earlier since 1940 with air temperatures increasing during the winter season at Lake Michigan |
| North America | Assel et al. (2003) | 1963-2001 | - Great Lakes show a reduction in the maximum fraction of lake surface ice coverage |
| North America | Bai et al. (2012) | 1963-2010 | - Great Lakes show ice cover has detectable relationships with NAO and ENSO |
| North America | Bennington et al. (2010) | 1979-2006 | - Model results show increased Lake Superior surface temperatures and declining ice coverage of 886 km$^2$ per year |
| North America | Bonsal et al. (2006) | 1950-1999 | - Ice phenology influenced by extreme phases of PNA, PDO, ENSO and NP in Canada<br>- Lake have a stronger and more coherent pattern compared to rivers |
| North America | Brammer et al. (2015) | 1972-2013 | - Ice season length decreased over the time period and was driven by earlier breakup |
| North America | Duguay et al. (2006) | 1951-2000 | - Earlier breakup trends in most lakes that were consistent with snow cover duration<br>- Freezeup trends were more variable with later and earlier dates<br>- Strong relationship is shown between 0 °C and breakup/freezeup dates in Canada |
| North America | Futter (2003) | 1853-2001 | - In Southern Ontario significant trends towards earlier breakup and an extension to the ice season length |
| North America | Ghanbari et al. (2009) | 1855-2005 | - PDO, ENSO, and NAO explain some, but not all ice phenology variability at Lake Mendota |
| North America | Hewitt et al. (2018) | 1981-2015 | - Lake ice breakup occurred 1.4 days per decade earlier and freezeup 2.3 days per decade later over the time period<br>- Strong association with warming air temperature patterns |
| North America | Hodgkins et al. (2005) | 1930-2000 | - River sites in New England show a decrease in ice season length of 20 days per year |
| North America | Jensen et al. (2007) | 1975-2004 | - Recent trends for changes in breakup/freezeup dates were larger than historical trends, with ice duration decreasing by 5.3 days per decade in the Great Lakes region |



| North America | Lacroix et al. (2005) | 1822-1999 | - Across Canada breakup dates tend to be earlier whilst freezeup trends tend to be spatiotemporally more variable |
|---|---|---|---|
| North America | Latifovic and Pouliot (2007) | 1950-2004 | - Average of 0.18 days per year earlier breakup and 0.12 days per year later freezeup for the majority of sites in Canada |
| North America | Magnuson et al. (2005) | 1977-2002 | - Lakes in the Great Lakes region show a generally coherent pattern for breakup |
| North America | Sharma et al. (2013) | 1905-2004 | - Linear trends in rain and snowfall in the month prior to breakup, air temperature in the winter, and large-scale climatic oscillations all significantly influence breakup timing |
| North America | White et al. (2007) | 1912-2001 | - Earlier breakup and later freezeup for a number of river sites across Alaska and Maine |
| Europe | Blenckner et al. (2004) | 1961-2002 | - NAO and ice cover show strong relationship that is less pronounced in the north compared to the south in Sweden and Finland |
| Europe | Gebre and Alfredsen (2011) | 1864-2009 | - Variable trends towards later and earlier breakup/freezeup for rivers in Norway<br>- Temperature and river discharge important for breakup/freezeup |
| Europe | George (2007) | 1933-2000 | - Reduction in the number of days with ice and frequency of ice cover<br>- NAO strong influence on annual variability at Lake Windermere |
| Europe | Korhonen (2006) | 1693-2002 | - In Finland there are significant trends towards earlier breakup in the later 19th century to 2002<br>- Trends toward later freezeup leading to a reduction in ice season length |
| Europe | Marszelewski and Skowron (2006) | 1961-2000 | - Ice season length has been reducing by 0.8-0.9 days per year at six lakes in northern Poland |
| Europe | Nõges and Nõges (2014) | 1922-2011 | - Greater levels of snowfall associated with later breakup<br>- Lake ice phenology trends were weak, despite significant air and lake surface temperature trends |
| Europe | Šarauskienė and Jurgelėnaitė (2008) | 1931-2005 | - In Lithuania warmer winters caused later freezeup and reduced ice season length |
| Europe | Stonevicius et al. (2008) | 1812–2000 | - Reduction in ice season length for the Nemunas River, Lithuania |
| Europe | Weyhenmeyer et al. (2004) | 1960-2002 | - Results from 196 Swedish lakes showing a nonlinear temperature response of breakup dates<br>- Future climate change impacts will likely vary along a temperature gradient |
| Russia | Borshch et al. (2001) | 1893-1991 | - In European Russia freezeup occurs later and breakup occurs earlier<br>- Rivers assessed in Siberia show insignificant and occasionally opposite trends |
| Russia | Karetnikov and Naumenko (2008) | 1943-2007 | - NAO is well correlated with the ice cover at Lake Ladoga |
| Russia | Kouraev et al. (2007) | 1869-2004 | - Lake Baikal trends change through time with period from 1990-2004 characterised by an increased ice season length |
| Russia | Livingstone (1999) | 1869-1996 | - Breakup relationship with NAO after 1920 at Lake Baikal |





| Russia | Smith (2000) | 1917-1994 | - Fluctuations of patterns between longer and shorter ice season lengths that are generally consistent with temperature trends |
|---|---|---|---|
| Russia | Todd and Mackay, (2003) | 1869-1996 | - Significant trends towards reduced ice season and ice thickness at Lake Baikal over the period of study |
| Russia | Vuglinsky (2002) | 1917-1994 | - Rivers in Asian Russia form earlier and breakup later compared to rivers in European Russia<br>- This is due to antecedent climatological conditions |
| Asia | Batima et al. (Batima et al., 2004) | 1945-1999 | - River ice thickness and ice season length have decreased over the time period |
| Asia | Jiang et al. (2008) | 1968-2001 | - Yellow River in has experienced later freezeup and earlier breakup, leading to a reduction of the ice season 12-38 days at different sites along the river |
| Northern Hemisphere | Benson et al. (2012) | 1855-2005 | - For 75 lakes the trends towards earlier breakup, later freezeup and a shorter ice season duration were stronger for the most recent time period studied |
| Northern Hemisphere | Livingstone (2000b) | 1865-1996 | - NAO signal detected at a number of sites, but with variable strength across several Northern Hemisphere sites |
| Northern Hemisphere | Magnuson et al. (2000a) | 1846-1995 | - Breakup on average 6.3 days per century earlier across multiple Northern Hemisphere sites<br>- Freezeup on average 5.7 days later per century |
| Northern Hemisphere | Sharma et al. (2014) | 1854-2004 | - All 13 lake study sites demonstrated oscillatory dynamics were influential on ice breakup |
| Northern Hemisphere | Sharma et al. (2016) | 1443-2014 | - Trends towards later freezeup in Japan and earlier breakup in Finland<br>- Strong linkage between these trends and climate change and variability |
| Northern Hemisphere | Sharma et al. (2019) | 1443-2018 | - Analysis of 513 sites shows the importance of air temperature, lake morphometry, elevation and shoreline geometry in governing ice cover<br>- Future projections suggest an extensive loss of lake ice over the next generation |
| Northern Hemisphere | Šmejkalová et al. (2016) | 2000-2013 | - All areas showed significant trends of earlier breakup<br>- The 0 °C isotherm shows the strongest relationship with ice phenology trends |


**Table 1**: Summary of ice phenology trend observations from across the Northern Hemisphere. Note
this is not meant to be an exhaustive list, but intends to provide a general overview of ice phenology
changes.



## 2. Materials and methods

The Global Lake and River Ice Phenology Database from the National Snow and Ice Data Centre (NSIDC) (available at: https://nsidc.org/data/lake_river_ice/ – Benson et al. (2013)) provides breakup/freezeup dates for 865 Northern Hemisphere sites. In this database the freezeup date is defined as the first day in which the water is completely ice covered and the breakup is the date of the last ice breakup before the open water season. Whilst the specific definitions for breakup/freezeup may vary between different sites, the precise definition is thought to be consistent at each site. Thus, if climate signals are present in the ice phenology data then they should still be observable and broadly comparable. This database is supplemented with data from the Swedish Meteorological and Hydrological Institute (SMHI) which contains 749 lakes and rivers using similar terminology. Data for 122 lakes and rivers were provided by the Finnish Meteorological Institute. Several sites were already in the NSIDC dataset but were updated where necessary. The three datasets were integrated to create the Ice Phenology Database (IPD) containing data across North America and Eurasia (Fig. 1). It is important to note that in the later part of the 1980s and 1990s many Russian and Canadian sites stopped recording data.

Prior to 1931 data are sparse and many of the longer time series have been explored by Magnuson et al. (2000b) and Benson et al. (2012). To understand the spatiotemporal patterns of ice phenology, four time periods were studied: 1931-1960, 1961-1990, 1991-2005, and 1931-2005. All study sites in the database which fall within these time periods and have a maximum of 10% missing values were included. These specific time periods have been chosen as they offer the opportunity to include as much data from the IPD as possible. Initial analysis showed that of the 1736 lakes and rivers in the IPD, 644 sites had data with at least 90% coverage for either freezeup or breakup for at least one time period. These data provide ~2600 individual time series and are spread across the Northern Hemisphere (Fig. 1a) but are primarily concentrated in North America (Fig. 1b) and Fennoscandia (Fig. 1c). Time series covering 1931-2005 were available, with 88 sites (three rivers) having breakup data, 48 sites (two rivers) with freezeup data, and 37 sites (one river) with data for the number of annual open water days. Of these sites the majority in North America are to the east and west of the Laurentian Great Lakes (Fig. 1d). In northwest Europe the sites are predominantly in Sweden and Finland (Fig. 1e), with one site (Lej da San Murezzan) in Switzerland. In Russia there is only one site in the southwest of Lake Baikal.



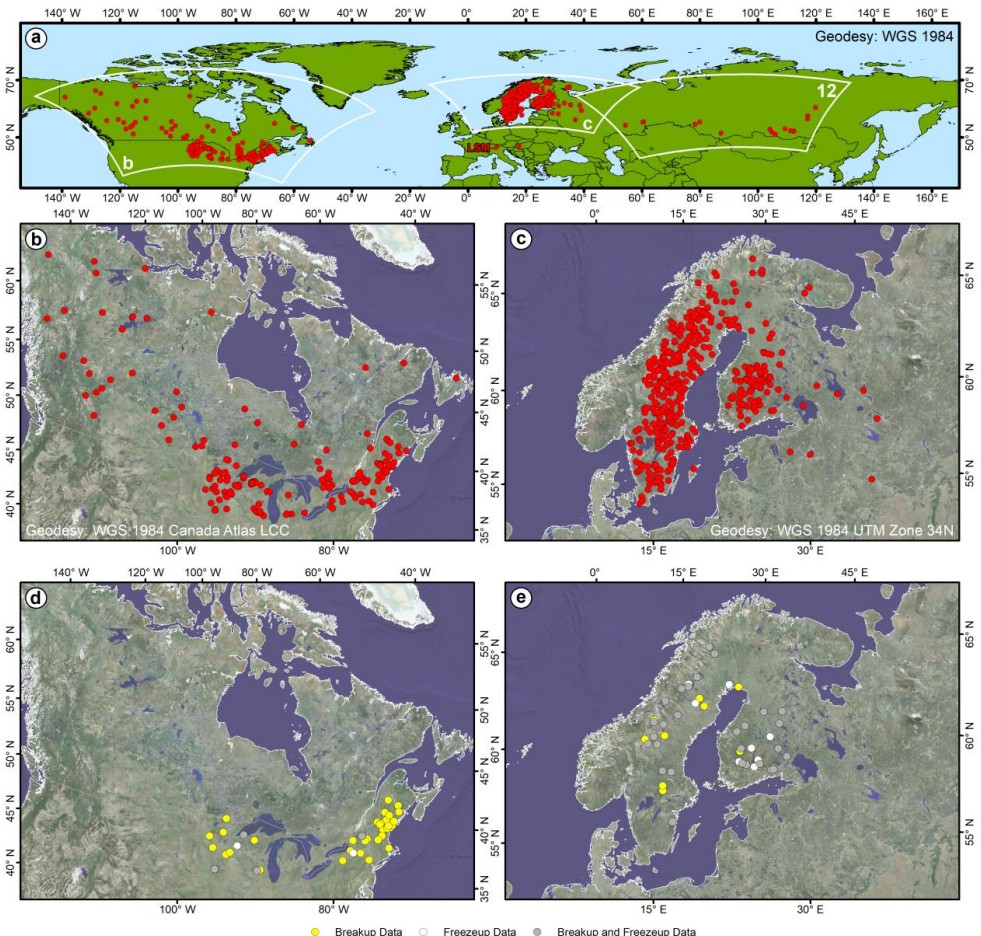

91

**Figure 1**: **a**) Map showing study sites (red circles) with time series containing at least 90% coverage for breakup

and/or freezeup during at least one time period. Location of other panels and figures are shown; **b**) North American

study sites; **c**) Fennoscandian study sites. Note that different geodesies are used to best display sites; **d**) North

American sites with data for 1931-2005; **e**) Fennoscandian sites with data for 1931-2005. The satellite imagery

shown is from the MDA NaturalVue satellite layer from ArcMAP Online. Geodesy, geographical extent, and

satellite imagery used in panels (a-e) are used in subsequent figures.


Breakup/freezeup dates were first converted to Julian days. For some sites, freezeup or breakup in a specific year
occasionally fell in a preceding or succeeding year and the Julian date reflects this by providing a relative date –
i.e. if freezeup for the 1941 ice season occurred on 5th January 1942 then the Julian day allocated was 370. Likewise,
if breakup for the 1943 ice season occurred on the 28th December 1942 then the Julian date allocated was -3. These





records were adjusted as necessary to calculate the number of annual open water days. The Julian day records were
tested using the Mann-Kendall test where the null hypothesis ($H_o$) of no trend was tested against the alternative
hypothesis ($H_1$) that there is a monotonic trend in the time series. The Mann-Kendall test does not calculate trend
magnitude, so Sen's slope was used (Yue et al., 2002). These two statistical techniques are explained briefly below
and a full definition is provided by Salmi et al. (2002).
The Mann-Kendall test is a nonparametric test which detects trends without specifying if it is linear or nonlinear.
It is used widely in environmental science as it can account for missing values. It is based on the test statistic $S$
which is defined in eq.1 and 2:


$$S = \sum_{k=1}^{n-1} \sum_{j=k+1}^{n} sgn(x_j - x_k) \tag{1}$$


where $x_j$ and $x_k$ are the sequential data values in years $j$ and $k$, $n$ is the length of the dataset, and


$$sgn(x_j - x_k) = \begin{cases} +1 & \text{if } x_j - x_k > 0 \\ 0 & \text{if } x_j - x_k = 0 \\ -1 & \text{if } x_j - x_k < 0 \end{cases} \tag{2}$$


When $n \geq 10$, the variance of $S$ is approximated by eq.3 which accounts for ties in the data:

$$\tag{3}$$


$$VAR(S) = \frac{1}{18}[n(n-1)(2n+5) - \sum_{p=1}^{q} t_p(t_p - 1)(2t_p + 5)]$$

Where $q$ is the number of tied groups and $t_p$ is the number of data values in the $p^{th}$ group. The standard $Z$ statistic
is then computed by using eq.4 to show how significant any trends are and whether the $H_o$ can be rejected at $\alpha$
significance level.



$$Z = \begin{cases} \dfrac{S-1}{\sqrt{VAR(S)}} & \text{if } S > 0 \\ 0 & \text{if } S = 0 \\ \dfrac{S+1}{\sqrt{VAR(S)}} & \text{if } S < 0 \end{cases}$$
(4)


If any trends are present then they can be estimated using the nonparametric Sen's slope. The trend slope (β) is
given by eq.5 where $x_k$ is the $k^{\text{th}}$ observation:

$$\beta = \text{median} \left( \frac{x_j - x_k}{j - k} \right) \text{for all } j > k$$
(5)


A new database was created which included significance, slope, and decadal change for each site. These were
mapped to show spatiotemporal change during different time periods.
A range of climate variables and atmospheric/oceanic modes of variability were downloaded from KNMI Climate
Explorer (http://climexp.knmi.nl/) to facilitate examination of potential drivers of ice phenology change. Monthly
mean temperatures and precipitation were extracted from the Climatic Research Unit (CRU) Time-Series (TS)
Version 4.01 (Harris et al., 2014). CRU TS4.01 applies angular-distance weighting (ADW) interpolation to monthly
observational data derived from national meteorological services to produce monthly gridded mean temperatures
and precipitation at a spatial resolution of 0.5° latitude x 0.5° longitude. Wind speed data were extracted from the
International Comprehensive Ocean-Atmosphere Data Set (ICOADS) 2-degree Enhanced Dataset, which provides
simple gridded monthly wind speeds for 2° latitude x 2° longitude grid boxes (Freeman et al., 2017).  All these data
were downloaded as a spatially averaged regional time series for three geographical regions – Fennoscandia (FEN):
57.5-68.5°N, 12-29°E; North America (NAM): 42.5-47°N, 73.5-95.5°W; and Russia (RUS): 51.5-52°N, 104.5-
105°E. Data were extracted for 1931-2005 to correspond with the length of the IPD. For 1931-2005 monthly data
on the Arctic Oscillation (AO) (Thompson and Wallace, 2000), the Atlantic Multidecadal Oscillation (AMO) (van
Oldenborgh et al., 2009), and the North Atlantic Oscillation (NAO) (Jones et al., 1997) were also extracted.
Ice breakup/freezeup records from the IPD were spatially averaged into three regional composite records
corresponding to the three geographical regions (FEN, NAM, and RUS) defined above. Statistical relationships
were then examined between ice breakup/freezeup dates and climate records (maximum temperatures and modes





of variability) using Pearson Product-Moment Correlation. These relationships were analysed on a monthly basis,
first for each of the twelve calendar months, and second for twelve sliding windows of three-month means (e.g.
mean of January, February, March, then mean of February, March, April etc.).

**3. Ice phenology change**
A climate regime with increasing mean temperatures would be expected to increase the number of annual open
water days for sites with seasonal freezing. This reduction in ice cover could result from earlier breakup, later
freezeup dates, or a combination of both that leaves a relative increase in the number of open water days. Decadal
trends for the number of annual open water days allows for an integrated observation of breakup/freezeup date
changes relative to each other – i.e. the longevity of ice covers rather than a specific shift in the precise
freezeup/break dates. The statistical analysis outlined in the methods has been carried out for each study site with
freezeup and/or breakup dates shown in Fig. 2. This is used to determine decadal trend directions in each time
period for ~2600 individual time series. These have been summarised in Fig. 3 as a proportion of total observations
for each time period and in Table 2 as mean values for breakup, freezeup, and the number of open water days. The
analyses carried out suggest that although spatiotemporally variable, there is a dominance for trends to display a
signal of reduced ice cover and an increase in the magnitude of that reduction through time. In this section the
general patterns are presented before an in-depth analysis of the changes observed in the three main study areas,
and the 1931-2005 trends for sites with continuous data.

**3.1. General trends**
The combined time series and spread of dates for breakup/freezeup across each time period are summarised in Fig.
2. This shows that for breakup, the median date of ice breakup in each period is correlated with the study site
latitude. This is the case for both Europe and North America, but not Russia, likely owing to the extensive
geographical spread of only a few study sites. In Europe, successive time periods show a general shift toward earlier
median breakup dates, as is shown by the increased number of study sites with median breakup dates within the
first 100-125 days of each year. The earliest breakup dates observed at each European site shows that many of the
study sites have experienced a systematic shift in the breakup date to earlier in the year. This is evidenced by the



increased clustering of breakup dates within the first 100 days of the year during 1991-2005 compared to 1931-
1960, particularly at study sites between 55°N and 60°N (Fig. 2). The North American sites show a similar shift in
the spread of breakup dates to earlier in the year. Russian observations are too few to draw strong conclusions. An
important observation is the reduction in North American (mainly Canadian) and Russian study sites in the mid-
latitudes (47°N to 55°N) from 1991-2005.

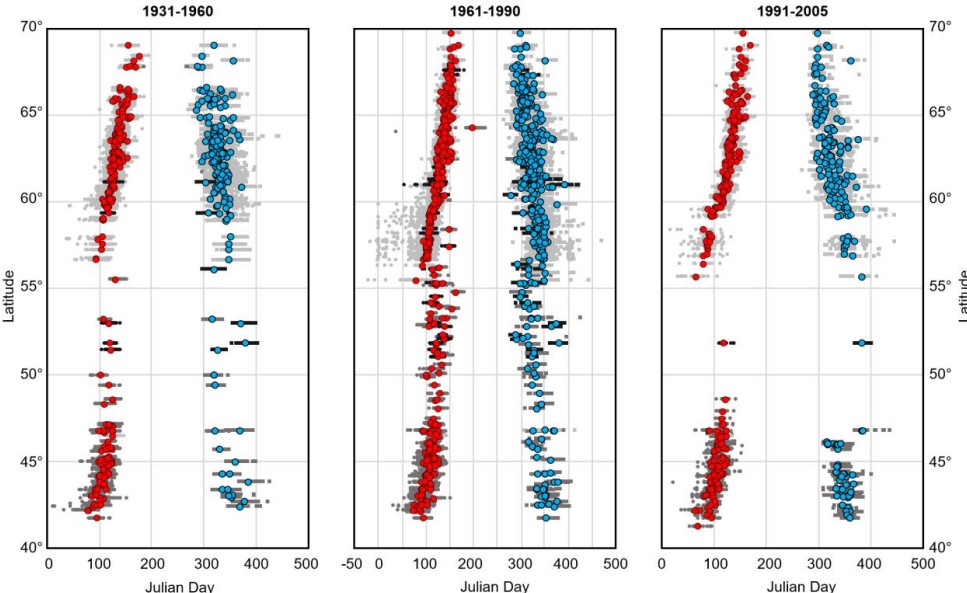


**Figure 2**: Summary graphs showing breakup and freezeup dates against latitude for the three short time periods.
Red and blue data points represent study site median breakup and freezeup dates, respectively. Light grey
observations are breakup/freezeup dates for study sites in Europe, dark grey sites across Russia, and intermediate
grey is North American sites. Note that some European breakup observations between 1961-1990 demonstrate that
breakup occurred in the December preceding the start of that years' open water season.

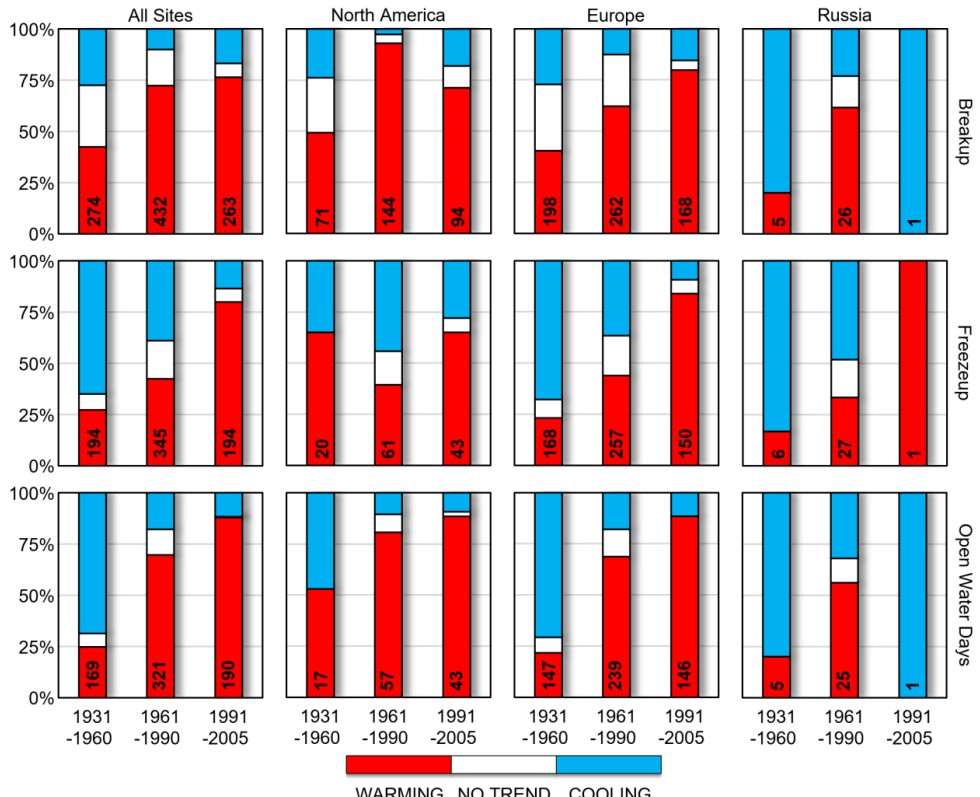


**Figure 3**: Summary charts showing generalised decadal patterns. The percentages are calculated as a proportion of

the total number of sites for each time period (bold text – e.g. in the first panel, across the Northern Hemisphere

data there are 263 sites with 1991-2005 breakup data. Note that a warming trend for breakup or freezeup dates is

determined by a negative (earlier date) or positive (later date) trend, respectively. A cooling trend for breakup or

freezeup dates displays a positive (later date) or negative (earlier date) trend, respectively. For the open water charts

a positive trend indicates an increase in the number of annual open water days.

195

When all sites are considered there is a clear increase through time in the proportion of sites displaying earlier

breakup. Sites displaying no or later breakup decadal trends decrease through time, albeit with the smallest

proportion observed from 1961-1990 (Fig. 3). At a regional extent the breakup trends for Europe are similar to

those shown for all sites. For North America the pattern is temporally complex as 1961-1990 shows the greatest

proportion of sites with earlier breakup trends. However, from the 1931-1960 period to the 1991-2005 period there

is an overall increase in the proportion of sites experiencing earlier breakup. For Russian sites it is clear that from



1931-1960 to 1961-1990 there is a considerable increase in the proportion of sites with warming trends, but a caveat
is the small sample size for 1931-1960 over a large area. For 1991-2005 there is only one site showing a cooling
trend (Fig. 3).
Mean values for decadal change are summarised in Table 2 for the three shorter time periods. This shows that for
breakup there are warming trends (i.e. negative values) that are broadly consistent through time, with a notable
increase in the magnitude of decadal change– i.e. from 0.16 days decade$^{-1}$ earlier breakup during 1931-1960 to 3.44
days decade$^{-1}$ earlier breakup during 1991-2005. This is the case for both lakes and rivers in Europe and North
America, with a minor exception that during 1961-1990 the magnitude of warming in North America was larger
than the time periods before and after. In Russia, long-term trends are limited by the single site for 1991-2005, but
it does show from 1931-1960 to 1961-1990 the decadal change values move from cooling to warming trends.
The data show that across the Northern Hemisphere there is a larger spread in freezeup dates (Fig. 2). A correlation
between freezeup date and latitude is observed, but this is not as strong compared to breakup. In Europe there is a
general pattern toward a greater proportion of the freezeup dates occurring later in the year by 1991-2005 compared
to 1931-1960. In North America freezeup dates spread between 40°N and 46°N do not appear to change
significantly between all three time periods, with the exception that median dates are less spread from 1991-2005.
Russian sites are difficult to assess due to geographical coverage, but it is notable that from 1961-1990 the sites at
similar latitudes can have very different median freezeup dates.
The decadal trends through time show an increased proportion of sites with later freezeup and a decreased
proportion displaying earlier freezeup (Fig. 3). The proportion of sites with no decadal trend is similar from the
earliest time period to the latest, but with the middle period (1961-1990) showing an increase in the proportion of
sites displaying no decadal trends. The same patterns are evident for the European sites. Freezeup trends in North
America show that the earliest and latest periods (1931-1960 and 1991-2005) have similar proportions of sites
showing later freezeup trends – the same is true for earlier freezeup trends. The interim 1961-1990 shows a
pronounced increase in the proportion of sites with earlier freezeup trends and a decrease in the number of sites
with later freezeup trends compared with the previous and subsequent time periods. In Russia, from 1931-1960 to
1961-1990 there is an increased proportion of sites with later freezeup decadal trends and a reduction in earlier
freezeup trends through time. Only one site is available for 1991-2005 and it shows later freezeup (Fig. 3).



|  |  | Breakup | | | Freezeup | | | Open Water | | |
|---|---|---|---|---|---|---|---|---|---|---|
|  |  | Lakes | Rivers | Total | Lakes | Rivers | Total | Lakes | Rivers | Total |
| **Northern Hemisphere** | 1931-1960 | -0.14 | -0.6 | -0.16 | -2.11 | 3.08 | -1.82 | -2.18 | 3.33 | -1.92 |
|  | 1961-1990 | -1.51 | -1.98 | -1.53 | 0.22 | 0.07 | 0.21 | 1.94 | 1.46 | 1.92 |
|  | 1991-2005 | -3.47 | -2.23 | -3.44 | 7.78 | 11.98 | 7.83 | 12.25 | 14.77 | 12.28 |
|  | 1931-2005 | -0.59 | -0.23 | -0.58 | -0.01 | 0.70 | 0.02 | 0.60 | 1.62 | 0.63 |
| **Europe** | 1931-1960 | -0.11 | -0.22 | -0.12 | -2.31 | 3.52 | -2.13 | -2.24 | 3.38 | -2.09 |
|  | 1961-1990 | -0.78 | -2.17 | -0.81 | 0.34 | 0.18 | 0.33 | 1.81 | 1.99 | 1.81 |
|  | 1991-2005 | -3.91 | -2.23 | -3.86 | 8.59 | 11.98 | 8.63 | 13.24 | 14.77 | 13.26 |
|  | 1931-2005 | -0.53 | -0.23 | -0.51 | 0.25 | 0.70 | -0.20 | 0.35 | 1.62 | 0.39 |
| **North America** | 1931-1960 | -0.28 | -1.09 | -0.36 | 0.05 | 2.71 | 0.85 | -1.39 | 3.29 | -0.29 |
|  | 1961-1990 | -3.11 | -1.92 | -2.98 | -0.24 | 0.04 | -0.18 | 3.08 | 1.30 | 2.77 |
|  | 1991-2005 | -2.79 | N/A | -2.79 | 5.09 | N/A | 5.09 | 9.29 | N/A | 9.29 |
|  | 1931-2005 | -0.66 | N/A | -0.66 | 0.84 | N/A | 0.84 | 1.49 | N/A | 1.49 |
| **Russia** | 1931-1960 | 0.83 | N/A | 0.83 | -1.92 | N/A | -1.92 | -2.47 | N/A | -2.47 |
|  | 1961-1990 | -0.83 | N/A | -0.83 | -0.03 | N/A | -0.03 | 1.03 | N/A | 1.03 |
|  | 1991-2005 | 5.00 | N/A | 5.00 | 5.00 | N/A | 5.00 | -2.00 | N/A | -2.00 |
|  | 1931-2005 | 0.00 | N/A | 0.00 | 1.04 | N/A | 1.04 | 1.53 | N/A | 1.53 |


**Table 2**: Breakdown of mean decadal trends for each time period where each value is the number of days change

per decade. Negative values represent earlier breakup (warming trend), earlier freezeup (cooling trend) and reduced

number of open water days (cooling trend). Positive values indicate the opposite.


Decadal lake freezeup changes across all Northern Hemisphere sites show a clear change in the trend direction

from earlier freezeup (negative trends) to later freezeup (positive trends) through time (Table 2). Similar to the

values for breakup, the magnitude of decadal changes for freezeup increases through time, though they are notably

higher for freezeup than breakup – i.e. from 1.82 days decade$^{-1}$ earlier freezeup during 1931-1960 to 7.83 days

decade$^{-1}$ later freezeup during 1991-2005. River sites appear to experience the trends toward later freezeup dates

earlier than for lakes, possibly suggesting differences in how lakes and rivers have responded to climatic changes.

In Europe there is a steady change from earlier freezeup trends that match the hemispheric pattern. In North





America the lakes experience low magnitude trends close to zero for the first two time periods, suggesting limited
changes in freezeup dates. From 1991-2005 the lakes in North America (which are only in the United States due to
a reduction in Canadian monitoring) display later freezeup trends (i.e. 5.09 days decade$^{-1}$). The Russian sites show
a decline in strength of the cooling trend from 1931-1960 to 1961-1990, with only one site record for the 1991-
2005 period.
Trends in the number of open water days per year allow for any changes in breakup and freezeup dates to be
integrated together to explore the relative changes. This means that a general signal can be extracted from sites that
may have conflicting patterns of warming and cooling for the breakup/freezeup dates – i.e. if breakup dates were
becoming earlier at a faster rate than earlier freezeup then this will be reflected with a relative widening of the open
water season. When all sites are considered there is a shift towards an increased proportion of sites with warming
trends (i.e. less days with ice cover) (Fig. 3). This is the same for both Europe and North America. Russian sites
are hampered by the lack of sites for 1991-2005 but show from 1931-1960 to 1961-1990 there is an increase in the
proportion of warming trends. European and (to a lesser extent) the Russian sites show that the increased proportion
of sites with more open water days is matched by an increased proportion of sites displaying earlier breakup and
later freezeup. In North America across the three time periods the proportion of sites with a warming trends
increases through time. During 1961-1990 although there is an increase in the number of open water days,
compared to the previous period, this is primarily due to earlier breakup trends that are stronger than earlier freezeup
trends. These movements show that whilst the ice-free season is widening through time in North America, the
relative contribution from changes in breakup/freezeup date changes through time (Fig. 3).
All Northern Hemisphere sites, and when considered on a more regional scale (acknowledging the caveat associated
with the Russian data availability for 1991-2005) there is a clear trend toward an increased number of annual open
water days. This trend is similar to that experienced for both breakup and freezeup, showing a clear increase in
magnitude through time for each region with a 2.18 days decade$^{-1}$ reduction in the number of open water days for
1931-1960 to an increase in the number of open water days by 12.28 days decade$^{-1}$ for 1991-2005.

**3.2. North America**
**3.2.1. 1931-1960**



In North America, interpretation of regional ice phenology changes is limited by continuity and location of sites
(Fig. 4). The only area consistently documented is around the Laurentian Great Lakes. During 1931-1960 a
longitudinal split in the dominant decadal breakup patterns is apparent (Fig. 4a). In the east warming decadal trends
(earlier breakup) dominate and in the west cooling trends (later breakup). Breakup trends appear to correlate well
with rising temperatures in coastal regions and cooling temperatures in the interior (Balling and Idso, 1989). Mean
trends suggest that breakup was occurring 0.4 days decade$^{-1}$ earlier during this period, with the trend being stronger
for rivers (1.1 days decade$^{-1}$) compared to lakes (0.28 days decade$^{-1}$).
Compared to breakup, there is typically less than 50% of the number of sites available for freezeup from 1931-
1960 and these are again concentrated around the Great Lakes (Fig. 4b). The mean trend was toward later freezeup
of 0.9 days decade$^{-1}$. Like breakup trends, river freezeup changes (2.7 days decade$^{-1}$ later) were greater than lakes
(0.1 days decade$^{-1}$ later). No dominant spatial patterns are apparent. Whilst most North American rivers display
later freezeup, the Liard River in northern Canada displays a statistically significant ($\alpha$=0.1) earlier freezeup trend
of 3.1 days decade$^{-1}$ from 1931-1960, showing that not all the rivers are responding in the same trend direction.
Trends for annual open water days for 1931-1960 are mixed, with sites in the east and west both exhibiting reduced
and extended seasons (Fig. 4c). This is broadly similar to the heterogeneous patterns observed for breakup/freezeup,
with an overall reduction of 0.3 days decade$^{-1}$ in the number of open water days (Table 2). At a number of sites,
the open water season has reduced due to later breakup and earlier freezeup. At other sites this reduction is reflected
by warming and cooling trends (e.g. earlier breakup and earlier freezeup) where larger magnitude cooling trends
have reduced open water season length (Fig. 5). The result at Lake Monona is peculiar in that it demonstrates no
discernible trends for changes in the breakup date but a warming trend in the freezeup date, culminating in a
reduction in the open water season. It is not clear why this is the case but it likely reflects the greater variability of
freezeup dates (which varied across two months) compared to the variability of breakup dates (varied across one
month). This greater range is also reflected by the greater standard deviation (used as a measure of interannual
variability) for freezeup (14) compared to breakup (eight). Greater freezeup date variability means that the
estimated trend is more vulnerable to years with extreme dates.



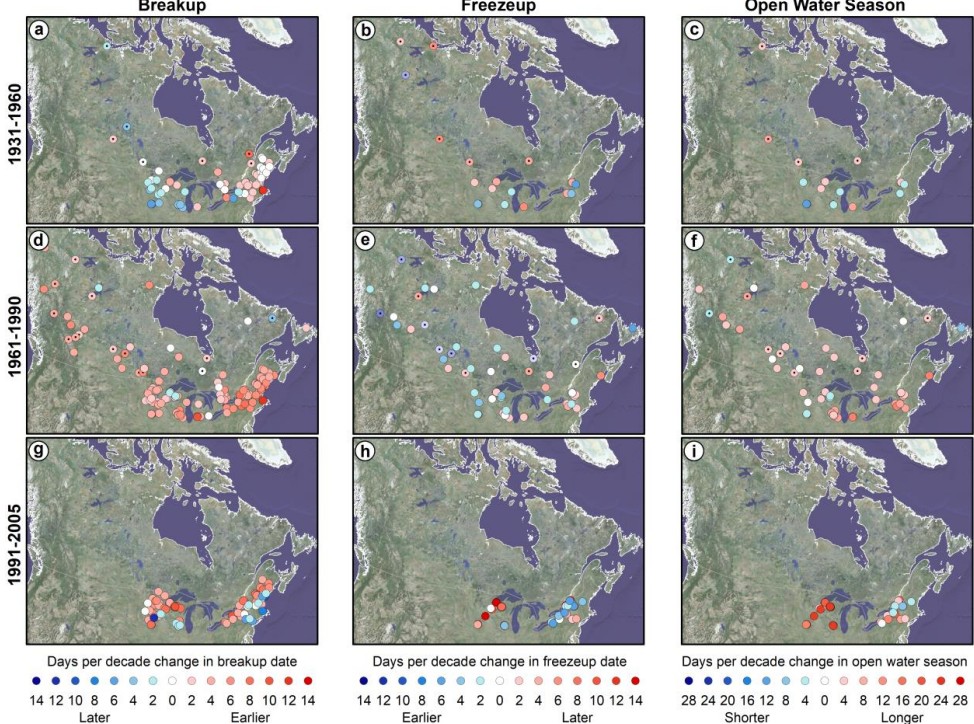


**Figure 4**: Decadal trends for breakup (**a**, **d**, and **g**), freezeup (**b**, **e**, and **h**), and length of the open water season (**c**,

f, and **i**) in North America for the three individual time periods. Sites with a dot in the centre of the circle are river

sites. Blue and red tones on the scales related to cooling and warming trends, respectively.


### 3.2.2. 1961-1990

During 1961-1990 the number of study sites increases and these sites show earlier breakup trends expanding to the

west (Fig. 4d). The mean trends show breakup was earlier by 3.0 days decade$^{-1}$, an increased magnitude of change

compared to 1931-1960. A number of the sites that experienced later breakup during 1931-1960 show a change

toward earlier breakup decadal trends. Only four North American sites experience later breakup during 1961-1990

and appear to be out of sync with adjacent sites, or have been subjected to unique circumstances that can explain

opposing trends. For example, Frame Lake in northwest Canada shows a later breakup of 0.2 days decade$^{-1}$ and

this is contrasted by two adjacent sites, Back Bay Lake and the Hay River, that experienced breakup occurring 1.3

and 1.7 days decade$^{-1}$ earlier, respectively. No immediate reasons are clear why Frame Lake is responding

differently to adjacent sites. A second site in Canada, the Churchill River near the Gulf of St. Lawrence, displays



later breakup of 3.1 days decade[-1] (α = 0.05). On this particular river, discharge has decreased due to dam
construction in the 1960s (Déry et al., 2005; Déry and Wood, 2005) and since the rising limb of the spring
hydrograph is important for mechanical ice breakup (Prowse and Beltaos, 2002), flow diversion likely made the
rising limb a shallower gradient, meaning it takes longer for ice to reach a mechanical threshold where it breaks
apart.
In the United States, Chequamegon Bay in southern Lake Superior displays later breakup trends of 0.7 days decade[-1]
for 1961-1990. Two other sites on Lake Superior at Bayfield and Madeline Island show earlier breakup of 2.1 and
1.9 days decade[-1], respectively. The dominant circulation in the southern part of Lake Superior is toward
Chequamegon Bay (Bennington et al., 2010), concentrating ice in the bay, hindering mechanical ice breakup. Thus,
removal of ice is reliant on thermal processes as spring surface air temperatures increase. It is also worth noting
that for Chequamegon Bay during 1931-1960 the trends lean toward an earlier breakup date of 1.4 days decade[-1],
highlighting the complex temporal variability. The other site in the United States with a later breakup trend is
Mystery Lake. Here trends of later breakup by 0.2 days decade[-1] for 1961-1990 are contrasted by two adjacent sites
within just 2 km (Lakes Nebish and Escanaba) that show earlier breakup trends of ~1.4 days decade[-1]. There is no
clear explanation for these differences, but given the small magnitude of change at Lake Mystery it may be that it
is responding differently due to site specific factors (e.g. bathymetry).
Freezeup date changes show that a longitudinal split occurs during 1961-1990. Sites are generally dominated by
earlier freezeup trends in the west and later trends in the east (Fig. 4e). The mean trend direction across North
America is for freezeup dates to become earlier by 0.2 days decade[-1]. However, it is clear that there is almost an
equal number of sites showing later freezeup (24 sites) as there are showing earlier freezeup (27 sites). Ten sites
display no trends at all. Whilst lakes show an overall trend direction change to earlier freezeup (0.2 days decade[-1]),
rivers maintained a small trend toward later freezeup (0.04 days decade[-1]), albeit at a considerably reduced
magnitude compared to 1931-1960, and with a split between sites showing different trends (five later freezeup, six
earlier freezeup, and one no trend).
During 1961-1990, 15 sites have continuous data from the first time period, with eight sites maintaining the same
trend direction, and the rest showing changes in trend direction. Of these changes, two sites changed to earlier
freezeup, two changed to later freezeup, and three changed to no trend from 1931-1960 to 1961-1990 (Table 3).
Trends for the full 1931-1990 period show the majority of sites had later freezeup trends, three displayed no trends,





and three showed earlier freezeup (two of which are statistically significant at α=0.1) (Table 3). At most sites, even
if there is no trend direction change between the two periods, during 1961-1990 the magnitude of the trend
experienced is reduced compared to 1931-1960. Similar to breakup, a number of the sites demonstrate that shorter
30-year trends are occasionally superimposed onto longer 60-years trends. Of specific interest is how this
superimposition relates to the two sites demonstrating statistically significant trends toward earlier freezeup, East
Okoboji Lake and George. Whilst these sites show trends indicative of cooling, it is notable that the trends towards
cooling reduce in magnitude through time from 1931-1960 to 1961-1990 – possibly hinting that cooling trends are
beginning to diminish and warming air temperatures are taking longer to be manifest as changes in trend direction
compared to other sites.

| SITE | 1931-1960 | 1961-1990 | 1991-2005 | 1931-1990 | α | 1961-2005 | α | 1931-2005 | α |
|---|---|---|---|---|---|---|---|---|---|
| BRANT | - | 1.6 | 10.0 | - | | 1.3 | | - | |
| CAZENOVIA | 4.0 | 0.5 | 3.8 | 0.0 | | 2.5 | * | 1.1 | + |
| DETROIT | 0.9 | -0.6 | - | 1.1 | * | - | | - | |
| EAST OKOBOJI LAKE | -4.5 | -2.5 | 15.0 | -1.3 | + | 0.2 | | -0.3 | |
| GEORGE | -4.4 | 1.1 | - | -1.8 | + | - | | - | |
| GULL LAKE | 4.5 | 0.3 | - | 2.1 | + | - | | - | |
| KAPUSKASING RIVER | 3.8 | 2.2 | - | 1.8 | * | - | | - | |
| LAKE KEGONSA | - | 0.0 | 4.0 | - | | 1.0 | | - | |
| LAKE MENDOTA | -0.7 | 1.2 | 8.0 | 0.0 | | 2.5 | + | 0.9 | |
| LAKE MONONA | 1.0 | 0.0 | -2.2 | 1.0 | | 1.4 | | 1.3 | * |
| LAKE SUPERIOR AT BAYFIELD | 2.1 | 0.4 | 15.0 | 0.4 | | 4.5 | *** | 2.5 | ** |
| LAKE SUPERIOR AT MADELINE ISLAND | - | 0.4 | 15.0 | - | | 4.5 | *** | - | |
| MAPLE LAKE | - | -3.1 | 14.2 | - | | -1.3 | | - | |
| MIRROR | 1.7 | 1.2 | 11.3 | 0.7 | | 1.3 | | 0.7 | + |
| MOHONK | - | 1.5 | 3.3 | - | | 2.5 | * | - | |
| NORTH SASKATCHEWAN RIVER | 4.3 | -0.4 | - | 2.1 | ** | - | | - | |
| OTSEGO | -3.3 | -0.8 | - | -1.3 | | - | | - | |
| PLACID | 3.1 | 0.0 | -5.7 | 0.5 | | -0.4 | | 0.2 | |
| SHELL LAKE | 0.8 | 0.0 | 10.0 | 1.3 | * | 0.8 | | 1.3 | * |
| SPIRIT LAKE | - | -5.0 | 14.2 | - | | -0.9 | | - | |
| WEST OKOBOJI LAKE | -3.3 | -1.6 | 2.9 | 0.0 | | -0.3 | | 0 | |


**Table 3**: Summary decadal change statistics for sites across North America with freezeup data available for either
the 1931-1990 or 1961-2005. The statistical significance values are for the longer-term 1931-1990 period. Levels
of significance (α) are: *** = 0.001, ** = 0.01, * = 0.05, + = 0.1. The negative value indicates the direction of the
trend, i.e. earlier freezeup.



Changes in the length of the open water season from 1961-1990 (Fig. 4f) show similarity to breakup trends (Fig.
4c). The mean trend shows the open water season was growing by 2.8 days decade$^{-1}$ (Table 2). This is contrasted,
particularly in the western part of the region, with freezeup trends that generally show earlier freezeup dates (Fig.
4e). Similar to some sites in the previous time period, often the magnitude of earlier breakup trends is larger than
earlier freezeup trends, increasing the open water season length (Fig. 5). At several sites changes in open water
season length appears more complex than the trends for individual breakup/freezeup dates. For example, Big
Trough Lake and Frame Lake display no trend and a cooling trend for breakup, respectively, and a cooling trend
and no trend for freezeup, respectively. Despite neither site demonstrating a warming breakup/freezeup trend, the
length of their open water season increased by ~1 day decade$^{-1}$, suggesting a general warming pattern that is more
complicated than its constituent parts.





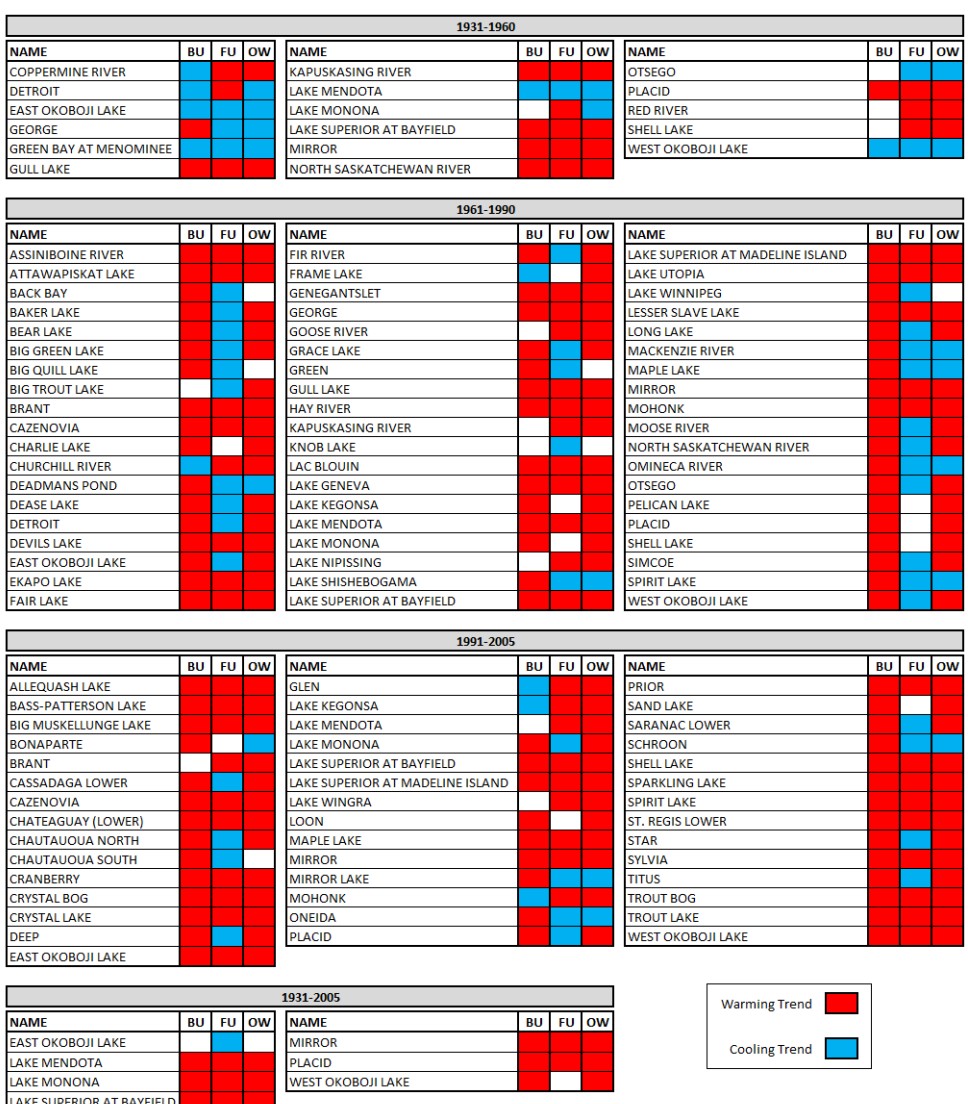


**Figure 5**: Comparison of how sites in North America with an open water season calculation. This reflects the relative changes in dates for breakup and freezeup. The red, blue, and white colours demonstrate whether the calculated trend reflects a warming trend (earlier breakup, later freezeup, or increased open water season), a cooling trend (the opposite), and no trend, respectively. Abbreviations are: BU – breakup, FU – freezeup, and OW open water.


**3.2.3. 1991-2005**





From 1991-2005 most sites display earlier breakup trends (Fig. 4g). Whilst the overall trend of 2.8 days decade$^{-1}$
earlier breakup is reduced compared to 1961-1990 (3.0 days decade$^{-1}$), it is evident that the magnitude of change
has remained similar or increased at a number of sites with 1961-2005 time series. For example, Lake Monona
displays earlier breakup trends of 2.5 days decade$^{-1}$ for 1961-1990 that increases to 7.1 days decade$^{-1}$ earlier breakup
for 1991-2005. There are a small number of sites displaying the opposite pattern – e.g. at Houghtons Pond the
breakup trends were 7.5 days decade$^{-1}$ earlier for 1961-1990 and 1.25 days decade$^{-1}$ later from 1991-2005. This
highlights that whilst earlier breakup trends dominate, this is not temporally consistent at all sites. It is also not
spatially consistent, as is indicated by a number of sites along the eastern seaboard displaying later breakup. These
sites tend to display higher standard deviation values for breakup dates compared to elsewhere, reflecting a greater
annual variability in breakup dates and that shorter trends are more likely to fluctuate. The maritime location of
these sites and adjacent warm ocean currents that propagate past the region likely modulate temperatures so they
are closer to zero and require only minor temperature changes to go between frozen and melting conditions. Thus,
they experience greater variability compared to sites further inland. Importantly, these eastern seaboard sites show
that, when there are data available for 1961-2005, the trend is toward earlier breakup (Table 4) – suggesting later
breakup trends are superimposed onto longer earlier breakup trends.
Freezeup trends from 1991-2005 (Fig. 4h) show a mixed signal, but with a general trend toward later freezeup of
5.1 days decade$^{-1}$ (Table 2). Similar to 1961-1990 there is a longitudinal split, albeit with the caveat that the data
available for 1991-2005 are considerably more spatially-restricted. In the western part of the Great Lakes region
there is an increase in the number of sites showing warming decadal freezeup trends. Sites demonstrating earlier
freezeup are located to the east of the Great Lakes. Of these sites only two have data for the preceding period; Lake
Monona and Placid. These sites show that from 1961-2005 the trends are again mixed, with Lake Monona
displaying later freezeup of 1.4 days decade$^{-1}$ and Placid showing earlier freezeup of 0.4 days decade$^{-1}$ (Table 3).
Some of the sites displaying large magnitude trends towards later freezeup from 1991-2005 also show low
magnitude earlier freezeup trends over the longer 1961-2005 period (e.g. Spirit Lake; Table 3). At East Okoboji
Lake, fluctuations in the magnitude and trend direction are observed across all time periods that result in earlier
freezeup trends of 1.3 days decade$^{-1}$ during 1931-1990 and later freezeup trends of 0.2 days decade$^{-1}$ during the
1961-2005 period. This is marked by a change in direction and magnitude of the trends at this site through time,
indicating at first a reduction in the earlier freezeup trends before switching to later freezeup trends (Table 3). The
variability in trend magnitude and direction is likely due to the combination of factors that control water freezeup



– e.g. even if water temperatures are low enough to freeze, wind and water movement can mechanically prevent
freezeup as the kinetic energy means that it is harder for the water to stabilise or smaller ice patches to agglomerate
(Beltaos and Prowse, 2009), potentially requiring more heat and kinetic energy to be withdrawn – the complexity
involved in allowing water freezeup likely acts as an important control on these fluctuating trends.

| SITE | $\alpha$ | DECADEL CHANGE |
|---|---|---|
| DAMARISCOTTA LAKE | | -2.2 |
| HOUGHTONS (HOOSICWHISICK) POND | + | -3.6 |
| LAKE AUBURN | ** | -3.2 |
| LAKE WINNIPESAUKEE | * | -1.8 |
| MARANACOOK LAKE | * | -2.5 |
| MOHONK | + | -2.3 |
| PONKAPOAG POND | ** | -6.4 |
| SWAN LAKE | * | -2.5 |


**Table 4**: Decadal breakup trends for the 1961-2005 period at sites along the eastern seaboard of the United States
that demonstrate a cooling trend during the 1991-2005 period. Levels of significance ($\alpha$) are: *** = 0.001, ** =
0.01, * = 0.05, + = 0.1. This table shows that, despite the cooling trend during the 1991-2005 period that is observed
at these sites, these trends are superimposed onto a longer-term warming trend. The negative value indicates the
direction of the trend, i.e. earlier breakup.

From 1991-2005 overall trends reflect a widening of the open water season (Fig. 4i) and an increased magnitude
of change to 9.3 days decade$^{-1}$ (Table 2). This warming pattern is reflected in the general patterns of earlier breakup
and later freezeup (Fig. 5). At sites with conflicting trend directions between breakup/freezeup a variable response
in the ice season length is observed. At some sites with a cooling breakup/freezeup trend and an opposing warming
breakup/freezeup trend the changes in the length of the open water season typically reflect the breakup/freezeup
trend that is larger. However, as noted above, this is not always the case (e.g. Bonaparte) and reflects that
interpreting changes in the length of the open water season requires the context of breakup/freezeup changes.

**3.3. Fennoscandia**
**3.3.1. 1931-1960**





In Fennoscandia, apart from northern Finland and northwest Russia, there is a higher density of data compared with
North America (Fig. 6). Breakup dates during 1931-1960 display earlier and later trends, as well as sites exhibiting
no observable trend (Fig. 6a). The only spatial trend is that most sites with warming decadal trends are in southern
Finland and later breakup trends in southern Sweden. Whilst there appears to be a latitudinal split separating the
trend directions in the northern and southern areas of Fennoscandia, the overall values show breakup from 1931-
1960 occurring 0.1 days decade$^{-1}$ earlier. Earlier breakup trends were stronger for rivers (0.2 days decade$^{-1}$) than
for lakes (0.1 days decade$^{-1}$).
During 1931-1960 (Fig. 6b) Finland is predominantly characterised by low magnitude earlier freezeup trends and
much of Sweden shows both warming and cooling trends with no obvious spatial patterns. Mean values show a
general earlier freezeup pattern by 2.1 days decade$^{-1}$. However, this is only the case for lakes as rivers display later
freezeup trends of 3.5 days decade$^{-1}$, suggesting that rivers have responded quicker to increases in air temperature.
During 1931-1960 trends are almost identical between freezeup and open water season length in Finland (Fig. 6b,
6c). In Sweden, trends are varied, reflecting similar heterogeneous patterns of change for breakup/freezeup dates.
Where individual sites demonstrate opposing warming and cooling trends for breakup and freezeup the typical
response in the length of the open water season is to see a relative reduction in its length (Fig. 7), suggesting that
changes in freezeup dates are bigger than breakup date changes, causing the open water season to shift and reduce.
This is reflected by the observation that the majority of sites display cooling trends (Fig. 3) and mean values show
in Fennoscandia a reduction in the open water season by 2.2 days decade$^{-1}$. This is only the case for lakes, as rivers
show strong trends toward an increase in the length of the open water season by 3.4 days decade$^{-1}$ – suggesting that
rivers are responding faster to temperature increases.





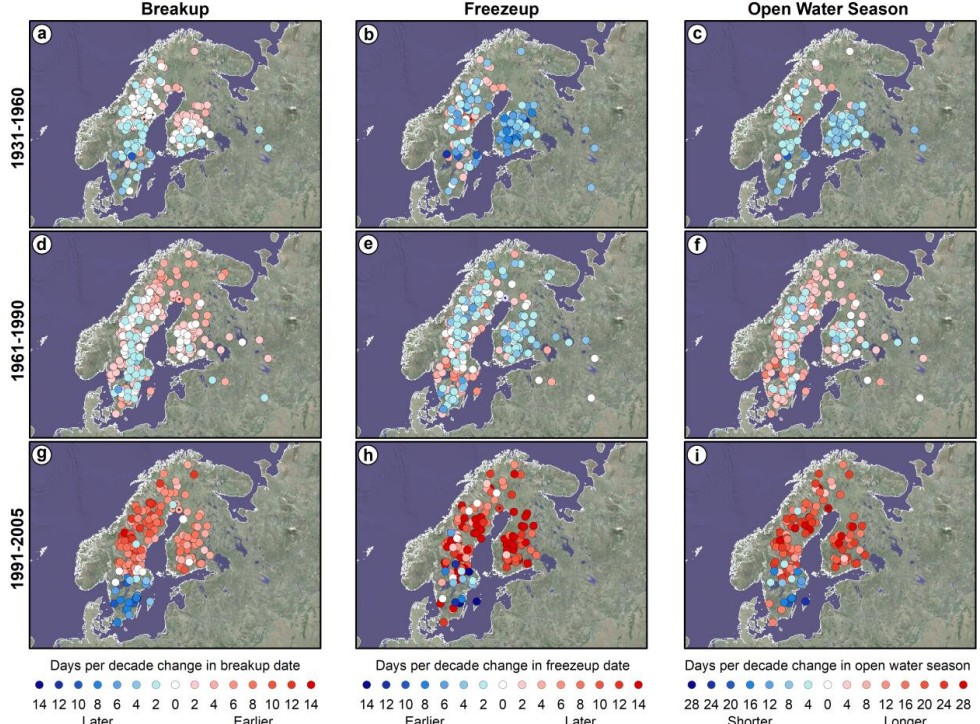

**Figure 6**: Decadal trends for breakup (**a**, **d**, and **g**), freezeup (**b**, **e**, and **h**), and length of the open water season (**c**, **f**, and **i**) in Fennoscandia for the three individual time periods. Sites with a dot in the centre of the circle are river sites. Blue and red tones on the scales related to cooling and warming signals, respectively.

2000





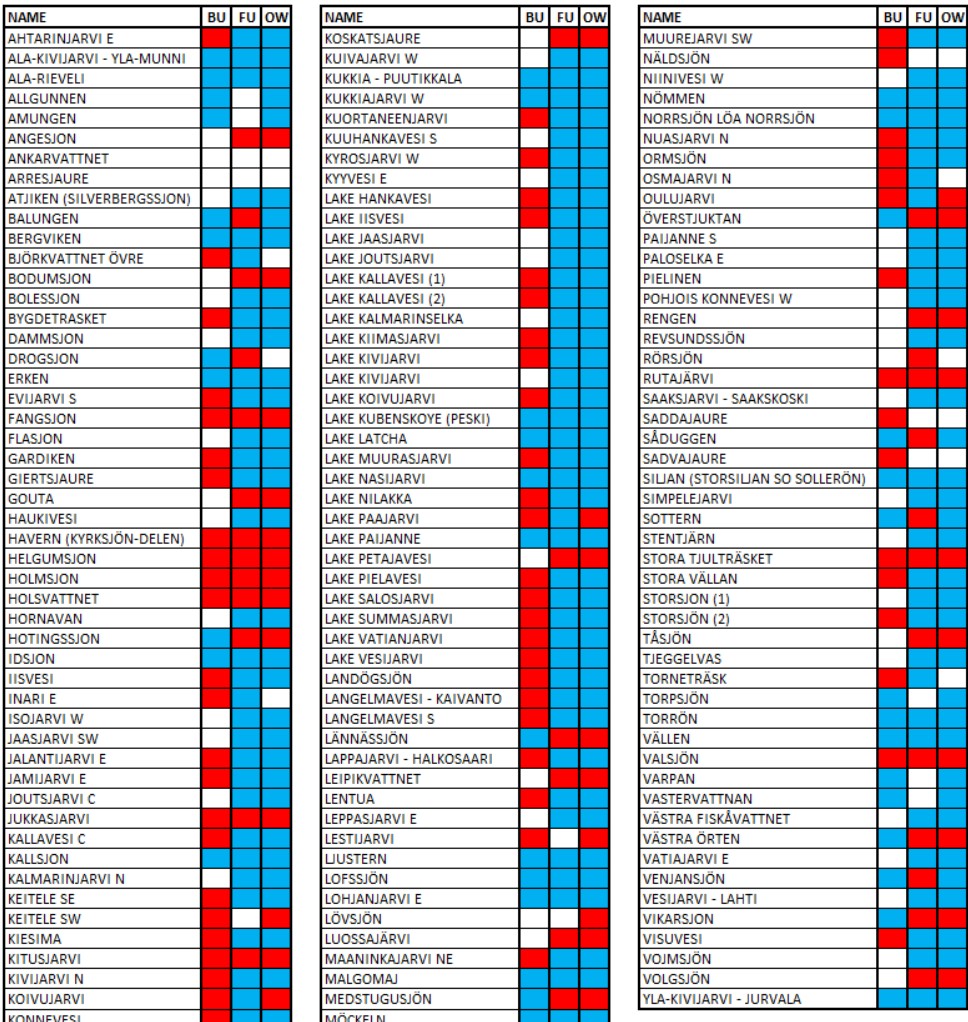

**Figure 7**: Comparison of how sites in Fennoscandia with an open water season calculation. This reflects the relative changes in dates for breakup and freezeup. The red, blue, and white colours demonstrate whether the calculated trend reflects a warming trend (earlier breakup, later freezeup, or increased open water season), a cooling trend (the opposite), or no trend, respectively. Abbreviations are: BU – breakup, FU – freezeup, and OW open water.

### 3.3.2. 1961-1990

By 1961-1990 (Fig. 6d) additional sites are available across the region and more dominant trends have developed. Trends toward earlier breakup in northern Sweden and Finland have increased in magnitude compared to 1931-



1960, whilst many of the trends in southern Sweden that previously indicated later breakup (Fig. 6a) have switched
to earlier breakup. Similar to the patterns observed for 1931-1960, the magnitude of change leading to earlier
breakup is stronger for rivers (2.2 days decade$^{-1}$) than for lakes (0.8 days decade$^{-1}$). These changes are also reflected
in the increased magnitude for all sites toward an earlier breakup date of 0.8 days decade$^{-1}$ from 1961-1990.
In the 1961-1990 period (Fig. 6e) no clear spatial patterns have yet developed, but there is a clear decrease in sites
with earlier freezeup trends compared to 1931-1960, with mean freezeup dates occurring 0.3 days decade$^{-1}$ later.
This is partly due to 89 new sites with freezeup data and many sites with data covering both periods switching from
earlier freezeup to later freezeup, or from earlier freezeup to no trend. This is shown in Table 5 where sites with
data covering 1931-1990 generally show a cooling trend in the 1931-1960 period that either reduced in magnitude
(n=28) or completely switched to no trend (n=11) or a warming trend (n=17). This potentially indicates that rising
air temperatures were beginning to take effect from 1961-1990.


| TYPE OF TREND CHANGE | NUMBER OF SITES |
|---|---|
| Increased magnitude of a cooling trend | 4 |
| Switch to cooling trend from warming trend | 7 |
| Switch to cooling trend from no trend | 1 |
| Decreased magnitude of a cooling trend | 28 |
| Switch to no trend from cooling trend | 11 |
| Switch to no trend from warming trend | 9 |
| Decreased magnitude of a warming trend | 3 |
| Switch to warming trend from no trend | 7 |
| Switch to warming trend from cooling trend | 17 |
| Increased magnitude of a warming trend | 3 |


**Table 5**: The number of sites and different types of changes in decadal trend direction from the 1931-1960 time
period to the 1961-1990 time period in Fennoscandia. This shows that the majority of sites with data covering the
two time periods are either experiencing reduced earlier freezeup trends, or they are completely switching to later
freezeup trends.





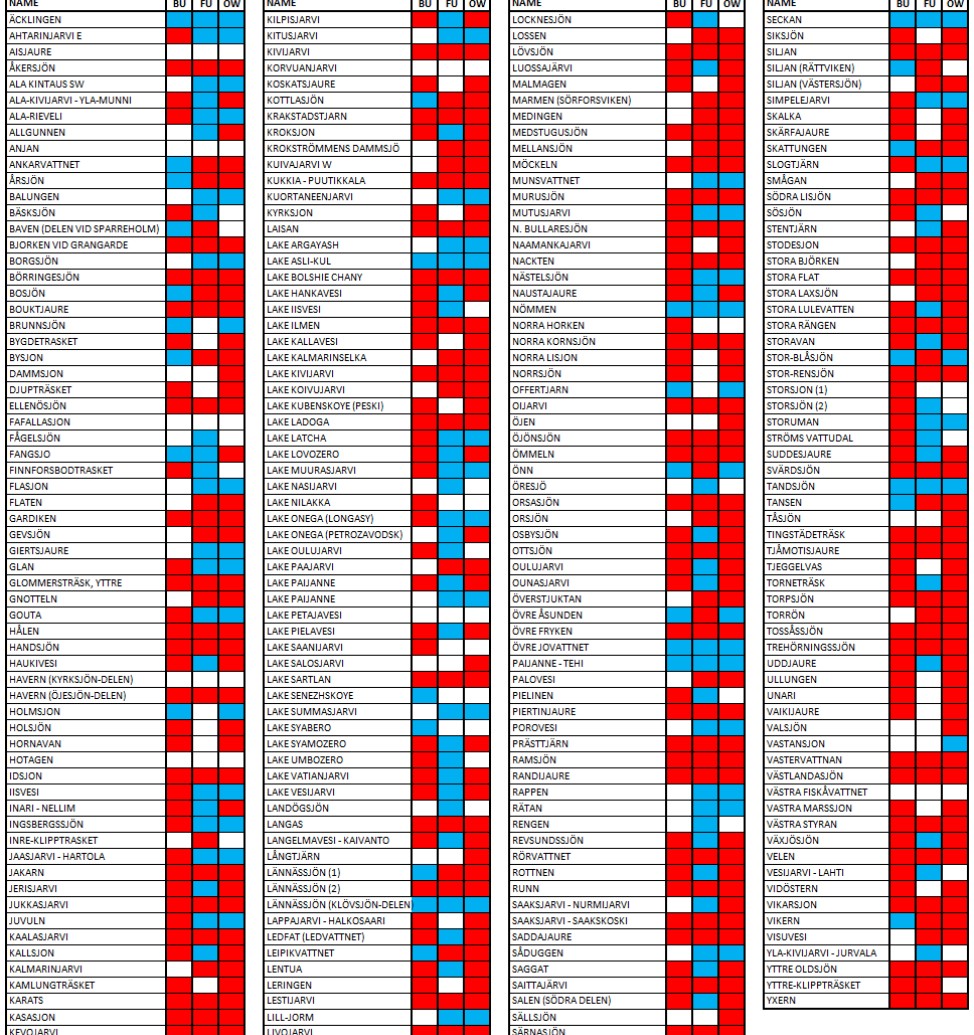

**Figure 8**: Comparison of how sites in Fennoscandia with an open water season calculation for the 1961-1990 time period. This reflects the relative changes in dates for breakup and freezeup. The red, blue, and white colours demonstrate whether the calculated trend reflects a warming trend (earlier breakup, later freezeup, or increased open water season), a cooling trend (the opposite), or no trend, respectively. Abbreviations are: BU – breakup, FU – freezeup, and OW open water.

Across Fennoscandia during the 1961-1990 time period the majority of sites exhibit a trend towards an increase in the open water season length (Fig. 6f), with a mean increase of 1.8 days decade$^{-1}$ (Table 2). This pattern is similar





to breakup (Fig. 6d) but markedly different to freezeup, where many sites in the east demonstrate earlier freezeup
trends (Fig. 6e). During this time period it is clear that breakup trends are changing the most and are driving the
observed changes in the open water season. It is notable that most sites that show a reduction in the open water
season do so because either they have both cooling breakup and freezeup trends, or because cooling freezeup trends
are stronger than warming breakup trends (Fig. 8).

**3.3.3. 1991-2005**
In the 1991-2005 period there is a clear dominance for earlier breakup trends in Sweden, with the only area that
experienced later breakup trends in the south (Fig. 6g). Across all sites in Fennoscandia breakup has occurred
earlier at a rate 3.9 days decade$^{-1}$. Unlike the previous time periods, the magnitude of change was larger for lakes
(3.9 days decade$^{-1}$) than it was for rivers (2.2 days decade$^{-1}$). A key observation from this final time period is that
for the majority of sites that had data for the 1961-1990 period not only have they maintained the earlier breakup
trends but they also increased in magnitude.
A clear latitude distinction has developed at ~60 °N which partitions later and earlier breakup trends to the north
and south, respectively (Fig. 6g). Northern latitudes showed a gradual change from cooling to warming trends but
the south has shown through the different periods a switch in trend direction. Two reasons are proposed for why
this boundary has developed. Firstly, it has been acknowledged, and increasingly observed, that rising global air
temperatures have tended to be amplified at higher latitudes compared to the global mean (Chylek et al., 2009;
Serreze et al., 2000). This pattern has been observed across northern parts of Fennoscandia where temperature
increase since 1950 has been greater than in the south (Hansen et al., 2006). This amplification correlates well with
the changes observed towards earlier ice breakup dates. Secondly, the position of the winter 0 °C isotherm is likely
to have had an important influence on the ice phenology in southern Sweden. As is shown in Fig. 9 the mean
decadal position of the winter 0 °C isotherm has, albeit with some variation, migrated northward between 1931-
2000 – suggesting the colder, perennial temperatures of the Arctic are not penetrating as far south. For the 1991-
2005 period the positon of the 1991-2000 isotherm (Fig. 9) shares a latitudinal position separating warming and
cooling trends. This relationship between the breakup date and the position of the 0 °C isotherm is also reflected in
the standard deviation of breakup date (used as a measure of interannual variability), which shows that the sites
with the greatest interannual variability are south of or around the 0 °C winter isotherm (Fig. 10). This association





is not surprising as sites that are situated near the 0 °C isotherm will require only small temperature changes to
move between melting and freezing conditions. Thus, short term weather events have a greater influence on the
breakup date when temperature conditions are already close to 0 °C – unlike at higher latitudes where the
temperature changes need to be sustained for longer to bring about breakup.

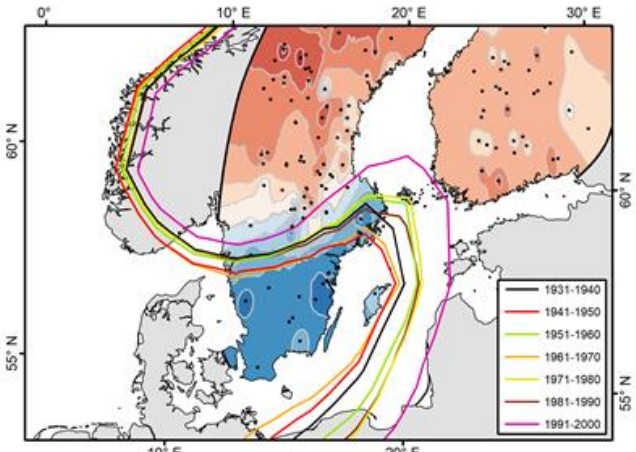


**Figure 9**: Breakup date trends in southern Sweden from 1991-2005 (see Fig. 6g for full area and scale). Note that
site values have been interpolated into a surface for easier comparison. The polylines show the mean position of
the 0 °C winter isotherm for several decades. Calculated from the 20[th] Century Reanalysis project (cf. Compo et
al., 2011).

The amplification of high latitude temperature increases and the position of the 0 °C isotherm can, perhaps explain
why there is a clear boundary between the climatic response of the different regions, but it is insufficient to explain
why the trends are opposing. Breakup dates at sites in this region show a pattern where initial trends toward later
breakup occurred from 1991-1996 before they became earlier after 1998. The difference in the slopes for later and
earlier segments of the time series for breakup are such that the steeper gradient for later breakup from 1991-1996
was large enough to override the shallower gradient for earlier breakup in the second half of the period – thus,
resulting in an overall trend from 1991-2005 of later breakup dates. As southern Swedish climate is strongly
correlated to the NAO, the decreasing temperature trend leading to later breakup is associated with a weakening of
the NAO in the early 1990s (Blenckner et al., 2004). Therefore, a reduction in the pressure gradient (a weakened



NAO) allowing colder Arctic air to descend southward could have shifted temperatures already close to 0 °C below
freezing.

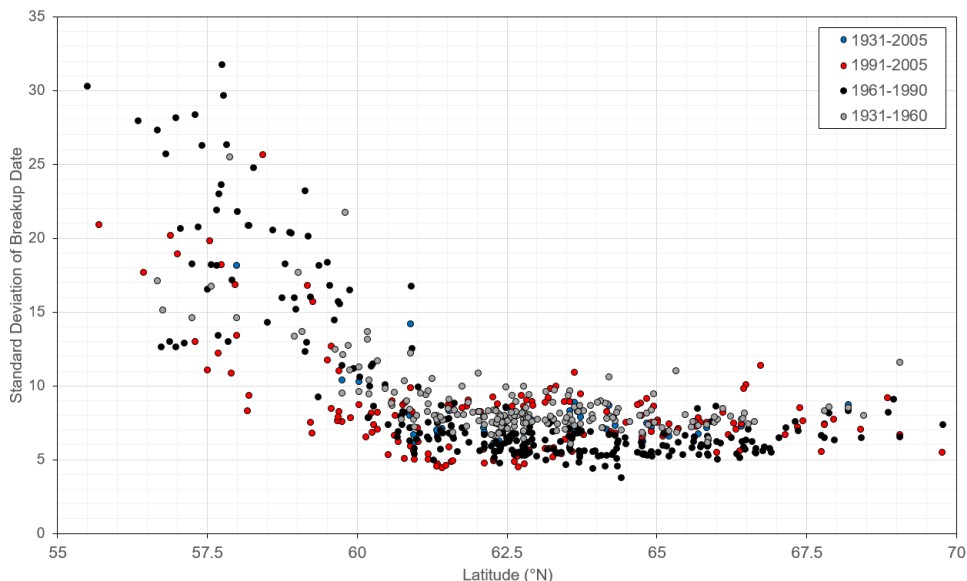


**Figure 10**: Standard deviation for breakup dates at Fennoscandian sites for different time periods. Note that, in
accordance with Fig. 9, the sites that demonstrate the largest interannual variability are south of 60 °N and appear
to be associated with the winter 0 °C isotherm.
By exploring the individual time series for sites in this area in more detail and over a longer timescale, it becomes
clear that these trends towards later breakup in southern Sweden during 1991-2005 are short-term and
superimposed on top of long-term warming trends and earlier breakup. Of the sites in southern Sweden that
demonstrate a cooling trend for 1991-2005, one site has continuous data covering 1931-2005. At Såduggen the
breakup dates were occurring 1.5 days decade$^{-1}$ earlier during 1931-2005 even though the most recent time period
shows strong cooling trends (see section 3.5; Fig. 9). For the southern sites 12 have data covering 1961-2005 and
show earlier breakup trends of 3.6-5.0 days decade$^{-1}$ (Table 6). At 11 of the 12 sites analysed the earlier trends in
breakup date were significant at or above the $\alpha = 0.1$ significance level with several being significant at $\alpha = 0.01$
(Table 6). At two other sites in Fennoscandia there are trends towards later breakup during 1991-2005. However,
as above for the sites at lower latitudes, the Djupträsket and Sösjön lakes show that when breakup trends are
investigated over the 1961-2005 period they display earlier breakup of 1.6 and 0.8 days decade$^{-1}$, respectively.
Again, this highlights that recent cooling trends appear to be short term fluctuations of a longer warming trend.



| SITE | α | DECADEL CHANGE |
|---|---|---|
| ASPEN | *** | -4.4 |
| FAFALLASJON | * | -4.3 |
| FANGSJO | | -3.6 |
| GNOTTELN | * | -4.1 |
| KRAKSTADSTJARN | ** | -5.3 |
| NORRA LISJON | ** | -3.7 |
| ÖNN | + | -3.6 |
| SÅDUGGEN | * | -3.9 |
| SÖDRA LISJÖN | ** | -3.8 |
| TINGSTÄDETRÄSK | * | -4.4 |
| TREHÖRNINGSSJÖN | * | -5.0 |
| VIKERN | * | -5.0 |

**Table 6**: Decadal breakup trends for the 1961-2005 period at sites in Sweden that demonstrate a cooling trend during the 1991-2005 period. Levels of significance (α) are: *** = 0.001, ** = 0.01, * = 0.05, + = 0.1. This table shows that, despite the cooling trend during the 1991-2005 period that is observed at these sites, these trends are superimposed onto a longer-term warming trend. The negative value indicates the direction of the trend, i.e. earlier breakup.

From 1991-2005 (Fig. 6h) freezeup trends closely resemble those for breakup (Fig. 6g and 9) and the development of a latitudinal boundary at ~ 60 °N, albeit slightly less consistent. The main difference in this region for freezeup patterns compared to breakup patterns is that the southern Sweden sites do not display a completely uniform pattern of cooling, with the western area showing later freezeup trends similar to those observed elsewhere in Fennoscandia. At the sites with earlier freezeup patterns for 1961-2005 it can be shown that these cooling trends are again short term fluctuations on a longer-term warming trend (Table 7). As described above for breakup, the southern latitudes display greater variation due to their proximity to the 0 °C isotherm where only small temperature changes are required to move between frozen and unfrozen conditions (assuming other parameters, such as wind, allow freezing to occur).




| SITE | α | DECADEL CHANGE |
|------|---|----------------|
| **GNOTTELN** | | 3.3 |
| **KALLSJON** | | 1.8 |
| **KYRKSJON** | + | 3.6 |
| **NORRA LISJON** | * | 3.7 |
| **SILJAN (VÄSTERSJÖN)** | + | 3.5 |
| **SÖDRA LISJÖN** | * | 3.6 |
| **TINGSTÄDETRÄSK** | * | 5.4 |
| **VELEN** | ** | 4.9 |


**Table 7**: Decadal freezeup trends for 1961-2005 at sites in Sweden that demonstrate a cooling trend during the
1991-2005 period. Levels of significance (α) are: *** = 0.001, ** = 0.01, * = 0.05, + = 0.1. This table shows that,
despite the cooling trend during the 1991-2005 period that is observed at these sites, these trends are superimposed
onto a longer-term warming trend. The positive value indicates the direction of the trend, i.e. later freezeup.

The open water season length from 1991-2005 shows ubiquitous trends, with the majority of sites (Fig. 6i) showing
an increased length and a mean increase of 13.3 day decade$^{-1}$ (Table 2). Sites demonstrating a reduction in the
length of the open water season are typically the same sites that experienced earlier freezeup during this period
(Fig. 6h). Sites in southern Sweden with a reduced open water season length from 1991-2005, show, when
compared against the 1961-2005 time series that all sites have longer-term trends for an increased water season
length (Table 8). One of these sites (Såduggen) also has data for 1931-1960 and shows that across the entire time
period these short-term fluctuations of a reduced open water season are superimposed on top of a longer-term
increase (see section 3.5; Fig. 11). What is clearly noticeable from comparing Fig. 7, 8, and 11 is that through time
more sites are displaying warming patterns of earlier breakup, later freezeup, and increased open water season
lengths, suggesting that these records provide compelling evidence that ongoing surface temperature changes have
led to concomitant changes in lake and river ice phenology patterns.








| SITE | α | DECADEL CHANGE |
|---|---|---|
| GNOTTELN | * | 0.7 |
| KRAKSTADSTJARN | ** | 1.1 |
| KYRKSJON | ** | 0.7 |
| NORRA LISJON | ** | 0.8 |
| SÅDUGGEN | * | 0.6 |
| SILJAN (VÄSTERSJÖN) | ** | 0.8 |
| SÖDRA LISJÖN | ** | 0.8 |
| TINGSTÄDETRÄSK | ** | 0.9 |
| VELEN | *** | 0.9 |


**Table 8**: Decadal trends in the open water season length for 1961-2005 at sites in southern Sweden that demonstrate a cooling trend during the 1991-2005 period. Levels of significance (α) are: *** = 0.001, ** = 0.01, * = 0.05, + = 0.1. This table shows that, despite the cooling trend during the 1991-2005 period that is observed at these sites, these trends are superimposed onto a longer-term warming trend. The positive value indicates the direction of the trend, i.e. increased length of the open water season

606





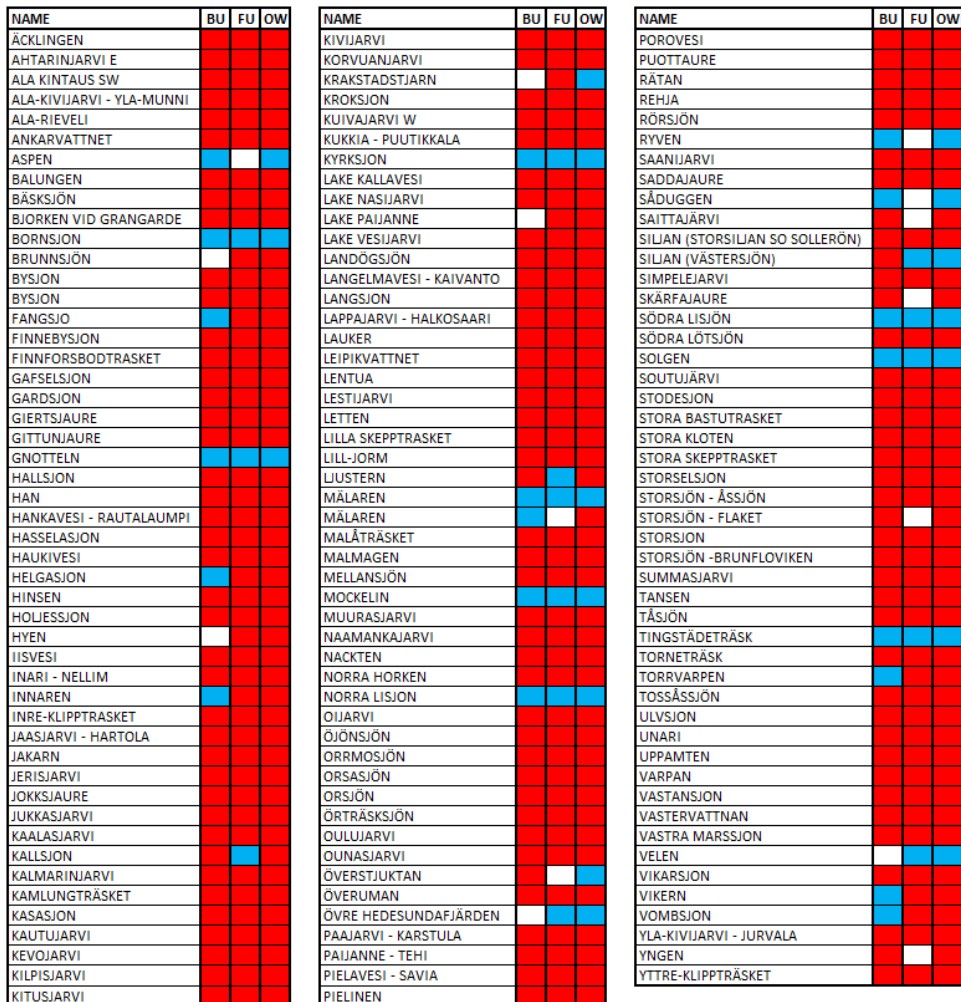

607

**Figure 11**: Comparison of how sites in Fennoscandia with an open water season calculation for the 1991-2005 time

period. This reflects the relative changes in dates for breakup and freezeup. The red, blue, and white colours

demonstrate whether the calculated trend reflects a warming trend (earlier breakup, later freezeup, or increased

open water season), a cooling trend (opposite), or no trend, respectively. Abbreviations are: BU – breakup, FU –

freezeup, and OW open water.


**3.4. Russia**


The sparse availability of Russian data means that broad spatiotemporal trends are not possible. Only a few sites

with breakup, freezeup, or open water data are covered by both the 1931-1960 and 1961-1990 periods. During



1991-2005 data are only available for Lake Baikal, which is discussed above and displayed on Fig. 3. Sites with
data available for 1931-1990 periods generally show trends towards later breakup during 1931-1960 of 0.8 days
decade$^{-1}$ that have changed towards earlier breakup of 0.8 days decade$^{-1}$ from 1961-1990 (Table 2 and 9; Figs. 12a,
12b). However, the limited spatiotemporal availability of the data make any inferences somewhat spurious on a
large scale. Freezeup trends show a similar switch from earlier freezeup trends of 0.8 days decade$^{-1}$ in the earlier
time period to either later freezeup trends or no trends in the latter period (Table 2 and 9; Figs. 12c, 12d).

| SITE | BREAKUP | | FREEZEUP | | OPEN WATER | |
|---|---|---|---|---|---|---|
| | 1931-60 | 1961-90 | 1931-60 | 1961-90 | 1931-60 | 1961-90 |
| LAKE BAIKAL | 1.1 | 0.0 | -1.8 | 3.0 | -2.1 | 3.3 |
| LAKE BAYKAL (P.BUKHTA) | -0.6 | -0.9 | 2.5 | -2.1 | 3.3 | -1.7 |
| LAKE KUBENSKOYE (PESKI) | 1.4 | -1.2 | -3.9 | 0.0 | -5.8 | 1.5 |
| LAKE LATCHA | 0.6 | -1.5 | -3.0 | -2.8 | -4.4 | -2.3 |
| LAKE SENEZHSKOYE | - | - | -3.3 | 0.0 | - | - |
| LAKE TELETSKOYE | 1.7 | -2.0 | -2.0 | 1.9 | -3.3 | 3.6 |


**Table 9**: Summary statistics sites across Russia with data available for more than one time period. The red and blue
colouring indicates whether the decadal trend is a warming or cooling trend, respectively.

Only five sites have sufficient data to generate the open water season and it demonstrates a mixed climate signal.
From 1931-1960 the majority of sites display increased ice season length due to both later breakup and earlier
freezeup (Table 9). During 1961-1990 most sites show an increased open water season length due to earlier breakup
and later freezeup. However, this is not the case for all sites as two sites demonstrating earlier freezeup trends for
1961-1990 have a larger magnitude of change than earlier breakup trends (Table 9). This means that whilst breakup
is becoming earlier, freezeup is also becoming earlier but at a faster rate, resulting in an increase in the ice season.
Data for 1991-2005, and therefore the full 1931-2005 time period, are only available for Lake Baikal (Fig. 13;
Table 9). This shows that the breakup date occurred earlier over the course of the 1931-1960 and 1991-2005
periods, with the intervening 1961-1990 period demonstrating no trend. Freezeup data for 1991-2005 show a trend
towards later freezeup of 5.0 days decade$^{-1}$. This site shows large variability in breakup/freezeup through each of
the time periods, resulting in fluctuations in the length of the open water season through time, with the 1991-2005
period showing a reduction in the number of open water days.



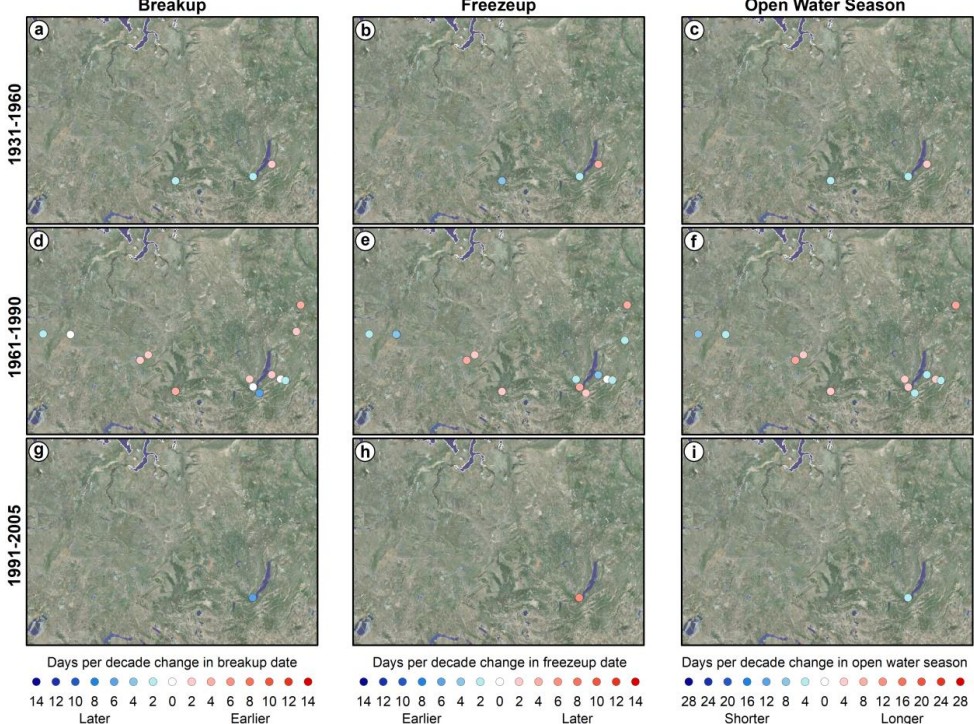

**Figure 12**: Decadal trends for breakup (**a**, **d**, and **g**), freezeup (**b**, **e**, and **h**), and length of the open water season (**c**,
**f**, and **i**) in Russia for the three individual time periods. Blue and red tones on the scales related to cooling and
warming signals inferred from changes in the open water season length, respectively.

**3.5. Sites with continuous data – 1931-2005**
**3.5.1. Breakup**
Decadal changes in breakup dates for each site with data for 1931-2005 suggest large-scale and broadly uniform
hemispheric changes (Fig. 13a). From 1931-1960 period there is considerable heterogeneity in the decadal changes
that have been observed, but by 1961-1990 the response becomes considerably more homogenous as more sites
demonstrate earlier breakup trends. In the 1991-2005 period there is an increase in the magnitude of the warming
trends for the vast majority of sites (Fig. 13a) – i.e. a general ramping up in the magnitude of warming trends
through time. During each of the latter two periods the dominant trend toward warming is, however, not reflective
of all sites. Indeed, a number of sites, such as Ponkapoag Pond in North America and Såduggen in Fennoscandia,
demonstrate a high magnitude decadal change towards a later breakup date that is out of synchrony with the patterns





observed in the preceding time periods – highlighting a temporally complex response. It is notable at these sites
that despite these fluctuations there is a longer-term shift towards earlier breakup dates at both sites – i.e. short-
term trends superimposed on to longer-term trends.
When the decadal trends for all 88 sites are plotted for 1931-2005 the majority (84%) demonstrate earlier breakup
decadal trends. The remaining sites suggest no trend in decadal change for breakup dates, whilst none suggest that
breakup dates became later. For the 44 North American sites that cover 1931-2005, 42 of them display earlier
breakup trends with a mean of 0.7 days decade$^{-1}$. The other two sites display no clear decadal trend. Across
Fennoscandia there are 43 sites with continuous data covering 1931-2005. Whilst these sites are generally restricted
to the southern part of Finland, in Sweden they cover a full transect across the length of the country. Of these 43
sites, none show a trend towards later breakup dates, 11 show no trends, and the rest are showing earlier breakup.
Broadly across Fennoscandia there is a shift toward earlier breakup dates of 0.5 days decade$^{-1}$.





| NORTH AMERICA | | | | |
| --- | --- | --- | --- | --- |
| STUDY SITE | 1931-60 | 1961-90 | 1991-05 | 1931-05 |
| AZISCOHOS LAKE | -0.6 | -4.2 | -1.7 | -0.3 |
| BLACK OAK LAKE | -0.8 | -1.3 | -6.4 | -0.5 |
| BONAPARTE | -1.7 | -5.0 | -0.9 | -0.6 |
| CAZENOVIA | 0.0 | -5.7 | -4.4 | -1.2 |
| COBBOSSEECONTEE LAKE | -0.9 | -4.3 | 0.0 | -0.9 |
| DAMARISCOTTA LAKE | -0.6 | -5.7 | -2.2 | -0.9 |
| DETROIT | 1.8 | -1.9 | 0.0 | -0.5 |
| EAST OKOBOJI LAKE | 5.4 | -5.3 | -2.9 | 0.0 |
| EMBDEN POND | 0.0 | -3.8 | -6.3 | -0.3 |
| FIRST CONNECTICUT LAKE | -1.3 | -2.9 | -6.7 | 0.0 |
| GEORGE | -1.4 | -2.9 | 1.1 | -0.5 |
| HOUGHTONS (HOOSICWHISICK) POND | -8.3 | -7.5 | 1.3 | -1.4 |
| KEZAR LAKE | -1.4 | -2.9 | -5.0 | -0.6 |
| LAKE AUBURN | -0.6 | -4.3 | 1.1 | -1.0 |
| LAKE MENDOTA | 2.5 | -4.3 | 0.0 | -0.9 |
| LAKE MONONA | 0.0 | -2.5 | -7.1 | -1.4 |
| LAKE SUPERIOR AT BAYFIELD | -2.6 | -2.1 | -3.3 | -2.0 |
| LAKE WINNIPESAUKEE | -1.0 | -2.5 | 1.7 | -0.2 |
| LOON | 4.1 | -5.0 | -2.3 | -0.7 |
| MARANACOOK LAKE | 0.0 | -4.3 | 4.3 | -0.7 |
| MINNETONKA | 1.0 | -2.5 | -2.5 | -0.7 |
| MINNEWASKA | 3.1 | -1.7 | -3.3 | -0.3 |
| MIRROR | -0.6 | -3.1 | -3.6 | -0.3 |
| MOHONK | -0.8 | -5.6 | 2.0 | -0.6 |
| MOOSEHEAD LAKE | 0.0 | -3.3 | -4.0 | -0.2 |
| ONEIDA | 1.1 | -3.3 | -2.2 | -0.2 |
| OTSEGO | 0.0 | -7.3 | 0.0 | -0.7 |
| PLACID | -1.4 | -3.3 | -5.0 | -0.5 |
| PONKAPOAG POND | -11.5 | -10.0 | 6.3 | -2.6 |
| PORTAGE LAKE | 0.0 | -3.8 | -3.8 | -0.6 |
| RAINY | 0.0 | -3.0 | -2.7 | -0.8 |
| RICHARDSON LAKE | -1.1 | -2.7 | -3.8 | -0.1 |
| ROCK LAKE | 3.2 | -4.3 | 1.4 | -1.2 |
| SCHROON | -1.4 | -3.3 | -1.1 | -0.9 |
| SEBEC LAKE | 0.0 | -2.7 | -6.3 | -0.4 |
| SHELL LAKE | 0.0 | -1.3 | -6.7 | -0.6 |
| SKIFF LAKE | -1.2 | -3.3 | -4.4 | -1.0 |
| SUNAPEE LAKE | -0.6 | -4.7 | 0.0 | -0.8 |
| SWAN LAKE | 0.0 | -5.7 | 2.0 | -0.6 |
| UMBAGOG LAKE | -1.1 | -2.8 | -5.6 | -0.3 |
| VERMILION | 0.9 | -2.6 | -4.2 | -0.2 |
| WEST GRAND LAKE | 0.0 | -3.8 | -5.0 | -0.3 |
| WEST OKOBOJI LAKE | 6.0 | -3.8 | -6.9 | -0.3 |
| WHITE BEAR | 1.4 | -2.9 | 1.4 | -0.2 |

| FENNOSCANDIA | | | | |
| --- | --- | --- | --- | --- |
| STUDY SITE | 1931-60 | 1961-90 | 1991-05 | 1931-05 |
| ÄCKLINGEN | 0.9 | 1.0 | -7.5 | -0.3 |
| ALA-KIVIJARVI - YLA-MUNNI (1489) | 1.2 | -0.6 | -7.5 | -0.5 |
| ALA-RIEVELI | 0.4 | -0.6 | -3.3 | -0.6 |
| ANKARVATTNET | 0.0 | 0.5 | -5.6 | 0.0 |
| BALUNGEN | 1.7 | 0.0 | -8.3 | 0.0 |
| DAMMSJON | 0.0 | 0.0 | -2.5 | -0.5 |
| GIERTSJAURE | -0.8 | 0.0 | -7.1 | -0.4 |
| HAUKIVESI | 0.0 | -0.8 | -1.4 | -0.6 |
| INRE-KLIPPTRASKET | 0.0 | 0.0 | -6.7 | 0.0 |
| JAASJARVI SW | 0.0 | -0.6 | -5.0 | -0.8 |
| JUKKASJARVI | -1.0 | -3.8 | -4.3 | -0.4 |
| KALLSJON | 1.3 | -0.8 | -14.0 | -1.1 |
| KITUSJARVI (3548) | -1.9 | -1.4 | -5.7 | -0.8 |
| KUKKIA - PUUTIKKALA (3512) | 0.4 | -1.1 | -4.3 | -0.8 |
| LAKE NASIJARVI | 1.0 | 0.0 | -1.5 | -0.8 |
| LANDÖGSJÖN | -2.1 | 0.0 | -9.2 | -0.2 |
| LANGELMAVESI - KAIVANTO (3506) | -1.0 | -1.4 | -2.5 | -1.1 |
| LAPPAJARVI - HALKOSAARI (4703) | -0.7 | -0.5 | -5.0 | -0.9 |
| LEIPIKVATTNET | 0.0 | 2.0 | -7.8 | 0.0 |
| LEJ DA SAN MUREZZAN | -2.9 | 1.1 | 0.0 | -0.2 |
| LENTUA | -1.1 | -2.0 | -5.0 | -1.0 |
| LESTIJARVI | -1.5 | -0.5 | -3.3 | -0.8 |
| LILLA SKEPPTRASKET | 0.0 | -1.1 | -8.6 | 0.0 |
| MALMAGEN | -0.8 | -1.1 | -6.7 | -0.7 |
| NACKTEN | -0.2 | -0.9 | -7.5 | -0.4 |
| NORRA HORKEN | 0.4 | -1.0 | -8.8 | -0.7 |
| OULUJARVI | -2.4 | -2.0 | -6.0 | -1.0 |
| ÖVERSTJUKTAN | 0.6 | 0.0 | -5.6 | 0.0 |
| PIELINEN | -2.5 | -2.3 | -3.3 | -0.8 |
| RIVER TORNIONJOKI | -0.5 | -2.6 | 0.0 | -0.3 |
| SADDAJAURE | -0.4 | -3.0 | -10.0 | -0.7 |
| SÄDUGGEN | 1.4 | 0.0 | 7.5 | -1.5 |
| SILJAN | 0.9 | -1.3 | -3.6 | -0.8 |
| SIMPELEJARVI | 0.0 | -1.3 | -3.0 | -0.9 |
| STORSJON | 0.0 | -0.5 | -5.4 | 0.0 |
| TÅSJÖN | 0.0 | 0.0 | -10.0 | 0.0 |
| TORNEÄLVEN VID HAPARANDA | -1.0 | -2.5 | -2.0 | 0.0 |
| TORNETRÄSK | -2.1 | -3.8 | -10.0 | -1.2 |
| VASTERVATTNAN | 1.9 | -1.5 | -6.0 | 0.0 |
| VIKARSJON | 0.7 | -1.7 | -7.5 | -0.4 |
| VISUVESI | -1.4 | 0.0 | -6.3 | -0.7 |
| YLA-KIVIJARVI - JURVALA | 0.8 | 0.0 | -5.0 | -0.3 |
| YTTRE-KLIPPTRÄSKET | -1.3 | -1.2 | -7.7 | 0.0 |

| RUSSIA | | | | |
| --- | --- | --- | --- | --- |
| STUDY SITE | 1931-60 | 1961-90 | 1991-05 | 1931-05 |
| LAKE BAIKAL | 1.1 | 0.0 | 5.0 | 0.0 |

EARLIER — LATER
-14    0    14    (a)
CHANGE IN BREAKUP DATE IN DAYS PER DECADE

(b)

(c)

Days per decade change in breakup date
3 2 1 0 1 2 3
Later      Earlier


**Figure 13**: Summary of evidence for the long-term breakup date trends for the 1931-2005 period: **a)** heat table demonstrating the decadal change for each site during each time period. The colouring of each cell shows the relative magnitude of that change compared to other sites and time periods; **b-c)** spatial pattern of the decadal trends



for North America and Fennoscandia during the 1931-2005 time period. Two sites are not displayed on the maps:
Lej da San Murezzan (location shown on Fig. 1a) and Lake Baikal (location Fig. 12a).

In Fennoscandia, during the earliest two time periods the magnitude of earlier river breakup dates was shown to be
larger than for lakes until the final 1991-2005 period. When these individual trends are explored for 1931-2005 it
can be shown that on a longer timescale lakes experienced a greater change in breakup dates, on average 0.6 days
decade$^{-1}$ earlier, compared to 0.2 days decade$^{-1}$ earlier for rivers. Whilst acknowledging the caveats of a limited
number of sites, the above evidence suggests that during the early and middle part of the 20$^{th}$ century rivers were
responding to increased surface air temperatures faster than lakes. This may be explained, possibly, by the river
flow gradient causing waves and ripples which instigates air turbulence and greater interaction of water and air
causing a faster transfer of atmospheric heat. Whilst ripples and waves do form on still water bodies, this is likely
limited compared to actively flowing rivers, causing a slower response time in lake temperatures to air temperature
increases. As the lakes gradually experience this warming the same reasons may also restrict heat exchange from
the lake to the atmosphere. Though the physics require further study, it is possible this thermal legacy allowed lakes
to gradually become a heat sink and might explain why over longer timescales the lakes begin to demonstrate larger
changes than for rivers.
In Russia only Lake Baikal has data for 1931-2005, which shows no trend in the breakup date. Lake Baikal has
been the focus of other studies investigating changes in ice phenology at timescales over 100 years. This showed
that breakup was occurring earlier by 5.1-5.2 days century$^{-1}$ (Livingstone, 1999; Magnuson et al., 2000b). Through
different parts of these time series it has been shown that the magnitude of change varied by as much as 30 days
century$^{-1}$ but this was not sustained through the duration of the time period (Livingstone, 1999; Magnuson et al.,
2000b).

**3.5.2. Freezeup**
Decadal trend changes for the 48 sites with freezeup dates for 1931-2005 show a less clear picture compared to
breakup (Fig. 14a). From 1931-1960 there is a mixed pattern with most sites displaying either a warming or cooling
trend in the date of freezeup. When this is compared to 1961-1990 there is similar dichotomy in the patterns





observed, albeit with a small reduction in the number of sites with cooling decadal change (down from 28 sites

during 1931-1960 to 23 sites during 1961-1990). For many of sites, particularly those that had a cooling pattern

from 1931-1960, the magnitude of the cooling decadal change is reduced during 1961-1990. From 1991-2005 there

is a clear increase in the number and magnitude of warming trends. The decadal trends for 1931-2005 do not display

any clear spatial pattern (Fig. 14a). Of the sites available with 1931-2005 freezeup dates, 17 show a cooling and 22

a warming decadal trend, with the rest displaying no trend. It is noticeable that when the decadal patterns are

considered on longer timescales, the magnitude of the trends are also considerably reduced when compared to each

of the three shorter time periods. Sites with a decadal trend value between 1 and -1 days decade$^{-1}$ account for ~73%

of the sites – thus, highlighting that freezeup trends are clearly more complex than those observed for breakup.

Russia is limited to just the site at Lake Baikal, which trends towards later freezeup of 1.0 days decade$^{-1}$, showing

a gradual switch from cooling conditions to warmer conditions that increases in magnitude through time (Fig. 14).

In North America, whilst a majority of sites display a warming decadal trend for the freezeup dates during the

1931-2005 period, owing to the limited number of sites and their wide geographical spread it is impossible to draw

any clear conclusions (Fig. 14b). Long-term trends across North America are restricted to nine sites around the

Great Lakes. Of these sites, seven display long-term warming trends towards later freezeup, whilst West and East

Okoboji Lake demonstrate no trend and a cooling trend, respectively (Fig. 14). The trends at both of these sites

have shown a gradual transition from earlier freezeup to later freezeup. At East Okoboji Lake this progression

toward later freezeup is reflected in the long-term trends discussed above for 1961-2005 (Table 3). Thus, it is likely

that the large magnitude trend towards earlier freezeup experienced in the 1931-1960 is perhaps large enough to

skew the trend in a direction that is not representative of the more recent changes at the study site. The same can

be said of West Okoboji Lake where freezeup dates have become later.


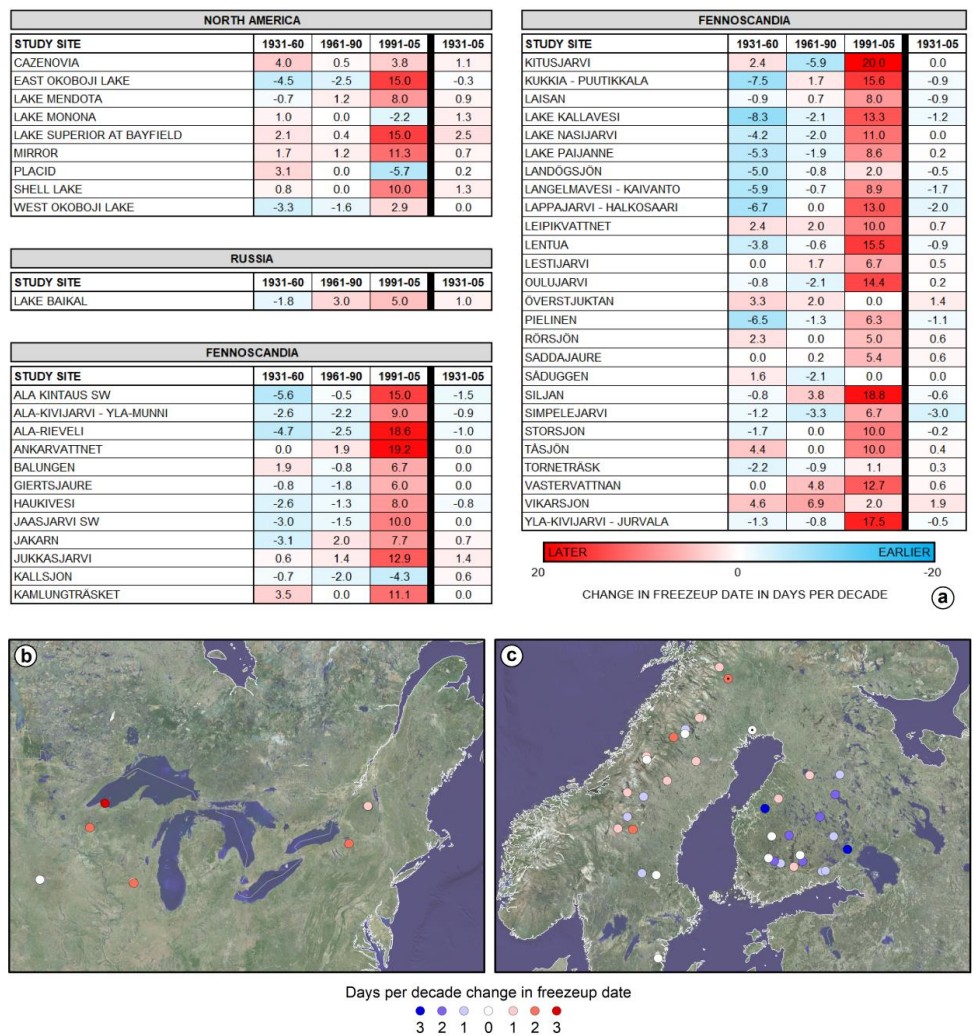

718

**Figure 14**: Summary of evidence for the long-term freezeup date trends for the 1931-2005 period: a) heat table

demonstrating the decadal change for each site during each time period. The colouring of each cell shows the

relative magnitude of that change compared to other sites and time periods; b-c) spatial pattern of the decadal trends

for North America and Fennoscandia during 1931-2005. One site is not displayed on the maps, Lake Baikal, which

is located on Fig. 12a.

---

In Fennoscandia, there are 38 sites with data for 1931-2005 that show a somewhat varied pattern (Fig. 6h and 14).

A total of 16 sites display earlier freezeup trends that are concentrated in southern Finland. There are 14 sites

displaying trends towards later freezeup that are primarily concentrated along the western margin of Sweden. Eight

sites display no trends at all. This heterogeneous pattern is markedly different to breakup trends (Fig. 13) and is



perhaps not surprising given the conditions that are required for ice crystal formation. Thus, breakup is dominated
by thermal characteristics of the climate whilst freezeup is a result of not just the thermal properties of the
environment but also water kinetics. This likely explains why the breakup and freezeup patterns do not simply
reflect observed increases in air temperatures.

**3.5.3. Open water season**
The decadal patterns for the number of annual open water days from 1931-1960 generally indicate a reduced open
water season (Fig. 15a). However, by 1961-1990 a systematic change causes the majority of sites (~73%) to display
an increase in the number of annual open water days. This swing from cooling patterns to warming patterns is
typically of an order of several days per decade. By 1991-2005 the number of sites with an increased open water
season increases to 93%. As the number of open water days encapsulates the relative changes in breakup/freezeup
dates, the increased magnitude shown above for breakup/freezeup (Figs 13a, 14a) is also captured in the open water
season length, with many sites demonstrating an order of magnitude increase in the warming trend (Fig. 15a).
For 1931-2005 the longer-term record shows that warming decadal trends account for 23 of the 37 sites. All of the
sites displaying a cooling trend for 1931-2005 show that in the shorter time periods that comprise this longer-term
record there is common pattern where cooling trends in the 1931-1960 period reduce in magnitude during the 1961-
1990 period, before reversing and becoming strong warming trends in the 1991-2005 period. As with the patterns
described above for freezeup, the limited availability of sites in North America make it impossible to discern any
spatial patterns (Fig. 15b). In Fennoscandia, similar to freezeup, there appears to be variability in warming and
cooling trends (Fig. 15c).



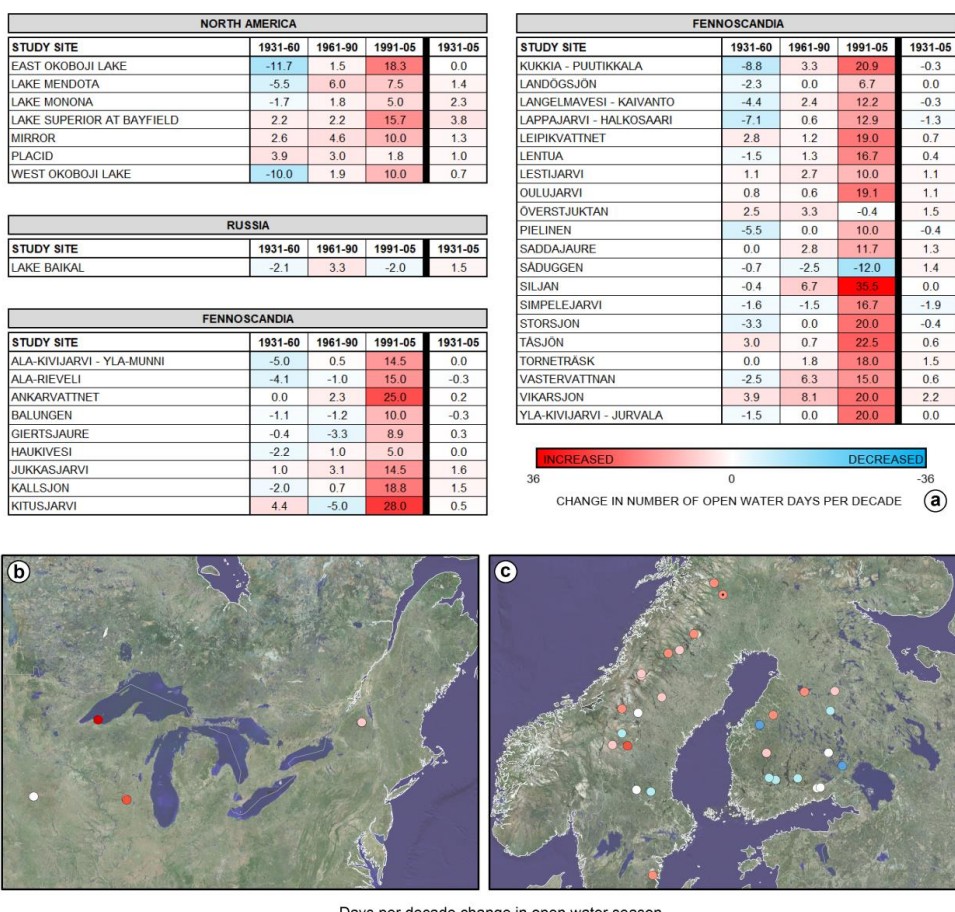


**Figure 15**: Summary of evidence for the long-term trends in the number of open water days each year for the 1931-2005 period: a) heat table demonstrating the decadal change for each site during each time period. The colouring of each cell shows the relative magnitude of that change compared to other sites and time periods; b-c) spatial pattern of the decadal trends for North America and Fennoscandia during 1931-2005. One site is not displayed on the maps, Lake Baikal, which is located on Fig. 7a.

754

Across the full 1931-2005 time period it is clear there is a long-term increase in the number of open water days per year at Lake Baikal in Russia (Fig. 15a). In North America, the number of sites with breakup/freezeup data for 1931-2005 is restricted to only seven sites (Fig. 15b) with a mean value of 1.5 days decade$^{-1}$ more open water days (Table 2). The long-term trends demonstrate a consistent pattern of both earlier breakup and later freezeup, resulting

 

in a lengthening of the open water season (Fig. 5). Only East Okoboji Lake shows a different pattern, where there
is no clear observable trend in the length of the open water season, despite the freezeup date becoming earlier
during this period. This clearly reflects the earlier freezeup trends of 0.3 days decade$^{-1}$ is not large enough to result
in a significant shift in the length of the open water season.

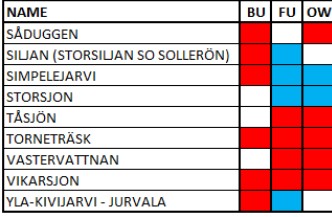


**Figure 16**:

Comparison of how sites in Fennoscandia with an open water season calculation for the 1931-2005 time period.
This reflects the relative changes in dates for breakup and freezeup. The red, white, and blue colours demonstrate
whether the calculated trend reflects a warming trend (earlier breakup, later freezeup, or increased open water
season), a cooling trend (the opposite), or no trend, respectively. Abbreviations are: BU – breakup, FU – freezeup,
and OW open water.

During 1931-2005 in Fennoscandia there is considerable spatial variability with no clear trends, except for a slight
tendency for sites in Sweden to display increased open water season lengths compared to Finland, which shows a
reduction (Fig. 15c). This variability is reflected by the low magnitude of the mean decadal change of 0.4 days
decade$^{-1}$ increase open water season length (Table 2). It is notable that on this longer time scale that none of the
sites show later breakup, with the vast majority showing a warming signal of earlier breakup (Fig. 16). As a
consequence, most of the sites showing a reduction in the length of the open water season do so because trends for
earlier freezeup are larger than those for earlier breakup. This suggests that not only is there a change in the precise
length of the open water season, but there is a shift in when it occurs. This is also the case for several sites, such as
Haukivesi (Fig. 16), that show both earlier breakup and freezeup, so although there is no change in open water
season length, there is change in when it occurs – potentially having implications for biogeochemical cycles.



**4. Causes of ice phenology change**

Figure 17 shows the correlations between breakup/freezeup and a series of climatic variables and indices for each of the three study regions: Fennoscandia (FEN), North America (NAM) and Russia (RUS), on a monthly basis and for three-monthly means over the time period 1931-2005. Unsurprisingly, rising temperatures appear to be the dominant control on the shift towards earlier breakup and later freezeup in the ice phenology records. Late winter and spring temperatures negatively correlate most strongly with breakup, which is expected since rising temperatures lead to more rapid ice melt and thus earlier breakup dates. Autumn and early winter temperatures positively correlate most strongly with freezeup, which is entirely as expected as increasing temperatures lead to delayed freezeup dates. At FEN and NAM, the month preceding breakup (April and March respectively) exhibits the strongest correlation with temperatures, whereas for freezeup the strongest correlation with temperatures occurs on the month of freezeup (November and December, respectively). This may relate to the gradual build-up of rising air temperatures required to break up ice to depth, as opposed to the more rapid onset of freezeup with falling autumn and winter air temperatures.

The three-month temperature means exhibit even stronger correlations with breakup and freezeup, with March-May temperatures and February-April temperatures correlating most strongly with breakup at FEN and NAM respectively, and October-December temperatures correlating most strongly with freezeup at both FEN and NAM. These correlations are physically sensible, with breakup/freezeup occurring towards the end of the three month means. In RUS strongest correlations with breakup occur in February – three months prior to the mean breakup date in early May, which may relate to an increased ice thickness and hence longer time period required to cause breakup. However, when considering the three month temperatures means, the strongest correlations with breakup occurs during February-May – which fits more closely with the mean breakup date. Temperatures during the month preceding freezeup (December) and particularly the three-month mean period October-December correlate most strongly with freezeup dates at RUS. This delayed response to falling winter temperatures at RUS compared to FEN and NAM may relate to the influence of other climatic or site-specific factors, especially since the RUS record applies to just a single lake.





| (a) | | Jan | Feb | Mar | Apr | May | Jun | Jul | Aug | Sep | Oct | Nov | Dec |
|---|---|---|---|---|---|---|---|---|---|---|---|---|---|
| BREAKUP | Temp | -0.31 | -0.48 | -0.50 | -0.77 | -0.45 | -0.17 | -0.09 | -0.08 | -0.20 | -0.08 | -0.12 | -0.26 |
| | | -0.16 | -0.43 | -0.74 | -0.55 | 0.07 | -0.10 | -0.20 | -0.06 | -0.15 | 0.02 | -0.15 | 0.06 |
| | | -0.24 | -0.47 | -0.32 | -0.37 | -0.06 | 0.15 | -0.05 | 0.13 | -0.07 | -0.04 | 0.18 | -0.04 |
| | Prcp | -0.38 | -0.30 | -0.22 | -0.11 | -0.12 | -0.15 | -0.13 | -0.03 | 0.12 | 0.02 | 0.06 | 0.01 |
| | | 0.11 | 0.00 | -0.21 | -0.10 | -0.13 | -0.06 | 0.04 | 0.12 | -0.15 | -0.17 | 0.04 | 0.03 |
| | | 0.04 | 0.08 | -0.01 | 0.10 | -0.05 | 0.00 | -0.01 | -0.13 | 0.21 | 0.00 | 0.00 | 0.09 |
| | Wind | 0.14 | 0.12 | 0.11 | 0.03 | -0.24 | -0.19 | -0.16 | -0.24 | -0.10 | -0.03 | 0.05 | 0.08 |
| | | 0.31 | -0.22 | -0.14 | 0.11 | 0.24 | 0.37 | 0.28 | 0.39 | 0.31 | 0.30 | 0.13 | 0.05 |
| | | | | | | | | | | | | | |
| | NAO | -0.41 | -0.39 | -0.34 | -0.17 | 0.07 | -0.21 | 0.09 | -0.18 | 0.06 | -0.10 | 0.03 | -0.01 |
| | | -0.11 | -0.22 | -0.19 | 0.17 | 0.17 | 0.07 | 0.06 | -0.01 | -0.02 | -0.13 | 0.04 | 0.02 |
| | | -0.29 | -0.30 | -0.25 | -0.02 | 0.15 | -0.12 | -0.01 | -0.16 | 0.23 | 0.06 | 0.12 | 0.08 |
| | AO | -0.31 | -0.41 | -0.41 | -0.29 | -0.16 | -0.15 | -0.03 | -0.04 | -0.16 | 0.03 | 0.03 | -0.18 |
| | | -0.10 | -0.10 | -0.16 | -0.03 | 0.04 | 0.04 | -0.11 | -0.08 | 0.07 | 0.03 | 0.12 | -0.12 |
| | | -0.29 | -0.40 | -0.25 | -0.19 | 0.12 | -0.06 | 0.03 | -0.11 | 0.11 | 0.17 | 0.11 | 0.04 |
| | AMO | -0.07 | -0.06 | -0.09 | -0.02 | -0.06 | -0.10 | -0.05 | -0.12 | -0.10 | -0.06 | 0.02 | -0.12 |
| | | 0.10 | 0.09 | -0.13 | -0.17 | -0.31 | -0.16 | -0.06 | 0.02 | -0.08 | 0.00 | -0.04 | -0.07 |
| | | -0.10 | -0.10 | -0.12 | -0.16 | -0.16 | -0.11 | -0.02 | -0.16 | -0.17 | -0.16 | -0.18 | -0.05 |
| FREEZEUP | Temp | 0.12 | 0.20 | 0.23 | 0.25 | -0.03 | -0.10 | -0.06 | 0.08 | 0.27 | 0.59 | 0.81 | 0.35 |
| | | -0.07 | 0.27 | -0.01 | 0.11 | -0.03 | 0.18 | -0.03 | -0.18 | 0.19 | 0.14 | 0.64 | 0.66 |
| | | 0.24 | 0.08 | 0.13 | -0.09 | -0.02 | -0.06 | -0.02 | 0.03 | 0.00 | -0.07 | 0.32 | 0.49 |
| | Prcp | 0.08 | 0.00 | 0.02 | 0.12 | 0.34 | 0.14 | 0.17 | 0.04 | 0.00 | 0.07 | 0.45 | 0.09 |
| | | 0.12 | 0.07 | -0.11 | -0.05 | 0.00 | 0.11 | -0.05 | -0.02 | 0.06 | -0.02 | 0.15 | 0.06 |
| | | 0.12 | -0.17 | 0.02 | 0.11 | -0.02 | -0.01 | -0.11 | -0.02 | 0.08 | 0.03 | 0.03 | 0.06 |
| | Wind | -0.18 | -0.34 | -0.16 | -0.04 | 0.12 | 0.13 | 0.16 | 0.14 | 0.05 | -0.09 | -0.25 | -0.09 |
| | | -0.14 | -0.07 | -0.02 | -0.16 | -0.14 | -0.29 | -0.11 | -0.07 | -0.28 | -0.19 | -0.20 | -0.27 |
| | | | | | | | | | | | | | |
| | NAO | 0.12 | 0.12 | 0.00 | 0.00 | 0.04 | 0.06 | 0.01 | 0.15 | 0.13 | 0.25 | 0.21 | 0.01 |
| | | -0.09 | 0.08 | 0.04 | -0.08 | -0.07 | -0.08 | -0.14 | -0.10 | -0.12 | -0.04 | 0.26 | 0.23 |
| | | 0.10 | 0.00 | 0.08 | -0.22 | 0.09 | -0.13 | 0.00 | 0.11 | -0.21 | -0.15 | 0.04 | -0.02 |
| | AO | 0.05 | 0.22 | 0.02 | 0.15 | 0.11 | -0.01 | -0.02 | -0.07 | 0.28 | 0.16 | 0.13 | -0.10 |
| | | -0.05 | 0.05 | 0.00 | -0.03 | 0.01 | -0.04 | 0.01 | -0.07 | -0.03 | -0.05 | 0.21 | -0.01 |
| | | 0.07 | 0.00 | -0.12 | -0.31 | 0.03 | -0.03 | -0.01 | 0.02 | -0.23 | -0.15 | 0.35 | 0.16 |
| | AMO | 0.16 | 0.13 | 0.22 | 0.10 | 0.13 | 0.18 | 0.15 | 0.19 | 0.15 | -0.01 | 0.17 | 0.23 |
| | | -0.01 | -0.03 | -0.06 | 0.03 | 0.00 | -0.09 | -0.22 | -0.08 | 0.02 | 0.07 | -0.03 | 0.13 |
| | | -0.21 | -0.09 | -0.06 | 0.03 | 0.01 | 0.01 | -0.03 | -0.07 | -0.03 | 0.02 | -0.12 | -0.18 |

807



| (b) | | JFM | FMA | MAM | AMJ | MJJ | JJA | JAS | ASO | SON | OND | NDJ | DJF | |
|---|---|---|---|---|---|---|---|---|---|---|---|---|---|---|
| | | -0.53 | -0.68 | -0.83 | -0.69 | -0.37 | -0.17 | -0.16 | -0.17 | -0.18 | -0.26 | -0.37 | -0.48 | FEN |
| | Temp | -0.71 | -0.81 | -0.66 | -0.34 | -0.08 | -0.16 | -0.19 | -0.09 | -0.13 | -0.03 | -0.13 | -0.29 | NAM |
| | | -0.46 | -0.53 | -0.37 | -0.18 | 0.03 | 0.12 | 0.01 | 0.01 | 0.08 | 0.05 | -0.05 | -0.37 | RUS |
| | | -0.45 | -0.32 | -0.23 | -0.23 | -0.23 | -0.15 | -0.03 | 0.06 | 0.13 | 0.05 | -0.19 | -0.36 | FEN |
| | Prcp | -0.18 | -0.29 | -0.41 | -0.36 | -0.25 | -0.05 | 0.08 | 0.10 | 0.08 | 0.06 | -0.03 | -0.08 | NAM |
| | | 0.10 | 0.16 | 0.12 | 0.00 | -0.08 | -0.17 | -0.18 | -0.15 | -0.03 | 0.03 | 0.03 | 0.02 | RUS |
| | | 0.13 | 0.15 | -0.07 | -0.22 | -0.20 | -0.20 | -0.19 | -0.20 | -0.07 | 0.01 | 0.09 | 0.12 | FEN |
| | Wind | 0.10 | 0.06 | 0.23 | 0.31 | 0.35 | 0.38 | 0.31 | 0.33 | 0.22 | 0.24 | 0.20 | 0.17 | NAM |
| BREAKUP | | | | | | | | | | | | | | RUS |
| | | -0.59 | -0.53 | -0.30 | -0.18 | -0.04 | -0.19 | -0.03 | -0.13 | 0.00 | -0.04 | -0.24 | -0.50 | FEN |
| | NAO | -0.26 | -0.15 | 0.09 | 0.23 | 0.19 | 0.07 | 0.02 | -0.09 | -0.06 | -0.03 | -0.03 | -0.19 | NAM |
| | | -0.43 | -0.34 | -0.10 | 0.00 | 0.02 | -0.18 | 0.04 | 0.08 | 0.24 | 0.15 | -0.07 | -0.32 | RUS |
| | | -0.51 | -0.53 | -0.47 | -0.36 | -0.19 | -0.12 | -0.13 | -0.08 | -0.13 | -0.18 | -0.19 | -0.19 | FEN |
| | AO | -0.16 | -0.15 | -0.12 | 0.02 | 0.00 | -0.07 | -0.05 | 0.02 | 0.12 | -0.12 | -0.12 | -0.12 | NAM |
| | | -0.43 | -0.41 | -0.22 | -0.10 | 0.05 | -0.09 | 0.01 | 0.10 | 0.20 | 0.04 | 0.04 | 0.03 | RUS |
| | | -0.08 | -0.06 | -0.07 | -0.07 | -0.08 | -0.10 | -0.10 | -0.10 | -0.05 | -0.06 | -0.07 | -0.10 | FEN |
| | AMO | 0.03 | -0.07 | -0.23 | -0.24 | -0.19 | -0.07 | -0.04 | -0.02 | -0.04 | -0.04 | 0.00 | 0.04 | NAM |
| | | -0.12 | -0.14 | -0.16 | -0.16 | -0.10 | -0.11 | -0.12 | -0.17 | -0.19 | -0.14 | -0.13 | -0.10 | RUS |
| | | 0.22 | 0.27 | 0.22 | 0.05 | -0.10 | -0.04 | 0.12 | 0.47 | 0.81 | 0.83 | 0.55 | 0.29 | FEN |
| | Temp | 0.09 | 0.17 | 0.04 | 0.13 | 0.05 | -0.01 | -0.01 | 0.09 | 0.52 | 0.75 | 0.62 | 0.44 | NAM |
| | | 0.19 | 0.07 | 0.03 | -0.08 | -0.05 | -0.02 | 0.01 | -0.03 | 0.20 | 0.42 | 0.53 | 0.40 | RUS |
| | | 0.05 | 0.07 | 0.27 | 0.35 | 0.34 | 0.18 | 0.13 | 0.07 | 0.28 | 0.33 | 0.39 | 0.09 | FEN |
| | Prcp | 0.13 | 0.05 | 0.01 | 0.05 | 0.07 | 0.04 | -0.05 | -0.15 | -0.13 | -0.10 | 0.07 | 0.14 | NAM |
| | | -0.08 | -0.20 | -0.05 | -0.10 | 0.03 | -0.02 | -0.02 | -0.06 | -0.08 | 0.10 | -0.02 | 0.01 | RUS |
| | | -0.24 | -0.29 | -0.06 | 0.09 | 0.14 | 0.15 | 0.14 | 0.08 | -0.17 | -0.14 | -0.18 | -0.21 | FEN |
| | Wind | 0.06 | -0.02 | -0.04 | -0.13 | -0.10 | -0.06 | -0.05 | -0.09 | -0.18 | -0.16 | 0.04 | 0.05 | NAM |
| FREEZEUP | | | | | | | | | | | | | | RUS |
| | | 0.13 | 0.08 | 0.03 | 0.05 | 0.07 | 0.14 | 0.17 | 0.31 | 0.34 | 0.27 | 0.19 | 0.15 | FEN |
| | NAO | 0.02 | 0.03 | -0.07 | -0.13 | -0.18 | -0.19 | -0.20 | -0.16 | 0.07 | 0.27 | 0.21 | 0.12 | NAM |
| | | 0.08 | -0.08 | -0.05 | -0.16 | -0.02 | 0.00 | -0.06 | -0.15 | -0.18 | -0.06 | 0.07 | 0.05 | RUS |
| | | 0.14 | 0.19 | 0.12 | 0.17 | 0.06 | -0.05 | 0.12 | 0.20 | 0.28 | -0.10 | -0.10 | -0.10 | FEN |
| | AO | 0.00 | 0.02 | -0.01 | -0.03 | -0.01 | -0.06 | -0.05 | -0.08 | 0.10 | -0.01 | -0.01 | -0.01 | NAM |
| | | -0.02 | -0.15 | -0.21 | -0.21 | 0.00 | -0.01 | -0.13 | -0.19 | 0.06 | 0.16 | 0.16 | 0.16 | RUS |
| | | 0.18 | 0.16 | 0.16 | 0.16 | 0.17 | 0.19 | 0.18 | 0.12 | 0.12 | 0.14 | 0.22 | 0.20 | FEN |
| | AMO | -0.03 | -0.02 | -0.01 | -0.03 | -0.12 | -0.14 | -0.10 | 0.00 | 0.02 | 0.06 | 0.04 | 0.04 | NAM |
| | | -0.13 | -0.04 | -0.01 | 0.02 | -0.01 | -0.03 | -0.05 | -0.03 | -0.05 | -0.10 | -0.19 | -0.18 | RUS |

**Figure 17**: 'Heatmap' illustrating correlations between breakup/freezeup and a series of climatic variables and indices for each of the three study regions: Fennoscandia (FEN), North America (NAM) and Russia (RUS) on a monthly basis (**a**) and for three-monthly means (**b**) where JFM is the mean of January, February and March etc., over the time period 1931-2005.

Although temperature exhibits the strongest correlations with both breakup and freezeup, precipitation also appears to play an important role in some instances. Increasing winter precipitation (January and particularly the January-March mean) is associated with earlier breakup in FEN, while increasing spring precipitation (March and particularly the March-May mean) appears to exert a stronger influence on earlier breakup in NAM. The latter likely relates to increasing precipitation as rainfall, which aids in the melting of ice (Beltaos and Burrell, 2003). The rising winter precipitation in FEN, presumably as snowfall, may also be associated with earlier breakup since snowfall settling on ice can insulate the ice surface and prevent further thickening during the winter (Park et al., 2016) – therefore potentially promoting earlier breakup. Rising precipitation in November (and to a lesser extent





the November-January mean) is associated with later freezeup in FEN. This may relate to increased discharge into
lakes or rivers, making it harder for surfaces to stabilise and freeze. The correlations between precipitation and
freezeup are weak at both NAM and RUS, while RUS also exhibits weak correlations between precipitation and
breakup. There are also some relatively close associations between wind speed and breakup/freezeup at FEN and
NAM (no wind speed data was available for RUS). Higher wind speeds in summer correlate most strongly with
later breakup and earlier freezeup in NAM. The latter seems counter-intuitive since high wind speeds are generally
thought to disrupt the water surface and delay freezup, while the former does not have any particularly relevant
temporal connection. These correlations are not particularly strong compared to those of temperature with
breakup/freezeup and to a lesser extent precipitation, while they could also simply be a product of chance.
In terms of the atmospheric/oceanic modes of variability, some strong correlations exist with breakup and to a
lesser extent freezeup in all regions. Most notably there are strong negative correlations between breakup and
winter/early spring NAO and AO, i.e. when NAO/AO are in a positive phase, breakup occurs earlier. This is
particularly true in FEN, where a strong positive phase of NAO and AO for the January-March mean and the
February-April mean respectively are associated with earlier breakup. Correlations for RUS at a similar time of
year are also apparent, while correlations in NAM are much weaker. Positive correlations (albeit not as strong)
between freezeup and NAO/AO occur in autumn in FEN and early winter in NAM, i.e. when NAO/AO are in a
positive phase, freezeup occurs later. These findings are expected, since a stronger positive NAO/AO phase results
in an increase in stronger westerly winds, drawing warmer air across northern Europe feeding from the North
Atlantic Drift and Norwegian Current (Hurrell, 1995). A strong positive NAO/AO promotes later freezeup in late
autumn/early winter, and earlier breakup in spring. Trends towards earlier breakup and later freezeup throughout
the latter third of the 20th century may relate to the positive trends of the NAO and the closely associated AO for
much of the 1970s and 1980s, with historical highs in the early 1990s (Cohen and Barlow, 2005). Correlations with
AMO for the full time period are generally not as strong, with the exception of negative correlations between late
spring AMO and breakup in NAM, i.e. when AMO experiences a warm phase, earlier break up occurs. During
warm phases of the AMO, elevated sea surface temperatures in the North Atlantic bring about warmer and drier
conditions across much of North America (Enfield et al., 2001) – hence the association between earlier breakup
with the AMO in this region.



**6. Summary and conclusions**

Utilising a number of different datasets, a series of analyses have been used to investigate how the number of open water days per year and the timing breakup/freezeup dates have changed for water bodies that ephemerally freeze across the Northern Hemisphere. Four time periods (1931-1960, 1961-1990, 1991-2005, and 1931-2005) have been investigated across 644 sites with data in at least one of the time periods to provide ~2600 time series of lake and river ice phenology change to be statistically, spatially, and temporally analysed. A warming signal has been observed that shows the breakup dates for sites with continuous data in the 1931-2005 period have occurred on average 0.6 days per decade earlier across the NH. Freezeup trends for the same time period show greater variation between later and earlier freezeup dates and indicate a more complex response to observed temperature rise. Thus, freezeup trends display a less predictable response to temperature changes when compared to breakup. When the time series are investigated on smaller timescales to explore temporal changes, the breakup trends show a consistent trajectory towards earlier breakup dates that is nonlinear with respect to magnitude – i.e. the magnitude of the shift toward earlier breakup increases through the three periods. The breakup trends display a strong correlation with temperature observations in the weeks preceding breakup and during winter ice growth, suggesting that temperature can be confidently used to predict breakup date. Freezeup trends are generally much more variable through time and display a complex relationship with climate. This is likely because freezeup is not guaranteed to occur simply because temperatures have moved below 0 °C as water kinetics can prevent freezeup. In general, the number of open water days tend to display similar spatiotemporal patterns to those observed for breakup. This shows that even with the inconsistent nature of the changes in freezeup dates, the relative changes between breakup and freezeup dates has led, through time, to a reduction in the length of the ice season and consequently an increase in the number of open water days across the Northern Hemisphere.

Five key conclusions have been drawn from this research; (1) a warming signal is clearly observable in breakup and many sites, (2) this warming signal has accelerated through time for many sites, (3) the causes of the spatiotemporal variability and magnitude of trends is generally well-aligned to temperature trends, (4) freezeup trends are more spatiotemporally complex and display weaker correlations with climate patterns, and (5) the length of the open water season has generally increased through time. The results presented here provide an important contribution that can be used to help understand how ice phenology patterns may change in the future with an expected rise in global mean temperatures. The observed acceleration of warming trends through time for many sites highlights the importance of non-linear responses to climate forcings and will require a greater understanding





of how this will impact not just lake and river hydrology, but also the impact that reduced ice cover will have on
local energy balances and biogeochemical processes. It is possible (if not probable) that changes in lake and river
ice phenology patterns, brought about by warmer air temperatures, may in turn begin to feedback into the climate
system by the release of additional greenhouse gases (e.g. $CH_4$). This highlights the need for a more detailed
understanding of historical changes and their causes to fully unravel the potential implications of ice phenology
change for the projection of future climate changes.

**Data availability**
All of the raw data are available through the National Snow and Ice Data Centre or by contacted the relevant
meteorological institutes.

**Author contribution**
AMWN led the project analysis, writing, and figure preparation with input on all from DM.

**Competing interests**
There are no competing interests to declare.

**Acknowledgements**
We would like to acknowledge the help of Johanna Korhonen and Ville Siiskonen at the Finnish Meteorological
Institute, Torny Axell at the Swedish Meteorological and Hydrological Institute, Ann Windnagel at the National
Snow and Ice Data Centre, and Ånund Kvambekk at the Norwegian Water Resources and Energy Directorate for
their help with building the ice phenology database. Chris Clark is thanked for discussions that were a precursor to
this work.

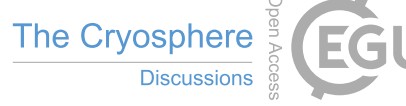

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
