# Peer review of "Climate change and Northern Hemisphere lake and river"

_The Cryosphere, 2020_

## Referee Comment (RC1) · John Magnuson (Referee) · 2 Sep 2020

Comments from reviewer John J. Magnuson Aug 28, 2020.

General comments: The paper provides a detailed analysis of a larger group of lakes and streams over 74 years from 1963 to 2005. Data are from the collection of data in the Snow and Ice Data Center supplemented largely with Swedish, Finnish data. The analysis of slopes were over 4 time periods, 1931-1960, 1961-1990, 1991-2005, and 1931-2005. There are a number of new findings that included comparisons of lakes vs streams, changes in open water duration, differences among regions, and differences among the year subgroupings. I have organized my comments below in terms of what

I liked and what I have concerns about.

What I liked.

Was a Northern Hemisphere analysis rather than a locally constrained analysis. The regional comparisons are useful. Made an honest attempt to include all of the data unless there were too many missing values. Good idea.

Used fixed time periods with less than 10% missing years. Good idea. Other researchers have occasionally had difficulties using these data because they did not constrain the time periods and therefore mixed the influences of longer term changes over time.

Analyzed rivers as well as lakes. Good idea.

Looked at the length of the open water duration. Good idea. Most limnologists have looked at the duration of ice cover over a winter season, rather than the duration of the open water over an entire open water season. Might have been interesting to think about how that changes one's perspective. Both ice cover and open water seasons play a role in the biogeochemical cycling and other ecological phenomena as related to ice cover. Is the change in one more important ecologically than the change in the other? Might have been useful to show relation over time of both ice cover and open water. Even though they relate to two sets of years, they should be highly correlated

Consideration of heterogeneity among lakes, time periods, and regions as well differences in variability. Good idea. I think they might have considered making the comparisons in regard to a set of issues, questions, or major or in respect from what they know from the ice cover literature. "Looking in some detail at regional differences" was a good idea such as northern and southern Sweden, or Europe and North America. But I am not convinced that I agree with their explanations.

I liked the quick reference list with findings from each published paper they found. Good idea. However, the results from these papers were not integrated or their findings or

compared with this paper's findings. Findings stated as new were already found in the various published analyses.

What concerned me.

1. Did not indicate what was actually a new finding. It was all new, perhaps because it included all of the usable data between 1963 and 2005. But items in their final list of findings have often been reported as general findings by earlier researchers using various time periods or regions. I would have appreciated their integrating what they found with earlier findings. Such interpretations were usually absent except for a few general statements in the introduction that was an incomplete citation history. Many of the papers were in the list of papers and major results, but they were not integrated into a discussion of their own findings. How do your results compare with Benson et al 2012 using a smaller number of lakes but with longer time series; she looked at 150 year trends, 100 year trends and 30 year trends using largely the same data source that the authors used. So, "what was new", was that they used the most robust data set that existed on inland-water ice phenology. I thought the most defensible grouping of years was of the full duration of years for the waters they analyzed. The least defensible was the short time series in the most recent years. Perhaps I missed it, but I would have enjoyed reading more about the rationales for the year subgroupings they decided to analyze. Why not use three equal year groupings of the same length. How does the number of years in a set influence the results?

2. Long term. They referred to the time series analyzed as long term, but were they long enough? Unfortunately, they did not cite an important paper by R. Wynne (2000) that first revealed that running slopes of the lake ice time series alternated from positive and negative when analyzed using consecutive moving windows. Wynne looked at 4 lakes with 100 year time series of ice breakup (2 in North America and 2 in Europe). The running means both of 20 and 50 years across the time series oscillated over the 100 years between positive and negative slopes. Whether one observed a positive or a negative slope depended on the start date of the subset. See (Wynne, R. H. 2000.

Statistical modeling of lake ice phenology: Issues and implications. Verhandlungen des Internationalen Verein Limnologie 27:2820–282).

3. Oscillatory dynamics. Papers on the oscillatory dynamics of longer term data reveal a number of interacting oscillations. (e.g. Sharma, Sapna, and John J. Magnuson. 2014. Oscillatory dynamics do not mask linear trends in the timing of ice breakup for Northern Hemisphere lakes from 1855 to 2004. Climatic Change 124:835-847.) Again, the slope of subsets selected would depend on the start date of the subset. Take a look at Sapna's Figure 5 panel D and consider the result of having started a 30-year period in 1948 versus 1961 or 1960. Some of the most significant oscillations in the Sharma paper were in the range of El Nino that could easily mess up the interpretations of the shortest time period the authors used for recent years. Their most recent date group was short enough that an El Nino near the end or the beginning of that short series could be relevant. Interpreting it as a more general long-term change is problematic. Many published papers have analyzed these oscillations and the authors should have at least discussed the issue and how that might have influenced their conclusions. They cited most of them in their listing. I think it would have been interesting to do an analysis like Sapna's over the 74 years for different regions. A simple set of running means as in her figure 5 might be sufficient. Then compare those rather than for the three periods the author's used (two of equal length and one short).

4. A point on the large-scale climate drivers like NAO or AO. The Livingstone (2000) paper they originally credited for the roll of these kinds of drivers on ice phenology found relations not only with NAO but also with the SOI. Lake Mendota, for example, was influence by both. I think the authors should have included SOI in their analyses. (Livingstone, D.M., 2000. Large-scale climatic forcing detected in historical observations of lake ice breakup. Verhandlungen des Internationalen Verein Limnologie 27:2775–2783.). Other papers the author's cited also found that SOI was also an important correlate.

5. What other papers pointed out that ice cover was related to air temperature and

the climatic variables. A discussion that included how these results compare with the author's findings would have been interesting to me.

6. The comparisons they made with lakes and streams were new and I liked that. The only other paper to have done that was the Magnuson et al. 2000 paper in Science. Did Magnuson et al. see a difference between lakes and streams? If not, this is a new finding. I thought that the authors causal explanation for the difference between lakes and flowing waters could be expanded as it did not include some of the most important processes. I think additional factors such as the greater depths of most lakes and the thermal stratification of lakes both should play a role. Rivers tend to mix over the entire water column, only shallow lakes do.

7. I would like to see more explanations of the results based on the authors findings and the literature on the inland water cryosphere. A more in depth explanation of why freeze dates are less directly related of air temperature than are breakup dates, for example. There are brief interpretations of that observation in many of the papers they cite. The literature on the various mechanisms generally are not cited but are known.

8. The paper was a bit exhausting owing to the detail and number of comparisons. It would have been good either to identify major issues rather than describing all of the detailed results or to focus the structure on the regional comparisons. This ended up being a description of what was happening everywhere in great detail. To some extent this is basically a detailed descriptive data report.

9. The authors frequently tried to explain differences among individual lakes rather superficially, eg. Mystery vs Nebish, or Mendota vs Monona. May not have the local knowledge to work at that fine individual lake scale. I would put the analyses into the broader result rather pick apart individual lakes. particular lake

Other suggestions: 1. I would give the years of observation in the title, i.e. 1931-2005.

Minor points.

In the maps take a look at whether the shades of grey are distinguishable from each other or the background. I had difficulties doing so.

Madeline Island and Bayfield data are essentially identical and rightly so. If it is the data set with which I am familiar it comes from the periods of ice cover that excludes boats and ferries from entering or leaving Bayfield. The ferry runs from Bayfield to Madeline island in open water seasons and an ice road is used in winter between the two locations. The early paper on these data is: (Howk, F. Changes in Lake Superior ice cover at Bayfield, Wisconsin. J. Great Lakes Res 35, 159–162 (2009)). There is an additional Chequamegon data set farther up the Bay.
* * *

---

## Referee Comment (RC2) · Anonymous Referee #2 · 19 Oct 2020

General comments: The authors have collected northern hemisphere ice phenology data from several large databases and climate data from an online portal. They have split the ice phenology data into three large geographic regions: North America, Fenoscandia, and Russia and within each region they have examined the overall ice phenology trends and also site (lake or river) specific ice phenology trends over four time peirods. Finally, the authors have used a correlation analysis approach to examine potential relationships between several climate drivers (air temperature, precipitation, wind speed) and ice phenology metrics for the three large regions. The manuscript contains at least two novel components: its geographic scope is large and is split into three reasonably large regions (North America, Fenoscandia, Russia), and the analysis con-

tains an assessment of potential changes in ice-phenology for both lakes and rivers. The general statistical approach of using Mann Kendall trend tests with Sen's slopes to evaluate both the significance and slopes of monotonic trends has been commonly applied to similar datasets. The use of correlation analysis to assess potential drivers is also common, but there is a strong potential for false positives and negatives due to the large number of tests. Further, the authors appear to have used mean climate conditions to within each study region for the correlation analyses. However, given the large size of the study regions and large inter-regional variation of trends (e.g. to the east and west of the Laurentian Great Lakes) such an approach may not be appropriate to assess the linkages between ice phenology trends and climate drivers. There is also some uncertainty of how the regional averages were calculated and if these changed over time as the number and location of sites within the regions changed. For example, there were no data included for Canada during the last time period (only US sites)....did the area used to calculate the 'mean' climatic conditions change? The use of the three time periods is helpful but the last time period was unequal in terms of length (the first two periods were of 30 year durations, while the last was only 15 years). I suspect the choice of years may have been logistical, but the authors should evaluate and discuss the implications of this decision. Third, as part of the online comments, a reviewer suggested the authors evaluate the implications of oscillatory dynamics that may affect the slopes of their relationships, particularly in the shorter 30 or 15 year time periods. This is quite important and the authors should consider this. In terms of methods, the authors included too much detail in some areas (i.e. mathematics supporting Mann Kendall are not needed here), but not enough information in others (see specific comments). While I found the manuscript and analysis interesting and useful, my largest criticisms relate to its length, lack of clear objectives and hypotheses, attempts to assess changes at both large regional and site specific scales, and limited attempts to compare their results with other large scale analyses in the literature. At 50 pages, 17 figures and 7 tables, the manuscript is far too long and unfocussed. I suggest the authors remove all site-specific analyses and focus the manuscript on the

broader regional trends and drivers and novel aspects (e.g. lakes vs. rivers). If the large regions require some additional partitioning due to interregional variability that is fine, but the analysis and discussion of individual sites is not particularly useful for the reader and takes away from the more important regional assessment. From there, the authors should more clearly state their objectives and hypotheses, and focus the discussion on not only their results, but also improve the discussion of the implications of their findings and how they compare and contrast with other large-scale analyses (i.e. what is new and novel). The authors would also need to defend their correlation approach to identifying potential drivers, acknowledging the high likelihood of false positives and negatives, potential challenges wit using regionally averaged values over such large geographic areas where there is both high intra-region variability and where the geographic distribution of sites change, and the use of climate indices over large geographic areas. I have provided some more detailed comments below.

Specific points   The introduction could use a clear set of objectives and hypotheses. This would greatly assist with improving the conciseness of the manuscript, which is quite long and unfocussed.   P9 L133+ The authors use climate variables (mean monthly temperature, precipitation) from KNMI Climate Explorer. The data were 'downscaled' (averaged?) over 3 geographic regions (Fennoscandia, North America, Russia) and then used to assess relationships with spatially averaged ice-freeze up or break up dates. Is spatially averaging over such a large geographic area a good idea? There is evidence that there are large differences in the magnitude and even direction of ice phenology trends within the North American dataset. Would the strength of the analysis not be affected by geographic scale with weaker correlations expected as the study area increased?   Should section 3 (L. 153) not read '3. Results'?   With 60 pages of text (including 17 figs and 9 tables), this manuscript is quite long and highly unlikely to fit into the space requirements of the journal. The manuscript is also largely unfocussed, moving from broad discussions of phenological changes among regions to significant text devoted to individual sites. Much of the site-specific material is extraneous and distracts from the broad patterns. The objectives and hypotheses

of the study are not clearly articulated and the closest we get is on page 3 where the authors state 'The paper explores the hemispheric spatiotemporal trends in ice phenology....Observed changes are then compared with climate records...' I would encourage the authors to devise more clear objectives and testable hypotheses. Further, I would encourage the authors to focus the current manuscript on a concise description of the broad phenological changes among the regions without delving into specific site responses. These are unnecessarily distracting. Such individual sites responses could either go into supplemental material or separate manuscripts. • In relation to methods, the Mann Kendall trend tests with Sen's slopes is an appropriate statistical test for the monotonic trends, however the methods section does not indicate at what level the trends were considered significant. The correlation analysis, examining the potential drivers, is more suspect for a few reasons. First, there are a large number of correlations being tested (>800) and therefore a large potential for false positives or negatives. Second, at least two of the regions (NAM and RUS) have large geographic extents and the authors acknowledge in other locations in the manuscript that there are large regional differences in phenological trends (e.g. east and west of the Laurentian Great Lakes). Thus the use of a single averaged climate (temperature, precipitation, wind, etc.) value for a study region would not be expected to correlate well with the phenological trends. While considerably more work, it would be more useful to examine the relationships between downscaled climate drivers and phenological responses at individual sites (or smaller regions). This would also avoid the changing locations of sites between study periods. • While I greatly appreciate the extent of analysis undertaken, the presentation of results is at times confusing and potentially misleading. There are many figures (e.g. Figure 5, 7, 8, 11, 13, 14, 15; Tables 2, 3 [first panel] ) where trends are reported. However, the reader is unable to distinguish if these individual site trends are significant and, if so, at what level? If they are not significant, then they should not be reported as so. The legend should also indicate what the white boxes represent (e.g. insufficient data, no trend, something else?). • Table 1. Assel et al. 2003. What is meant by 'maximum fraction of lake surface

ice coverage'? Should this be maximum extent of lake ice coverage'? • Table 1, while useful, is quite large and is probably best suited to supplemental material. It also misses many references (Magnuson, Benson, Sharma, etc. etc.). • Surprised Benson et al. 2013 was not included in Table 1 given it covers such a large number of lakes • P6 L86. The indication of rivers in brackets. Are these 88 sites on three rivers, or 85 lake sites and 3 river sites? • P6 L88 Change to are clustered around the Laurentian Great Lakes. • P7 L99 Change Julian days to Ordinal days? • Note: Mann Kendall examines unimodal trends only • P8 L103+ More information is required on the statistical approach. At what level were trends considered significant? The first I see of this is on page 16 L180 where an alpha value of 0.1 was considered significant. • P8 L108+ The Mann Kendall test is widely used for time series analysis and the mathematical details are widely available. No need to detail here. • Figure 2. The 3 shades of grey are not easily distinguishable. • P10. L154-166. This section reads more like introductory material and methods than results. • P10. Lake size or elevation can have large effects on freeze up or break up dates. Perhaps this (e.g. few lakes, highly variable lake size) might account for the lack of latitudinal trend. But note that in the description they indicated a fairly tight geographical extent in Russia (51.5-52 N, 104.5-105E). • P10. Comments on changes in European changes (L174+) are qualitative... not assessed using a statistical approach. These are not easily distinguishable in the figure. • P10. L171-172. With the exception of 3 sites on the eastern portion of the map, the sites in Russia (Figure 1a) appear largely to be of similar latitude, which precludes a latitudinal assessment. Thus, the case that 'This is the case for ...but not Russia' is not technically correct. Rather, a thorough assessment cannot be made due to limited data. • Figure 2. Why are the grids at different scales between the 3 figure panels? • Figure 3. The figure itself is good, but the caption requires editing. E.g. 'Trends in ice breakup, ice freeze-up (is 'freezeup' a word?), and the duration of the open water period for lakes across North America, Europe (is this not referred to in other locations such as Figure 1 as Fennoscandian? Be consistent), and Russia. • P12. L196. 'When all sites are considered there is a clear increase...'

I don't believe a statistical test was used to make this assertion. While it may appear obvious by the graphic, such visual assessments require a statistical assessment. • P12. L201. 'For Russian sites it is clear...' Again, there is no statistical test on which to base these claims. These are qualitative assessments only. • P13 L203. Again a reference to the 'large area' of the Russian sites. • P15 L60. Note there are many lakes in Canada being monitored for ice-phenology, they are just not recorded in the database or captured in this manuscript. • What was the criteria for whether a site could be included or not. Was it 90% of records for the 15, 30, or 75 year periods? For the 75 year trend test, were only lakes that had 90% of records included for the whole period included? • P14. Table 2. Please include some additional details in the table caption: The statistical test used, the significance value of the trend. Within the table n values and confidence intervals would be helpful. There is value in the table, but it is hampered by the large differences in study areas (e.g. latitudinal extent) and differences in the number of sites between time periods (esp. in Russia and North America). This can influence the qualitative assessments provided in the text, for example in North America between the middle and last time periods where a large number of northern sites dropped off. Further, it would be useful for Figure 1 (panels b-c) could have lake and river sites distinguished independently. This would help to see if lake and river sites were distributed homogenously or if they are clustered at certain latitudes. • P15 L261. Awkward sentence. Reword. • P16. L212 'The data show...'. Larger than what? Where is the statistical test? This is also true for the general pattern discussed for Europe and North America. The wording 'do not appear to change significantly...', was this an eyeball assessment? • P16. L279. While there is some value pointing out that there are exceptions to the general trend, this is obvious in Figure 3. I think this figure can be referred to rather than describing specific lake or river systems that behave differently. • P 16. The use of standard deviation to assess interannual variability should be introduced in the methods. Why use numbers (14) and letters (eight). • P17. L301. The reporting of mean values should be accompanied by estimates of variance, the degrees of freedom, and the significance of

the trends. • P17. L303+. 'Only four North American sites experience later breakup during...' The challenge here is that I can't tell if the authors are assessing significant or significant + non-significant trends. Was the example of Frame Lake a significant or non-significant change? The slope is reported (which appears quite close to zero), but no p value or measures of variance. • P17. L309+ Given the large scope of the manuscript, it seems ill advised to drill down to the extent here for individual lakes or rivers. A blow out box may be more appropriate for dealing with specific case studies if there is a strong need to provide an example. The removal of these details will make it much easier for the reader to follow the broader relevance of the study and more space for the authors to discuss the relevance of their analysis compared to similar studies. • P49. L826+. I'm not sure I follow the logic here. How do summer wind speeds alter ice break up in the spring (either the preceding or following spring)? This is likely a false positive and one would expect some false positives and negatives given the large number of tests. Second, the link between summer windspeed and effects on turbulence are only relevant once air temperatures drop below 0 deg C. Summer windspeeds could be important by deepening thermoclines and higher heat storage, leading to delayed ice out dates. Again, potential for false positive here. Further, I am not surprised by the poor relationships over North America and Russia given the large geographic area and changes in the distribution of sites. As noted by the authors in other locations (e.g. p18 L325), there can be large regional differences in climatic patterns within the study area that can lead to regional differences in ice phenology trends (e.g. NAM sites). While more labour intensive a better approach would be to examine the linkages between climate and ice phenology at the individual sites, and then bringing these relationships into a more global analysis (i.e. does ice phenology respond similarly to climate across sites with regional differences in climate causing regional differences in ice phenology, or are other factors (e.g. lake or watershed morphometry, lake vs. river, etc. critical). • Figure 4. Caption should indicate that trends were tested using Mann Kendall and Sens slopes. After looking at Figure 3 from Duguay et al. 2006 (which I think is largely the same data), most of the ice freeze up values

Interactive
comment

(panel e in this manuscript) would appear to be non-significant. This makes me wonder whether all trends are reported regardless of significance. The authors need to clearly state in the figure caption whether the trends being reported are significant and what level of significance. • Table 3 is confusing. The authors must indicate what statistics were used (Mann Kendall with Sen's slopes?). What do the numbers represent in the panels represent (Sen's slopes?). What are their units (days/decade?). Why are significance values included only in the last three panels/time periods but not the first panel? Does the dash represent 'no data available'? Why not simply use the first panel data? Note, that another interpretation of the data from this table is that few lakes or rivers are displaying significant trends. For example in the 1931-90 period only 4/15 sites show significant changes at the p=0.05 level, or 7/15 at the p=0.10 level. Overall the table is interesting, but these site specific tables should probably be moved to supplemental material. • Table 3 shows that from the period 1991-2005 the majority of N America Lakes included in the analysis had either no significant trend, or had later ice on dates (significant positive numbers). There were no significant negative numbers. Yet, when I examine Figure 4, I see a large number of earlier freeze up dates reported (particularly to the east of the Laurentian Great Lakes). Am I missing something? • Figure 5 caption is poorly written. What does 'Comparison of how sites in North America with an open water season calculation' mean? Perhaps 'Patterns of ice freeze-up and ice breakup....' What level of significance? What statistical test? Were non-significant trends reported as trends? Do white boxes represent no significant trend or no data? • Figure 17. This figure is important as it represents the only example of where the drivers of the phenological trends were being tested. Although the methods are not entirely clear, the analysis seems to have a few issues. I am assuming that the authors assessed the trends of the 'downscaled' annual mean or median values of the freeze-up breakup dates over the full study period (1931-2005) with each study region (FEN, NAM, RUS). How was the air temperature data treated? Was it the mean value of the whole study area? This presents some problems because in both North America and Russia, the study area is quite large and the number and
geography of sites changed dramatically over the study period. For example, no Canadian sites were available during the most recent period. In this case, I would think that this would have a large effect on the relationship between air temperature and ice phenology. For Russia, there was a wide spread of sites (25-27 sites) longitudinally during the 1961-1990 period, but much fewer sites during the periods before (5-6 sites) and after (1 site). A better approach would be to examine the downscaled air temperature values for each site specifically. This would also remove potential problems associated with elevation. Further, a large number of correlations are being tested (e.g. >400 in panel b), which leads to the potential for false positives. A minor point, but presumably the bolding pattern refers to the level of significance of the trends. This would be important to point out in the caption. Perhaps I have misunderstood a few aspects of these analyses and in general more clarification of the statistical approach is needed.

---

## Author Comment (AC1) · 30 Nov 2020

We are grateful to both reviewers for the time they have taken to offer such detailed and insightful comments. The comments from both reviewers are broadly well-aligned and we are happy to make the suggested changes as they will improve the manuscript. We are confident that these changes are achievable and upon making them, the manuscript will be significantly stronger as a result. The proposed changes are outlined in detail on the attached supplementary pdf, but the main changes will include:

- Condensing the text by refocusing the narrative on the larger scale and removing the finer detailed analyses.

[Figure]

- Improving the write-up of the methods, as well as including extra analyses on the SOI, such that the workflow and results are as clear as possible.

- Modifying the time spans used by adding a new one (1946-1975) and modifying an existing one (1991-2005 changed to 1976-2005). This will allow for a greater comparability between the different time periods and remove any bias caused with the implementation of the shorter time period in the original manuscript.

- Refining figures and text to ensure that the discussion of the statistics is as clear as possible.

- In general the text will also be modified to make the objectives clearer at the start of the paper. We will also modify the discussion to take account of specific points that have been raised by the reviewers and this will be improved upon by integrating the results with the published literature more effectively.

Please also note the supplement to this comment:
https://tc.copernicus.org/preprints/tc-2020-172/tc-2020-172-AC1-supplement.pdf

**Supplement:**

We are grateful to both reviewers for the time they have taken to offer such detailed and insightful comments. The comments from both reviewers are broadly well-aligned and we are happy to make the suggested changes as they will improve the manuscript. We are confident that these changes are achievable and upon making them, the manuscript will be significantly stronger as a result. The proposed changes are outlined in detail below, but the main changes will include:

- Condensing the text by refocusing the narrative on the larger scale and removing the finer detailed analyses.

- Improving the write-up of the methods, as well as including extra analyses on the SOI, such that the workflow and results are as clear as possible.

- Modifying the time spans used by adding a new one (1946-1975) and modifying an existing one (1991-2005 changed to 1976-2005). This will allow for a greater comparability between the different time periods and remove any bias caused with the implementation of the shorter time period in the original manuscript.

- Refining figures and text to ensure that the discussion of the statistics is as clear as possible.

- In general the text will also be modified to make the objectives clearer at the start of the paper. We will also modify the discussion to take account of specific points that have been raised by the reviewers and this will be improved upon by integrating the results with the published literature more effectively.

In the blue text below are the responses to the individual referee comments.

**Comments from John Magnuson**

General comments: The paper provides a detailed analysis of a larger group of lakes and streams over 74 years from 1963 to 2005. Data are from the collection of data in the Snow and Ice Data Center supplemented largely with Swedish, Finnish data. The analysis of slopes were over 4 time periods, 1931-1960, 1961-1990, 1991-2005, and 1931-2005. There are a number of new findings that included comparisons of lakes vs streams, changes in open water duration, differences among regions, and differences among the year subgroupings. I have organized my comments below in terms of what I liked and what I have concerns about.

What I liked.

Was a Northern Hemisphere analysis rather than a locally constrained analysis. The regional comparisons are useful. Made an honest attempt to include all of the data unless there were too many missing values. Good idea.

Used fixed time periods with less than 10% missing years. Good idea. Other researchers have occasionally had difficulties using these data because they did not constrain the time periods and therefore mixed the influences of longer term changes over time.

Analyzed rivers as well as lakes. Good idea.

Looked at the length of the open water duration. Good idea. Most limnologists have looked at the duration of ice cover over a winter season, rather than the duration of the open water over an entire open water season. Might have been interesting to think about how that changes one's perspective. Both ice cover and open water seasons play a role in the biogeochemical cycling and other ecological phenomena as related to ice cover. Is the change in one more important ecologically than the change in the other? Might have been useful to show relation over time of both ice cover and open water. Even though they relate to two sets of years, they should be highly correlated

> Interesting point. We are happy to investigate this further and consider in the discussion section any implications that arise from this.

Consideration of heterogeneity among lakes, time periods, and regions as well differences in variability. Good idea. I think they might have considered making the comparisons in regard to a set of issues, questions, or major or in respect from what they know from the ice cover literature. "Looking in some detail at regional differences" was a good idea such as northern and southern Sweden, or Europe and North America. But I am not convinced that I agree with their explanations.

> We will look in detail again at our explanations and consider how they can be improved, or any caveats more clearly mentioned.

I liked the quick reference list with findings from each published paper they found. Good idea. However, the results from these papers were not integrated or their findings or compared with this paper's findings. Findings stated as new were already found in the various published analyses.

> We will explore in the revision how we can condense the text and provide a better discussion of how this work fits with the papers we have cited.

What concerned me.

1. Did not indicate what was actually a new finding. It was all new, perhaps because it included all of the usable data between 1963 and 2005. But items in their final list of findings have often been reported as general findings by earlier researchers using various time periods or regions. I would have appreciated their integrating what they found with earlier findings. Such interpretations were usually absent except for a few general statements in the introduction that was an incomplete citation history. Many of the papers were in the list of papers and major results, but they were not integrated into a discussion of their own findings. How do your results compare with Benson et al 2012 using a smaller number of lakes but with longer time series; she looked at 150 year trends, 100 year trends and 30 year trends using largely the same data source that the authors used. So, "what was new", was that they used the most robust data set that existed on inland-water ice phenology. I thought the most defensible grouping of years was of the full duration of years for the waters they analyzed. The least defensible was the short time series in the most recent years. Perhaps I missed it, but I would have enjoyed reading more about the rationales for the year subgroupings they decided to analyze. Why not use three equal year groupings of the same length. How does the number of years in a set influence the results?

Fair points. In the revision we will make a better effort to present the main findings and more clearly integrate them with what has already been published. We understand the point on the timescale groups. We had originally presented it this way as something similar had been used in one of the papers we cited (Šarauskienė and Jurgelėnaitė, 2008) and we felt it was effective given the distribution of the data availability. There was a trade-off between including the maximum amount of data and we are happy to address these issues as we think they will ultimately make an important improvement to the paper. The groupings we used allowed us to look at the decadal change across three different periods of time and employ the maximum amount of data as possible. The second reviewer has also raised concerns with these timespans. We propose in the revision to explore how the timescales can be improved upon by perhaps adding a 1946-1975 time period and replacing the 1991-2005 time span with a 1976-2005 time period. This would require extra processing of the data (which is already ongoing as part of a follow-up piece of work – so it would not require too much additional work) and we think this would go a long way to rectifying the issues that have been highlighted. In the revision the results/discussion will be reworked to focus on: 1) the spatio-temporal variability between different time periods, and 2) how sites with continuous data appear to vary between them. We hope this suggested correction will remove the concerns raised.

2. Long term. They referred to the time series analyzed as long term, but were they long enough? Unfortunately, they did not cite an important paper by R. Wynne (2000) that first revealed that running slopes of the lake ice time series alternated from positive and negative when analyzed using consecutive moving windows. Wynne looked at 4 lakes with 100 year time series of ice breakup (2 in North America and 2 in Europe). The running means both of 20 and 50 years across the time series oscillated over the 100 years between positive and negative slopes. Whether one observed a positive or a negative slope depended on the start date of the subset. See (Wynne, R. H. 2000. Statistical modeling of lake ice phenology: Issues and implications. Verhandlungen des Internationalen Verein Limnologie 27:2820–282).

It was an oversight not to cite the paper or include it as a part of the discussion. We touched upon some of the main observations of switching trends indirectly when looking at the 1931-2005 time period, but we agree this needs to be more explicitly discussed and improved upon. The observations from the Wynne paper are something we are intending to pursue in the follow up to this current paper as it requires a considerable amount of statistical analyses. In the revision we will make an effort to explore the implications of this work and compare it with ours. We agree this is important.

3. Oscillatory dynamics. Papers on the oscillatory dynamics of longer term data reveal a number of interacting oscillations. (e.g. Sharma, Sapna, and John J. Magnuson. 2014. Oscillatory dynamics do not mask linear trends in the timing of ice breakup for Northern Hemisphere lakes from 1855 to 2004. Climatic Change 124:835-847.) Again, the slope of subsets selected would depend on the start date of the subset. Take a look at Sapna's Figure 5 panel D and consider the result of having started a 30-year period in 1948 versus 1961 or 1960. Some of the most significant oscillations in the Sharma paper were in the range of El Nino that could easily mess up the interpretations of the shortest time period the authors used for recent years. Their most recent date group was short enough that an El Nino near the end or the beginning of that short series could be relevant. Interpreting it as a more general longterm change is problematic. Many published papers have analyzed these oscillations and the authors should have at least discussed the issue and how that might have influenced their conclusions. They cited most of them in their listing. I think it would have been interesting to do an analysis like Sapna's over the 74 years for different regions. A simple set of running means as in her figure 5 might be sufficient. Then compare those rather than for the three periods the author's used (two of equal length and one short).

> These are good points that have been raised. The suggested edit to the timespans above will help to rectify some of the comments and we are happy to investigate how the work from Sharma et al. (2014) can be used to improve both our results and the discussion around them. In the revision we will improve more the discussion of this, as has rightly been suggested.

4. A point on the large-scale climate drivers like NAO or AO. The Livingstone (2000) paper they originally credited for the roll of these kinds of drivers on ice phenology found relations not only with NAO but also with the SOI. Lake Mendota, for example, was influence by both. I think the authors should have included SOI in their analyses. (Livingstone, D.M., 2000. Large-scale climatic forcing detected in historical observations of lake ice breakup. Verhandlungen des Internationalen Verein Limnologie 27:2775– 2783.). Other papers the author's cited also found that SOI was also an important correlate.

> Agreed, we will add the SOI into our analyses.

5. What other papers pointed out that ice cover was related to air temperature and the climatic variables. A discussion that included how these results compare with the author's findings would have been interesting to me.

> As part of the revamped discussion section we propose to more explicitly discuss this issue.

6. The comparisons they made with lakes and streams were new and I liked that. The only other paper to have done that was the Magnuson et al. 2000 paper in Science. Did Magnuson et al. see a difference between lakes and streams? If not, this is a new finding. I thought that the authors causal explanation for the difference between lakes and flowing waters could be expanded as it did not include some of the most important processes. I think additional factors such as the greater depths of most lakes and the thermal stratification of lakes both should play a role. Rivers tend to mix over the entire water column, only shallow lakes do.

> This is an important point and something we were keen to develop in follow-up work. However, we see the benefits here of a better discussion around these topics and will include this in the revised manuscript.

7. I would like to see more explanations of the results based on the authors findings and the literature on the inland water cryosphere. A more in depth explanation of why freeze dates are less directly related of air temperature than are breakup dates, for example. There are brief interpretations of that observation in many of the papers they cite. The literature on the various mechanisms generally are not cited but are known.

> In our edits of the discussion section we will make these discussion points much clearly and elaborate as suggested.

8. The paper was a bit exhausting owing to the detail and number of comparisons. It would have been good either to identify major issues rather than describing all of the detailed results or to focus the structure on the regional comparisons. This ended up being a description of what was happening everywhere in great detail. To some extent this is basically a detailed descriptive data report.

9. The authors frequently tried to explain differences among individual lakes rather superficially, eg. Mystery vs Nebish, or Mendota vs Monona. May not have the local knowledge to work at that fine individual lake scale. I would put the analyses into the broader result rather pick apart individual lakes.

> We consider these two points together and they are both fair. The authors were unsure about whether reviewers would request that detail or whether they would prefer the focus be on the larger-scale – there are merits on both sides of this and we figured it would be easier in any revision to broaden the scale of the analyses rather than the other way around. However, given the comments raised by both reviewers, and the format of publication, we are very happy to refocus and condense the text by concentrating on the major issues and instead leaving some of the finer details for either citations or follow-up work.

Other suggestions: 1. I would give the years of observation in the title, i.e. 1931-2005.

> Good idea.

Minor points.

In the maps take a look at whether the shades of grey are distinguishable from each other or the background. I had difficulties doing so.

> We will make these changes.

Madeline Island and Bayfield data are essentially identical and rightly so. If it is the data set with which I am familiar it comes from the periods of ice cover that excludes boats and ferries from entering or leaving Bayfield. The ferry runs from Bayfield to Madeline island in open water seasons and an ice road is used in winter between the two locations. The early paper on these data is: (Howk, F. Changes in Lake Superior ice cover at Bayfield, Wisconsin. J. Great Lakes Res 35, 159–162 (2009)). There is an additional Chequamegon data set farther up the Bay.

> We will refine the text around these points to make sure this is clear.

**Comments from Anonymous Referee #2**

General comments: The authors have collected northern hemisphere ice phenology data from several large databases and climate data from an online portal. They have split the ice phenology data into three large geographic regions: North America, Fenoscandia, and Russia and within each region they have examined the overall ice phenology trends and also site (lake or river) specific ice phenology trends over four time peirods. Finally, the authors have used a correlation analysis approach to examine potential relationships between several climate drivers (air temperature, precipitation, wind speed) and ice phenology metrics for the three large regions. The manuscript contains at least two novel components: its geographic scope is large and is split into three reasonably large regions (North America, Fenoscandia, Russia), and the analysis contains an assessment of potential changes in ice-phenology for both lakes and rivers. The general statistical approach of using Mann Kendall trend tests with Sen's slopes to evaluate both the significance and slopes of monotonic trends has been commonly applied to similar datasets. The use of correlation analysis to assess potential drivers is also common, but there is a strong potential for false positives and negatives due to the large number of tests.

> We will discuss these issues in the manuscript more clearly to outline any caveats. More detailed comments are below on the individual line comments.

Further, the authors appear to have used mean climate conditions to within each study region for the correlation analyses. However, given the large size of the study regions and large inter-regional variation of trends (e.g. to the east and west of the Laurentian Great Lakes) such an approach may not be appropriate to assess the linkages between ice phenology trends and climate drivers. There is also some uncertainty of how the regional averages were calculated and if these changed over time as the number and location of sites within the regions changed. For example, there were no data included for Canada during the last time period (only US sites)....did the area used to calculate the 'mean' climatic conditions change?

> The considerable workload involved with correlating each ice phenology record within its own local climate meant we instead assessed the regional trends. The fact that we pick up some strong correlations between regional climate variables/indices and ice phenology means perhaps that our method – while more spatially limited – nonetheless adds value in establishing some of the key drivers for ice phenology changes. We will discuss the value and limitations of our approach in the manuscript. We will also take more care to outline how regional averages were calculated etc.

The use of the three time periods is helpful but the last time period was unequal in terms of length (the first two periods were of 30 year durations, while the last was only 15 years). I suspect the choice of years may have been logistical, but the authors should evaluate and discuss the implications of this decision.

> The choice of years was one of data availability and a desire to exploit as much data as possible within the time constraints of available research time. As the other reviewer has mentioned this we propose to add an extra time period and extend the last one. This will provide five periods – 1931-1960, 1946-1975, 1961-1990, 1976-2005, and 1931-2005. This will allow all

> periods to be at least 30 years long and open up discussion issues mentioned by both reviewers. We believe the paper will significantly benefit from these extra analyses. We do not envisage this creating too much extra work as these data were already being processed as part of work that will follow this.

Third, as part of the online comments, a reviewer suggested the authors evaluate the implications of oscillatory dynamics that may affect the slopes of their relationships, particularly in the shorter 30 or 15 year time periods. This is quite important and the authors should consider this.

> Good point. We plan to rectify this as part of the modifications to the time periods investigated.

In terms of methods, the authors included too much detail in some areas (i.e. mathematics supporting Mann Kendall are not needed here), but not enough information in others (see specific comments).

> Fair points. We will seek to improve both the citation of the methods and provide extra clarity where needed, as outlined in response below and in those to reviewer 1.

While I found the manuscript and analysis interesting and useful, my largest criticisms relate to its length, lack of clear objectives and hypotheses, attempts to assess changes at both large regional and site specific scales, and limited attempts to compare their results with other large scale analyses in the literature. At 50 pages, 17 figures and 7 tables, the manuscript is far too long and unfocussed. I suggest the authors remove all site-specific analyses and focus the manuscript on the broader regional trends and drivers and novel aspects (e.g. lakes vs. rivers). If the large regions require some additional partitioning due to interregional variability that is fine, but the analysis and discussion of individual sites is not particularly useful for the reader and takes away from the more important regional assessment. From there, the authors should more clearly state their objectives and hypotheses, and focus the discussion on not only their results, but also improve the discussion of the implications of their findings and how they compare and contrast with other large-scale analyses (i.e. what is new and novel).

> These are all fair points and were also pointed out by reviewer 1. The authors were unsure about how best to tackle this specific issue as we worried some reviewers may request that extra level of detail (having presented it to others at a conference) and we thought it better to include it now, such that the reviewers can request it be removed. We are very happy to do this and believe there is scope with the other comments to improve the narrative as suggested.

The authors would also need to defend their correlation approach to identifying potential drivers, acknowledging the high likelihood of false positives and negatives, potential challenges wit using regionally averaged values over such large geographic areas where there is both high intra-region variability and where the geographic distribution of sites change, and the use of climate indices over large geographic areas. I have provided some more detailed comments below.

> Good points. We will make sure this is fully justified.

*There have been some issues in the below text owing to what are presumably some formatting issues with the online conversion to pdf and it is not always specifically clear which piece of the text the reviewer is referring to. Hopefully we have separated and interpreted the points raised accurately.*

Specific points

  c The introduction could use a clear set of objectives and hypotheses. This would greatly assist with improving the conciseness of the manuscript, which is quite long and unfocussed.

> These will be added to the revised manuscript as well as condensing of the text.

  c P9 L133+ The authors use climate variables (mean ´ monthly temperature, precipitation) from KNMI Climate Explorer. The data were 'downscaled' (averaged?) over 3 geographic regions (Fennoscandia, North America, Russia) and then used to assess relationships with spatially averaged ice-freeze up or break up dates. Is spatially averaging over such a large geographic area a good idea? There is evidence that there are large differences in the magnitude and even direction of ice phenology trends within the North American dataset. Would the strength of the analysis not be affected by geographic scale with weaker correlations expected as the study area increased?

> Fair point. We will improve the text discussing how this has been assessed. We would indeed expect weaker correlations as the study area increases but the considerable workload involved with correlating each ice phenology record within its own local climate meant we instead assessed the regional trends. The fact that we pick up some strong correlations between regional climate variables/indices and ice phenology means perhaps that our method – while more spatially limited – nonetheless adds value in establishing some of the key drivers for ice phenology changes. We will discuss the value and limitations of our approach in the manuscript.

  c Should section 3 (L. 153) not read '3. Results'?

> Sections 3 and 4 are a combination of results/discussion. We believed this was a good way of reducing the word count. With the suggested edits to remove the finer-scale analyses we will be able to refine titling.

â ´ AĎ c With ´ 60 pages of text (including 17 figs and 9 tables), this manuscript is quite long and highly unlikely to fit into the space requirements of the journal. The manuscript is also largely unfocussed, moving from broad discussions of phenological changes among regions to significant text devoted to individual sites. Much of the site-specific material is extraneous and distracts from the broad patterns. The objectives and hypotheses of the study are not clearly articulated and the closest we get is on page 3 where the authors state 'The paper explores the hemispheric spatiotemporal trends in ice phenology....Observed changes are then compared with climate records...' I would encourage the authors to devise more clear objectives and testable hypotheses. Further, I would encourage the authors to focus the current manuscript on a concise description of the broad phenological changes among the regions without delving into specific site responses. These are unnecessarily distracting. Such individual sites responses could either go into supplemental material or separate manuscripts.

As with other comments above, we will modify the text to remove the finer-scale discussions and focus the narrative at a larger-scale. We will also set the introduction up so it more clearly outlines the objectives of the paper.

âAˇ c In relation to ´ methods, the Mann Kendall trend tests with Sen's slopes is an appropriate statistical test for the monotonic trends, however the methods section does not indicate at what level the trends were considered significant. The correlation analysis, examining the potential drivers, is more suspect for a few reasons. First, there are a large number of correlations being tested (>800) and therefore a large potential for false positives or negatives. Second, at least two of the regions (NAM and RUS) have large geographic extents and the authors acknowledge in other locations in the manuscript that there are large regional differences in phenological trends (e.g. east and west of the Laurentian Great Lakes). Thus the use of a single averaged climate (temperature, precipitation, wind, etc.) value for a study region would not be expected to correlate well with the phenological trends. While considerably more work, it would be more useful to examine the relationships between downscaled climate drivers and phenological responses at individual sites (or smaller regions). This would also avoid the changing locations of sites between study periods.

The text around these issues will be clarified.

âAˇ c While I greatly appreciate the extent of ´ analysis undertaken, the presentation of results is at times confusing and potentially misleading. There are many figures (e.g. Figure 5, 7, 8, 11, 13, 14, 15; Tables 2, 3 [first panel] ) where trends are reported. However, the reader is unable to distinguish if these individual site trends are significant and, if so, at what level? If they are not significant, then they should not be reported as so. The legend should also indicate what the white boxes represent (e.g. insufficient data, no trend, something else?).

In the revisions we will make this clearer both in the text and on the figures. We have already drafted figure modifications that will make this easier.

âAˇ c Table 1. Assel et al. 2003. What is meant by 'maximum fraction of lake surface ce coverage'? Should this be maximum extent of lake ice coverage'?

Text will be reworked. It relates to the maximum extent as a proportion of total coverage reducing through time.

âAˇ c Table 1, ´ while useful, is quite large and is probably best suited to supplemental material. It also misses many references (Magnuson, Benson, Sharma, etc. etc.).

As stated in the caption the list was not designed to be exhaustive but to broadly cover papers showing the main trends in the different areas. Happy to add in additional references and move to the supplementary material if necessary.

âAˇ c Surprised ´ Benson et al. 2013 was not included in Table 1 given it covers such a large number of lakes

We are not sure of this paper. Is it the 2012 paper on extreme events? If so, this is cited. If not, we have missed the paper and apologise – we cannot find a reference for a 2013 paper led by Dr. Benson on google scholar.

âAˇ c P6 L86. The indication of rivers in brackets. Are these 88 sites on three ´ rivers, or 85 lake sites and 3 river sites?

The latter, the text will be clarified.

âAˇ c P6 L88 Change to are clustered around ´ the Laurentian Great Lakes.

Will change as requested.

âAˇ c P7 L99 Change Julian days to Ordinal days?

Will change as requested.

â ´ Aˇ c´ Note: Mann Kendall examines unimodal trends only

Will improve the text as requested.

âAˇ c P8 L103+ More information is ´ required on the statistical approach. At what level were trends considered significant? The first I see of this is on page 16 L180 where an alpha value of 0.1 was considered significant.

Text will be improved as suggested to make this clearer.

âAˇ c P8 L108+ The Mann Kendall test is widely used for time series analysis ´ and the mathematical details are widely available. No need to detail here.

Will remove and cite key references instead.

âAˇ c Figure ´ 2. The 3 shades of grey are not easily distinguishable.

Will modify to improve distinguishability.

âAˇ c P10. L154-166. This sec- ´ tion reads more like introductory material and methods than results.

This was meant to be a gentle introduction on how the results would follow. We will explore how this can be integrated elsewhere or removed entirely.

  P10. Lake ´ size or elevation can have large effects on freeze up or break up dates. Perhaps this (e.g. few lakes, highly variable lake size) might account for the lack of latitudinal trend. But note that in the description they indicated a fairly tight geographical extent in Russia (51.5-52 N, 104.5-105E).

Good point, will modify the text.

  P10. Comments on changes in European changes ´ (L174+) are qualitative... not assessed using a statistical approach. These are not easily distinguishable in the figure.

The text is indeed qualitative, we will explore how to improve the wording and add in some metrics that help to emphasise the key observations we want to point out.

  P10. L171-172. With the exception of 3 sites on ´ the eastern portion of the map, the sites in Russia (Figure 1a) appear largely to be of similar latitude, which precludes a latitudinal assessment. Thus, the case that 'This is the case for ...but not Russia' is not technically correct. Rather, a thorough assessment cannot be made due to limited data.

Good point, text will be tidied up regarding this issue.

  Figure 2. Why are the grids at different scales ´ between the 3 figure panels?

In the second time period there were values that were negative (i.e. breakup occurring prior to the year beginning, this is not the case in the other two time periods. We will edit this figure to give them a common x axis.

  Figure 3. The figure itself is good, but the caption ´ requires editing. E.g. 'Trends in ice breakup, ice freeze-up (is 'freezeup' a word?), and the duration of the open water period for lakes across North America, Europe (is this not referred to in other locations such as Figure 1 as Fennoscandian? Be consistent), and Russia.

Good points. We will edit the caption as suggested.

  P12. L196. 'When all sites are considered there is a clear increase...' I don't believe a statistical test was used to make this assertion. While it may appear obvious by the graphic, such visual assessments require a statistical assessment.

The graphic is designed to present a simple statistical summary of trend directions. We are not sure how to answer this query, it is clear a greater proportion of sites through time display a trend towards warming, we are describing the observations from the statistics.

  ´ P12. L201. 'For Russian sites it is clear...' Again, there is no statistical test on which to base these claims. These are qualitative assessments only.

> These are qualitative assessments of the trend statistics that have been summarised in the figure.

âAˇ c P13 L203. Again aˊ reference to the 'large area' of the Russian sites.

> Text will be clarified.

âAˇ c P15 L60. Note there are many ˊ lakes in Canada being monitored for ice-phenology, they are just not recorded in the database or captured in this manuscript.

> Text will be modified.

âAˇ c What was the criteria for whether a site ˊ could be included or not. Was it 90% of records for the 15, 30, or 75 year periods? For the 75 year trend test, were only lakes that had 90% of records included for the whole period included?

> The criteria were for each individual time period. During each time period a maximum of 10% missing data were allowed for each study site to be included. This provided the maximum flexibility in order to include the maximum amount of data.

âAˇ c P14. Table 2. Please include some additional details in ˊ the table caption: The statistical test used, the significance value of the trend. Within the table n values and confidence intervals would be helpful. There is value in the table, but it is hampered by the large differences in study areas (e.g. latitudinal extent) and differences in the number of sites between time periods (esp. in Russia and North America). This can influence the qualitative assessments provided in the text, for example in North America between the middle and last time periods where a large number of northern sites dropped off. Further, it would be useful for Figure 1 (panels b-c) could have lake and river sites distinguished independently. This would help to see if lake and river sites were distributed homogenously or if they are clustered at certain latitudes.

> We will make these changes and improve readability of the data in the table.

âAˇ c P15 L261. Awkward sentence. Reword.

> Agreed. Will reword.

âˊAˇ c P16. L212 'The ˊ data show…'. Larger than what? Where is the statistical test? This is also true for the general pattern discussed for Europe and North America. The wording 'do not appear to change significantly…', was this an eyeball assessment?

> We think this is the text on page 13(?). The text can be made clearer here in the first part. The discussion on the change is based on both a qualitative assessment of the graphs, but also an observation of the data that are included within it. We will modify the text to ensure the numbers reflecting the qualitative changes is clearer.

âAˇc P16. L279. While ´ there is some value pointing out that there are exceptions to the general trend, this is obvious in Figure 3. I think this figure can be referred to rather than describing specific lake or river systems that behave differently.

    We will make these changes.

âAˇc P 16. The use of standard deviation ´ to assess interannual variability should be introduced in the methods. Why use numbers (14) and letters (eight).

    We will introduce SD into the methods. Happy to modify the use of numbers/letters, they were presented like this because its states in the journal guidelines to "use words for cardinal numbers less than 10; use numerals for 10 and above (e.g. three flasks, seven trees, 6 m, 9 d, 10 desks)" – we have no strong opinion on this as it has no impact on the study and will change if necessary.

âAˇc P17. L301. The reporting of mean values should be ´ accompanied by estimates of variance, the degrees of freedom, and the significance of the trends.

    We will add in additional data to improve the statistics.

âAˇc P17. L303+. 'Only four North American sites experience later breakup ´ during...' The challenge here is that I can't tell if the authors are assessing significant or significant + non-significant trends. Was the example of Frame Lake a significant or non-significant change? The slope is reported (which appears quite close to zero), but no p value or measures of variance.

    Text will be clarified and partly removed as part of refocusing on larger-scale patterns requested by both reviewers.

âAˇc P17. L309+ Given the large scope of the ´ manuscript, it seems ill advised to drill down to the extent here for individual lakes or rivers. A blow out box may be more appropriate for dealing with specific case studies if there is a strong need to provide an example. The removal of these details will make it much easier for the reader to follow the broader relevance of the study and more space for the authors to discuss the relevance of their analysis compared to similar studies.

    These finer details will be removed. They were originally included as we felt they provided an interesting narrative on why sites might vary. This will be sacrificed in favour of the larger-scale analyses requested by both the reviewers.

âAˇc P49. L826+. I'm not sure I follow the logic here. How do summer wind ´ speeds alter ice break up in the spring (either the preceding or following spring)? This is likely a false positive and one would expect some false positives and negatives given the large number of tests. Second, the link between summer windspeed and effects on turbulence are only relevant once air temperatures drop below 0 deg C. Summer windspeeds could be important by deepening thermoclines and higher heat storage,

leading to delayed ice out dates. Again, potential for false positive here. Further, I am not surprised by the poor relationships over North America and Russia given the large geographic area and changes in the distribution of sites. As noted by the authors in other locations (e.g. p18 L325), there can be large regional differences in climatic patterns within the study area that can lead to regional differences in ice phenology trends (e.g. NAM sites). While more labour intensive a better approach would be to examine the linkages between climate and ice phenology at the individual sites, and then bringing these relationships into a more global analysis (i.e. does ice phenology respond similarly to climate across sites with regional differences in climate causing regional differences in ice phenology, or are other factors (e.g. lake or watershed morphometry, lake vs. river, etc. critical).

The text here will be clarified and analyses made clearer as requested (and described above).

âAˇ c Figure 4. Caption should indicate that trends were ´ tested using Mann Kendall and Sens slopes. After looking at Figure 3 from Duguay et al. 2006 (which I think is largely the same data), most of the ice freeze up values (panel e in this manuscript) would appear to be non-significant. This makes me wonder whether all trends are reported regardless of significance. The authors need to clearly state in the figure caption whether the trends being reported are significant and what level of significance.

All trends are reported, a lack of statistical significance does not mean a trend is not present. But we agree that in its current format this is potentially confusing. We will edit the figure and the text that accompanies it to make the distinctions clearer.

âAˇ c Table 3 is confusing. The authors must indicate what statis- ´ tics were used (Mann Kendall with Sen's slopes?). What do the numbers represent in the panels represent (Sen's slopes?). What are their units (days/decade?). Why are significance values included only in the last three panels/time periods but not the first panel? Does the dash represent 'no data available'? Why not simply use the first panel data? Note, that another interpretation of the data from this table is that few lakes or rivers are displaying significant trends. For example in the 1931-90 period only 4/15 sites show significant changes at the p=0.05 level, or 7/15 at the p=0.10 level. Overall the table is interesting, but these site specific tables should probably be moved to supplemental material.

As part of the refocusing of the text some of the material in this table will largely be removed. The units are days per decade, and significance was only shown for the longer time series as that was discussed in the text. In hindsight it was an error to leave this out for the shorter time periods. The table was meant to relate to how depending upon which timeframe you use the shorter time periods might display trends superimposed on longer-term trends that were more variable. We will improve this tables and present them in the supplementary material as they will become a smaller part of the text upon revision.

âAˇ c Table 3 shows that from the period 1991-2005 the major- ´ ity of N America Lakes included in the analysis had either no significant trend, or had later ice on dates (significant positive numbers). There were no significant negative numbers. Yet, when I examine Figure 4, I see a large number of earlier freeze up dates reported (particularly to the east of the Laurentian Great Lakes). Am I missing something?

This is related to the fact that there are study sites within the 1991-2005 time period that do not have 90% data in the other time periods. This means they were not included on the table as the aim of the table was to show how trends might vary at the sites with data available for multiple time periods. In the refocusing of the manuscript these issues will be resolved.

âAˇ c Figure 5 caption is poorly written. What does 'Comparison of how ´ sites in North America with an open water season calculation' mean? Perhaps 'Patterns of ice freeze-up and ice breakup....' What level of significance? What statistical test? Were non-significant trends reported as trends? Do white boxes represent no significant trend or no data?

Text will be fully clarified to include the information requested. The figure was meant to show how the changes in the open water seasons compares to changes in the breakup and freezeup dates that drive it i.e. the reader should be able to see the trend directions during that time period for either and compare that to the overall changes in the number of open water days per year.

âAˇ c Figure 17. This figure is important as it represents ´ the only example of where the drivers of the phenological trends were being tested. Although the methods are not entirely clear, the analysis seems to have a few issues. I am assuming that the authors assessed the trends of the 'downscaled' annual mean or median values of the freeze-up breakup dates over the full study period (1931-2005) with each study region (FEN, NAM, RUS). How was the air temperature data treated? Was it the mean value of the whole study area? This presents some problems because in both North America and Russia, the study area is quite large and the number and geography of sites changed dramatically over the study period. For example, no Canadian sites were available during the most recent period. In this case, I would think that this would have a large effect on the relationship between air temperature and ice phenology. For Russia, there was a wide spread of sites (25-27 sites) longitudinally during the 1961-1990 period, but much fewer sites during the periods before (5-6 sites) and after (1 site). A better approach would be to examine the downscaled air temperature values for each site specifically. This would also remove potential problems associated with elevation. Further, a large number of correlations are being tested (e.g. >400 in panel b), which leads to the potential for false positives. A minor point, but presumably the bolding pattern refers to the level of significance of the trends. This would be important to point out in the caption. Perhaps I have misunderstood a few aspects of these analyses and in general more clarification of the statistical approach is needed.

The considerable workload involved with correlating each ice phenology record within its own local climate meant we instead assessed the regional trends. The fact that we pick up some strong correlations between regional climate variables/indices and ice phenology means perhaps that our method – while more spatially limited – nonetheless adds value in establishing some of the key drivers for ice phenology changes. We will discuss the value and limitations of our approach in the manuscript.

---

## Author Response (AR1)

Dear Professor Duguay,

We are grateful to both reviewers for the time they have taken to offer such detailed and insightful comments. The comments from both reviewers were broadly well-aligned and we were happy to make the suggested changes. We are confident that the changes have made the manuscript significantly stronger as a result. All changes are outlined in detail below, but the main changes include:

- The text was refocused on the narrative around large-scale changes and discussion of the finer detailed analyses was removed. The paper is as a result significantly shorter (~25%) and contains fewer figures and tables. We believe the manuscript is now much easier and simpler to read.

- The methods were clarified and modified to include extra analyses on the SOI as request.

- The time spans used were modified to include a new one (1946-1975) and modify an existing one (1991-2005 changed to 1976-2005). This overlapping allows for a greater comparability between the different time periods and removes any bias caused with the implementation of the shorter time period in the original manuscript.

- Figures and captions were improved to ensure the discussion of the statistics is as clear as possible.

- In general the text was modified to make the objectives clearer at the start of the paper. We also modified the discussion to take account of specific points that have been raised by the reviewers and improved it by integrating the results with the published literature more effectively.

In the blue text below are the responses to the individual referee comments.

**Comments from John Magnuson**

General comments: The paper provides a detailed analysis of a larger group of lakes and streams over 74 years from 1963 to 2005. Data are from the collection of data in the Snow and Ice Data Center supplemented largely with Swedish, Finnish data. The analysis of slopes were over 4 time periods, 1931-1960, 1961-1990, 1991-2005, and 1931-2005. There are a number of new findings that included comparisons of lakes vs streams, changes in open water duration, differences among regions, and differences among the year subgroupings. I have organized my comments below in terms of what I liked and what I have concerns about.

What I liked.

Was a Northern Hemisphere analysis rather than a locally constrained analysis. The regional comparisons are useful. Made an honest attempt to include all of the data unless there were too many missing values. Good idea.

Used fixed time periods with less than 10% missing years. Good idea. Other researchers have occasionally had difficulties using these data because they did not constrain the time periods and therefore mixed the influences of longer term changes over time.

Analyzed rivers as well as lakes. Good idea.

Looked at the length of the open water duration. Good idea. Most limnologists have looked at the duration of ice cover over a winter season, rather than the duration of the open water over an entire open water season. Might have been interesting to think about how that changes one's perspective. Both ice cover and open water seasons play a role in the biogeochemical cycling and other ecological phenomena as related to ice cover. Is the change in one more important ecologically than the change in the other? Might have been useful to show relation over time of both ice cover and open water. Even though they relate to two sets of years, they should be highly correlated

> This is an interesting point and we looked at this further to see how we could include data related to the ice season length. However, the ice cover duration results did not really add anything to the narrative as, as the reviewer says, the data are highly correlated. There certainly is an interesting argument to be made about the geochemical cycling, though we preferred for the paper to concentrate on the climatology, so we have elected not to provide detail on this, but to instead present the possibility that such geochemical issues are worth investigating further.

Consideration of heterogeneity among lakes, time periods, and regions as well differences in variability. Good idea. I think they might have considered making the comparisons in regard to a set of issues, questions, or major or in respect from what they know from the ice cover literature. "Looking in some detail at regional differences" was a good idea such as northern and southern Sweden, or Europe and North America. But I am not convinced that I agree with their explanations.

> The entire text around this has been largely rewritten to account for this comment and the modification of the time periods.

I liked the quick reference list with findings from each published paper they found. Good idea. However, the results from these papers were not integrated or their findings or compared with this paper's findings. Findings stated as new were already found in the various published analyses.

> In the new discussion section we have made a better reference to published work as well as the implications of the current study.

What concerned me.

1. Did not indicate what was actually a new finding. It was all new, perhaps because it included all of the usable data between 1963 and 2005. But items in their final list of findings have often been reported as general findings by earlier researchers using various time periods or regions. I would have appreciated their integrating what they found with earlier findings. Such interpretations were usually absent except for a few general statements in the introduction that was an incomplete citation history. Many of the papers were in the list of papers and major results, but they were not integrated into a discussion of their own findings. How do your results compare with Benson et al 2012 using a smaller number of lakes but with longer time series; she looked at 150 year trends, 100 year trends and 30 year trends using largely the same data source that the authors used. So, "what was new", was that they used the most robust data set that existed on inland-water ice phenology. I thought the most defensible grouping of years was of the full duration of years for the waters they analyzed. The least defensible was the short time series in the most recent years. Perhaps I missed it, but I would have enjoyed reading more about the rationales for the year subgroupings they decided to analyze. Why not use three equal year groupings of the same length. How does the number of years in a set influence the results?

> Fair points. In the revision we have more clearly presented the main findings and integrated them with what has already been published. We understand the point on the timescale groups. We had originally presented it this way as something similar had been used in one of the papers we cited (Šarauskienė and Jurgelėnaitė, 2008) and we felt it was effective given the distribution of the data availability. There was a trade-off between including the maximum amount of data, and the groupings we originally used allowed us to look at the decadal change across three different periods of time and employ the maximum amount of data as possible without increasing the workload too much. The second reviewer has also raised concerns with these timespans. We have modified the timescales by adding a 1946-1975 time period and replacing the 1991-2005 time span with a 1976-2005 time period. We now have several overlapping time periods and ~1000 new time series added into the analyses. In the revision the results/discussion was extensively reworked to focus on: 1) the spatio-temporal variability between different time periods, and 2) how sites with continuous data appear to vary between them. We are confident this has significantly improved the paper and has removed the concerns raised.

2. Long term. They referred to the time series analyzed as long term, but were they long enough? Unfortunately, they did not cite an important paper by R. Wynne (2000) that first revealed that running slopes of the lake ice time series alternated from positive and negative when analyzed using consecutive moving windows. Wynne looked at 4 lakes with 100 year time series of ice breakup (2 in North America and 2 in Europe). The running means both of 20 and 50 years across the time series oscillated over the 100 years between positive and negative slopes. Whether one observed a positive

or a negative slope depended on the start date of the subset. See (Wynne, R. H. 2000. Statistical modeling of lake ice phenology: Issues and implications. Verhandlungen des Internationalen Verein Limnologie 27:2820–282).

> It was an oversight not to cite the paper or include it as a part of the discussion. We touched upon some of the main observations of switching trends indirectly when looking at the 1931-2005 time period, but we agree this needed to be more explicitly discussed and improved upon. The observations from the Wynne paper are something we intended to pursue in the follow up to this current paper as it requires a considerable amount of extra statistical analyses. However, we have added in the revision a greater effort to explore the implications of this work and compare it with ours, including a new figure looking at running means. We agree this is important and it is has led to a couple of new discussion points related to how we interpret ice phenology data and the potential pitfalls for sites with limited data.

3. Oscillatory dynamics. Papers on the oscillatory dynamics of longer term data reveal a number of interacting oscillations. (e.g. Sharma, Sapna, and John J. Magnuson. 2014. Oscillatory dynamics do not mask linear trends in the timing of ice breakup for Northern Hemisphere lakes from 1855 to 2004. Climatic Change 124:835-847.) Again, the slope of subsets selected would depend on the start date of the subset. Take a look at Sapna's Figure 5 panel D and consider the result of having started a 30-year period in 1948 versus 1961 or 1960. Some of the most significant oscillations in the Sharma paper were in the range of El Nino that could easily mess up the interpretations of the shortest time period the authors used for recent years. Their most recent date group was short enough that an El Nino near the end or the beginning of that short series could be relevant. Interpreting it as a more general long-term change is problematic. Many published papers have analyzed these oscillations and the authors should have at least discussed the issue and how that might have influenced their conclusions. They cited most of them in their listing. I think it would have been interesting to do an analysis like Sapna's over the 74 years for different regions. A simple set of running means as in her figure 5 might be sufficient. Then compare those rather than for the three periods the author's used (two of equal length and one short).

> These are good points that have been raised. The suggested edit to the timespans above has rectified some of the comments and we have used the ideas from Sharma and Magnuson (2014) to improve both our results and the discussion around them.

4. A point on the large-scale climate drivers like NAO or AO. The Livingstone (2000) paper they originally credited for the roll of these kinds of drivers on ice phenology found relations not only with NAO but also with the SOI. Lake Mendota, for example, was influence by both. I think the authors should have included SOI in their analyses. (Livingstone, D.M., 2000. Large-scale climatic forcing detected in historical observations of lake ice breakup. Verhandlungen des Internationalen Verein Limnologie 27:2775– 2783.). Other papers the author's cited also found that SOI was also an important correlate.

> In keeping with the reviewer comment, we have now included correlations with lake ice phenology and SOI. These correlations can be seen in Figure 8. We have updated the methodology to reflect this additional data analysis.

5. What other papers pointed out that ice cover was related to air temperature and the climatic variables. A discussion that included how these results compare with the author's findings would have been interesting to me.

> As part of the revamped discussion section we have more explicitly discussed this issue.

6. The comparisons they made with lakes and streams were new and I liked that. The only other paper to have done that was the Magnuson et al. 2000 paper in Science. Did Magnuson et al. see a difference between lakes and streams? If not, this is a new finding. I thought that the authors causal explanation for the difference between lakes and flowing waters could be expanded as it did not include some of the most important processes. I think additional factors such as the greater depths of most lakes and the thermal stratification of lakes both should play a role. Rivers tend to mix over the entire water column, only shallow lakes do.

> This is an important point and something we were keen to develop in follow-up work. However, we see the benefits here of a better discussion around these topics and have boosted this in the revised manuscript.

7. I would like to see more explanations of the results based on the authors findings and the literature on the inland water cryosphere. A more in depth explanation of why freeze dates are less directly related of air temperature than are breakup dates, for example. There are brief interpretations of that observation in many of the papers they cite. The literature on the various mechanisms generally are not cited but are known.

> In our edits of the discussion section we have added material on this. We have elected not to go into too much detail due to the previous desire to regionalise the analyses, but we have commented and cited where appropriate.

8. The paper was a bit exhausting owing to the detail and number of comparisons. It would have been good either to identify major issues rather than describing all of the detailed results or to focus the structure on the regional comparisons. This ended up being a description of what was happening everywhere in great detail. To some extent this is basically a detailed descriptive data report.

9. The authors frequently tried to explain differences among individual lakes rather superficially, eg. Mystery vs Nebish, or Mendota vs Monona. May not have the local knowledge to work at that fine individual lake scale. I would put the analyses into the broader result rather pick apart individual lakes.

> We consider these two points together and they are both very fair. The authors previously discussed extensively about whether reviewers would request that detail or whether they would prefer the focus be on the larger-scale – there are merits on both sides of this and we figured it would be easier in any revision to broaden the scale of the analyses rather than the other way around. Given the comments raised by both reviewers, and the format of publication, we were very happy to refocus and condense the text by concentrating on the major issues and instead leaving some of the finer details for either citations or follow-up work. The paper is significantly reduced in both figure and word count and we believe this slimmer version is hopefully less exhausting and consequently an easier read.

Other suggestions: 1. I would give the years of observation in the title, i.e. 1931-2005.

Good idea. Done.

Minor points.

In the maps take a look at whether the shades of grey are distinguishable from each other or the background. I had difficulties doing so.

We revamped the figures to ensure they had sufficient clarity. This generally involved simplifying the figures.

Madeline Island and Bayfield data are essentially identical and rightly so. If it is the data set with which I am familiar it comes from the periods of ice cover that excludes boats and ferries from entering or leaving Bayfield. The ferry runs from Bayfield to Madeline island in open water seasons and an ice road is used in winter between the two locations. The early paper on these data is: (Howk, F. Changes in Lake Superior ice cover at Bayfield, Wisconsin. J. Great Lakes Res 35, 159–162 (2009)). There is an additional Chequamegon data set farther up the Bay.

With the decision to regionalise the approach this is less of an issue now.

**Comments from Anonymous Referee #2**

General comments: The authors have collected northern hemisphere ice phenology data from several large databases and climate data from an online portal. They have split the ice phenology data into three large geographic regions: North America, Fenoscandia, and Russia and within each region they have examined the overall ice phenology trends and also site (lake or river) specific ice phenology trends over four time peirods. Finally, the authors have used a correlation analysis approach to examine potential relationships between several climate drivers (air temperature, precipitation, wind speed) and ice phenology metrics for the three large regions. The manuscript contains at least two novel components: its geographic scope is large and is split into three reasonably large regions (North America, Fenoscandia, Russia), and the analysis contains an assessment of potential changes in ice-phenology for both lakes and rivers. The general statistical approach of using Mann Kendall trend tests with Sen's slopes to evaluate both the significance and slopes of monotonic trends has been commonly applied to similar datasets. The use of correlation analysis to assess potential drivers is also common, but there is a strong potential for false positives and negatives due to the large number of tests.

> We have improved the discussion of these issues in the manuscript and more clearly outlined any caveats. More detailed comments are below on the individual line comments. The results and discussion have been completely revamped to take account of the comments from both reviewers.

Further, the authors appear to have used mean climate conditions to within each study region for the correlation analyses. However, given the large size of the study regions and large inter-regional variation of trends (e.g. to the east and west of the Laurentian Great Lakes) such an approach may not be appropriate to assess the linkages between ice phenology trends and climate drivers. There is also some uncertainty of how the regional averages were calculated and if these changed over time as the number and location of sites within the regions changed. For example, there were no data included for Canada during the last time period (only US sites)....did the area used to calculate the 'mean' climatic conditions change?

> We elected for this regionalised strategy because (1) the computational and human resources needed to analyse climate records for each individual site are much higher; and (2) we were interested in establishing broader regional climate drivers of ice phenology rather than developing correlations with local climate, which we would expect to be very strong. We have added this text to the Materials and Methods section. The fact that we pick up some strong correlations between regional climate variables/indices and ice phenology provides a less obvious and more interesting finding than expected correlations with local climate – i.e. that broad regional climate exerts considerable influence over ice phenology. We have discussed the value and limitations of our approach in the Causes of Ice Phenology Change section. We have also added a discussion point around the implications of this as we feel the underlying result is potentially quite important.

> We have also added text to clarify that the spatially-averaged regional time series for the three geographical regions encompassed only study sites with data for the full 1931-2005 time period. In this respect, the geographical area used to calculate the 'mean' climatic conditions did not need to change with changing time periods for sites.

The use of the three time periods is helpful but the last time period was unequal in terms of length (the first two periods were of 30 year durations, while the last was only 15 years). I suspect the choice of years may have been logistical, but the authors should evaluate and discuss the implications of this decision.

The choice of years was one of data availability and a desire to exploit as much data as possible within the time constraints of available research time. As the other reviewer has mentioned this we have added an extra time period and extended the last one. This now provides five periods – 1931-1960, 1946-1975, 1961-1990, 1976-2005, and 1931-2005. This allows all periods to be at least 30 years long and opens up discussion issues mentioned by both reviewers. We believe the paper has significantly benefitted from these extra analyses, with ~1000 extra time series being presented. The overlapping 30-year time periods also provides for a much cleaner and easier to justify format and we believe has significantly improved the manuscript by making it more logical and the results more robust.

Third, as part of the online comments, a reviewer suggested the authors evaluate the implications of oscillatory dynamics that may affect the slopes of their relationships, particularly in the shorter 30 or 15 year time periods. This is quite important and the authors should consider this.

Good point. This is rectified by the modifications to the time periods investigated.

In terms of methods, the authors included too much detail in some areas (i.e. mathematics supporting Mann Kendall are not needed here), but not enough information in others (see specific comments).

Fair points. We have improve the citation of the methods and provided extra clarity where needed, as outlined in response below and in those to reviewer 1.

While I found the manuscript and analysis interesting and useful, my largest criticisms relate to its length, lack of clear objectives and hypotheses, attempts to assess changes at both large regional and site specific scales, and limited attempts to compare their results with other large scale analyses in the literature. At 50 pages, 17 figures and 7 tables, the manuscript is far too long and unfocussed. I suggest the authors remove all site-specific analyses and focus the manuscript on the broader regional trends and drivers and novel aspects (e.g. lakes vs. rivers). If the large regions require some additional partitioning due to interregional variability that is fine, but the analysis and discussion of individual sites is not particularly useful for the reader and takes away from the more important regional assessment. From there, the authors should more clearly state their objectives and hypotheses, and focus the discussion on not only their results, but also improve the discussion of the implications of their findings and how they compare and contrast with other large-scale analyses (i.e. what is new and novel).

These are all fair points and were also pointed out by reviewer 1. The authors were unsure about how best to tackle this specific issue as we worried some reviewers may request that extra level of detail (having presented it to others at a conference) and we thought it better to include it now, such that the reviewers can request it be removed. The results and discussion have largely been reframed due to the shift in time periods and the suggestion to

regionalise the analyses. The manuscript is ~25% shorter and we have also more clearly outlined the objectives and done a better job with discussing our results. We believe the resulting manuscript edits have led to a more easily readable manuscript.

The authors would also need to defend their correlation approach to identifying potential drivers, acknowledging the high likelihood of false positives and negatives, potential challenges wit using regionally averaged values over such large geographic areas where there is both high intra-region variability and where the geographic distribution of sites change, and the use of climate indices over large geographic areas. I have provided some more detailed comments below.

Good points. We have modified the text to ensure clarity on this issues.

*There have been some issues in the below text owing to what are presumably some formatting issues with the online conversion to pdf and it is not always specifically clear which piece of the text the reviewer is referring to. Hopefully we have separated and interpreted the points raised accurately.*

Specific points

  c The introduction could use a clear set of objectives and hypotheses. This would greatly assist with improving the conciseness of the manuscript, which is quite long and unfocussed.

This has been modified as suggested.

  c P9 L133+ The authors use climate variables (mean ´ monthly temperature, precipitation) from KNMI Climate Explorer. The data were 'downscaled' (averaged?) over 3 geographic regions (Fennoscandia, North America, Russia) and then used to assess relationships with spatially averaged ice-freeze up or break up dates. Is spatially averaging over such a large geographic area a good idea? There is evidence that there are large differences in the magnitude and even direction of ice phenology trends within the North American dataset. Would the strength of the analysis not be affected by geographic scale with weaker correlations expected as the study area increased?

We elected for this regionalised strategy because (1) the computational and human resources needed to analyse climate records for each individual site are much higher; and (2) we were interested in establishing broader regional climate drivers of ice phenology rather than developing correlations with local climate, which we would expect to be very strong. We have added this text to the Materials and Methods section.

The fact that we pick up some strong correlations between regional climate variables/indices and ice phenology provides a less obvious and more interesting finding than expected correlations with local climate – i.e. that broad regional climate exerts considerable influence over ice phenology. We have discussed the value and limitations of our approach in the Causes of Ice Phenology Change section.

  c Should section 3 (L. 153) not read '3. Results'?

Sections 3 and 4 were originally a combination of results/discussion. We believed this was a good way of reducing the word count. However, with the suggested edits to remove the finer-scale analyses we have refined this into two results chapters (each focusing on different things) and a more clearly defined discussion chapter.

â´Ă˘ c With ´60 pages of text (including 17 figs and 9 tables), this manuscript is quite long and highly unlikely to fit into the space requirements of the journal. The manuscript is also largely unfocussed, moving from broad discussions of phenological changes among regions to significant text devoted to individual sites. Much of the site-specific material is extraneous and distracts from the broad patterns. The objectives and hypotheses of the study are not clearly articulated and the closest we get is on page 3 where the authors state 'The paper explores the hemispheric spatiotemporal trends in ice phenology....Observed changes are then compared with climate records...' I would encourage the authors to devise more clear objectives and testable hypotheses. Further, I would encourage the authors to focus the current manuscript on a concise description of the broad phenological changes among the regions without delving into specific site responses. These are unnecessarily distracting. Such individual sites responses could either go into supplemental material or separate manuscripts.

As with other comments above, we have modified the text significantly to account for these issues. The paper is now ~25% shorter and much more streamlined. The narrative is, we believe, significantly more focused and allows for a clearer view of the results we have developed and why we think they are important.

  c In relation to ´ methods, the Mann Kendall trend tests with Sen's slopes is an appropriate statistical test for the monotonic trends, however the methods section does not indicate at what level the trends were considered significant. The correlation analysis, examining the potential drivers, is more suspect for a few reasons. First, there are a large number of correlations being tested (>800) and therefore a large potential for false positives or negatives. Second, at least two of the regions (NAM and RUS) have large geographic extents and the authors acknowledge in other locations in the manuscript that there are large regional differences in phenological trends (e.g. east and west of the Laurentian Great Lakes). Thus the use of a single averaged climate (temperature, precipitation, wind, etc.) value for a study region would not be expected to correlate well with the phenological trends. While considerably more work, it would be more useful to examine the relationships between downscaled climate drivers and phenological responses at individual sites (or smaller regions). This would also avoid the changing locations of sites between study periods.

The text around these issues has been clarified.

  c While I greatly appreciate the extent of ´ analysis undertaken, the presentation of results is at times confusing and potentially misleading. There are many figures (e.g. Figure 5, 7, 8, 11, 13, 14, 15; Tables 2, 3 [first panel] ) where trends are reported. However, the reader is unable to distinguish if these individual site trends are significant and, if so, at what level? If they are not significant, then they should not be reported as so. The legend should also indicate what the white boxes represent (e.g. insufficient data, no trend, something else?).

This is a great point that we had not sufficiently thought through fully in figure preparation. The new figures contain a lot more information than previously, but are also simpler and fewer. This allows for all of these issues to be rectified.

  c Table 1. Assel et al. 2003. What is meant by 'maximum fraction of lake surface ce coverage'? Should this be maximum extent of lake ice coverage'?

Text modified.

  c Table 1, ´ while useful, is quite large and is probably best suited to supplemental material. It also misses many references (Magnuson, Benson, Sharma, etc. etc.).

As stated in the caption the list was not designed to be exhaustive but to broadly cover papers showing the main trends in the different areas, more papers could of course be added but would compromise the simplicity. If there are specific papers the reviewer wants added, then we will be happy to do so – a lack of a specific reference is not a comment on the paper, just that we tried to provide a range of studies. We have elected to keep this in the main manuscript as we believe it provides a nice cheat sheet for the study areas we are looking at in the paper (the other reviewer also liked this). Open to discussion on this issue if necessary, but we believe it is useful for the reader who might not be familiar with the

  c Surprised ´ Benson et al. 2013 was not included in Table 1 given it covers such a large number of lakes

We are not sure of this paper. Is it the 2012 paper on extreme events? If so, this is cited. If not, we have missed the paper and apologise – we cannot find a reference for a 2013 paper led by Dr. Benson on google scholar.

  c P6 L86. The indication of rivers in brackets. Are these 88 sites on three ´ rivers, or 85 lake sites and 3 river sites?

The latter, the text has been clarified.

  c P6 L88 Change to are clustered around ´ the Laurentian Great Lakes.

Changed.

  c P7 L99 Change Julian days to Ordinal days?

Changed as requested.

â ´ Aˇ c´ Note: Mann Kendall examines unimodal trends only

Changed as requested.

  c P8 L103+ More information is ´ required on the statistical approach. At what level were trends considered significant? The first I see of this is on page 16 L180 where an alpha value of 0.1 was considered significant.

Text improved as suggested to make this clearer.

  c P8 L108+ The Mann Kendall test is widely used for time series analysis ´ and the mathematical details are widely available. No need to detail here.

Removed and references cited instead.

  c Figure ´ 2. The 3 shades of grey are not easily distinguishable.

Figure has been significantly modified to improve distinguishability and information.

  c P10. L154-166. This sec- ´ tion reads more like introductory material and methods than results.

This was meant to be a gentle introduction on how the results would follow – we have reworked text around this.

  c P10. Lake ´ size or elevation can have large effects on freeze up or break up dates. Perhaps this (e.g. few lakes, highly variable lake size) might account for the lack of latitudinal trend. But note that in the description they indicated a fairly tight geographical extent in Russia (51.5-52 N, 104.5-105E).

Good point, text modified.

  c P10. Comments on changes in European changes ´ (L174+) are qualitative... not assessed using a statistical approach. These are not easily distinguishable in the figure.

The text is indeed qualitative, we have improved the wording and added in some metrics that help to emphasise the key observations we want to point out.

  c P10. L171-172. With the exception of 3 sites on ´ the eastern portion of the map, the sites in Russia (Figure 1a) appear largely to be of similar latitude, which precludes a latitudinal assessment. Thus, the case that 'This is the case for ...but not Russia' is not technically correct. Rather, a thorough assessment cannot be made due to limited data.

Good point, text has been tidied up regarding this issue.

  c Figure 2. Why are the grids at different scales ´ between the 3 figure panels?

In the second time period there were values that were negative (i.e. breakup occurring prior to the year beginning, this is not the case in the other two time periods. Figure has been significantly edited to account for this.

âAˇ c Figure 3. The figure itself is good, but the caption ´ requires editing. E.g. 'Trends in ice breakup, ice freeze-up (is 'freezeup' a word?), and the duration of the open water period for lakes across North America, Europe (is this not referred to in other locations such as Figure 1 as Fennoscandian? Be consistent), and Russia.

Good points. Edited the caption as suggested.

âAˇ c P12. L196. 'When all sites are considered there is a clear increase...' I don't believe a statistical test was used to make this assertion. While it may appear obvious by the graphic, such visual assessments require a statistical assessment.

The graphic is designed to present a simple statistical summary of trend directions. We are not sure how to answer this query, it is clear a greater proportion of sites through time display a trend towards warming, we are describing the observations from the statistics. We have made improvements to the figure to include statistical significance. Hopefully this helps.

âAˇ c´ P12. L201. 'For Russian sites it is clear...' Again, there is no statistical test on which to base these claims. These are qualitative assessments only.

These are qualitative assessments of the trend statistics that have been summarised in the figure.

âAˇ c P13 L203. Again a ´ reference to the 'large area' of the Russian sites.

Text had been clarified.

âAˇ c P15 L60. Note there are many ´ lakes in Canada being monitored for ice-phenology, they are just not recorded in the database or captured in this manuscript.

Text modified.

âAˇ c What was the criteria for whether a site ´ could be included or not. Was it 90% of records for the 15, 30, or 75 year periods? For the 75 year trend test, were only lakes that had 90% of records included for the whole period included?

The criteria were for each individual time period. During each time period a maximum of 10% missing data were allowed for each study site to be included. This provided the maximum flexibility in order to include the maximum amount of data. Text clarified.

  c P14. Table 2. Please include some additional details in ´ the table caption: The statistical test used, the significance value of the trend. Within the table n values and confidence intervals would be helpful. There is value in the table, but it is hampered by the large differences in study areas (e.g. latitudinal extent) and differences in the number of sites between time periods (esp. in Russia and North America). This can influence the qualitative assessments provided in the text, for example in North America between the middle and last time periods where a large number of northern sites dropped off. Further, it would be useful for Figure 1 (panels b-c) could have lake and river sites distinguished independently. This would help to see if lake and river sites were distributed homogenously or if they are clustered at certain latitudes.

> We made these changes and improved readability of the data in the table. The improvements to Figure 3-7 and Table 3 now mean that looking at them all together provides the basis of all the results presented. Figure 1 has been remade and accounts for suggestions from both reviewers.

  c P15 L261. Awkward sentence. Reword.

> Done.

â ´ Aˇ c P16. L212 'The ´ data show...'. Larger than what? Where is the statistical test? This is also true for the general pattern discussed for Europe and North America. The wording 'do not appear to change significantly...', was this an eyeball assessment?

> We think this is the text on page 13(?). The text has been made clearer here in the first part. The discussion on the change is based on both a qualitative assessment of the graphs, but also an observation of the data that are included within it. We have modified the text to ensure the numbers reflecting the qualitative changes is clearer.

  c P16. L279. While ´ there is some value pointing out that there are exceptions to the general trend, this is obvious in Figure 3. I think this figure can be referred to rather than describing specific lake or river systems that behave differently.

> Done.

  c P 16. The use of standard deviation ´ to assess interannual variability should be introduced in the methods. Why use numbers (14) and letters (eight).

> We introduced standard deviation into the methods. We did not modify the use of numbers/letters as they were presented like this because its states in the journal guidelines to "use words for cardinal numbers less than 10; use numerals for 10 and above (e.g. three flasks, seven trees, 6 m, 9 d, 10 desks)" – we have no strong opinion on this as it has no impact on the study and will change if necessary.

  P17. L301. The reporting of mean values should be ´ accompanied by estimates of variance, the degrees of freedom, and the significance of the trends.

> Done. This requires the reader to consult Figure 3-7 and Table 3 together, but all the necessary information and now contained and more easily retrievable from the figures and discussed in the text.

  P17. L303+. 'Only four North American sites experience later breakup ´ during...' The challenge here is that I can't tell if the authors are assessing significant or significant + non-significant trends. Was the example of Frame Lake a significant or non-significant change? The slope is reported (which appears quite close to zero), but no p value or measures of variance.

> Text clarified and partly removed as part of refocusing on larger-scale patterns requested by both reviewers.

  P17. L309+ Given the large scope of the ´ manuscript, it seems ill advised to drill down to the extent here for individual lakes or rivers. A blow out box may be more appropriate for dealing with specific case studies if there is a strong need to provide an example. The removal of these details will make it much easier for the reader to follow the broader relevance of the study and more space for the authors to discuss the relevance of their analysis compared to similar studies.

> These finer details were removed. They were originally included as we felt they provided an interesting narrative on why sites might vary. This was sacrificed in favour of the larger-scale analyses requested by both the reviewers.

  P49. L826+. I'm not sure I follow the logic here. How do summer wind ´ speeds alter ice break up in the spring (either the preceding or following spring)? This is likely a false positive and one would expect some false positives and negatives given the large number of tests. Second, the link between summer windspeed and effects on turbulence are only relevant once air temperatures drop below 0 deg C. Summer windspeeds could be important by deepening thermoclines and higher heat storage, leading to delayed ice out dates. Again, potential for false positive here. Further, I am not surprised by the poor relationships over North America and Russia given the large geographic area and changes in the distribution of sites. As noted by the authors in other locations (e.g. p18 L325), there can be large regional differences in climatic patterns within the study area that can lead to regional differences in ice phenology trends (e.g. NAM sites). While more labour intensive a better approach would be to examine the linkages between climate and ice phenology at the individual sites, and then bringing these relationships into a more global analysis (i.e. does ice phenology respond similarly to climate across sites with regional differences in climate causing regional differences in ice phenology, or are other factors (e.g. lake or watershed morphometry, lake vs. river, etc. critical).

> The text has been clarified and analyses made clearer as requested (and described above).

  Figure 4. Caption should indicate that trends were ´ tested using Mann Kendall and Sens slopes. After looking at Figure 3 from Duguay et al. 2006 (which I think is largely the same data), most of the ice freeze up values (panel e in this manuscript) would appear to be non-significant. This makes me

wonder whether all trends are reported regardless of significance. The authors need to clearly state in the figure caption whether the trends being reported are significant and what level of significance.

> All trends are reported, a lack of statistical significance does not mean a trend is not present, but we agree that in its original format this is confusing. We have edit a number of figures and the text that accompanies them to make the distinctions clearer. We believe this is considerably clearer now.

âAˇ c Table 3 is confusing. The authors must indicate what statis- ´tics were used (Mann Kendall with Sen's slopes?). What do the numbers represent in the panels represent (Sen's slopes?). What are their units (days/decade?). Why are significance values included only in the last three panels/time periods but not the first panel? Does the dash represent 'no data available'? Why not simply use the first panel data? Note, that another interpretation of the data from this table is that few lakes or rivers are displaying significant trends. For example in the 1931-90 period only 4/15 sites show significant changes at the p=0.05 level, or 7/15 at the p=0.10 level. Overall the table is interesting, but these site specific tables should probably be moved to supplemental material.

> As part of the refocusing of the material in this table was removed and any important issues mentioned in the text.

âAˇ c Table 3 shows that from the period 1991-2005 the major- ´ity of N America Lakes included in the analysis had either no significant trend, or had later ice on dates (significant positive numbers). There were no significant negative numbers. Yet, when I examine Figure 4, I see a large number of earlier freeze up dates reported (particularly to the east of the Laurentian Great Lakes). Am I missing something?

> Table has been removed and text reworked to ensure clarity around the this narrative.

âAˇ c Figure 5 caption is poorly written. What does 'Comparison of how ´sites in North America with an open water season calculation' mean? Perhaps 'Patterns of ice freeze-up and ice breakup....' What level of significance? What statistical test? Were non-significant trends reported as trends? Do white boxes represent no significant trend or no data?

> The figure was meant to show how the changes in the open water seasons compares to changes in the breakup and freezeup dates that drive it i.e. the reader should be able to see the trend directions during that time period for either and compare that to the overall changes in the number of open water days per year. However, as part of the downsizing of the paper we have cut this and similar figures and presented them in a simpler format on Figure 9, which is focussed on in the discussion section.

âAˇ c Figure 17. This figure is important as it represents ´the only example of where the drivers of the phenological trends were being tested. Although the methods are not entirely clear, the analysis seems to have a few issues. I am assuming that the authors assessed the trends of the 'downscaled' annual mean or median values of the freeze-up breakup dates over the full study period (1931-2005) with each study region (FEN, NAM, RUS). How was the air temperature data treated? Was it the mean

value of the whole study area? This presents some problems because in both North America and Russia, the study area is quite large and the number and geography of sites changed dramatically over the study period. For example, no Canadian sites were available during the most recent period. In this case, I would think that this would have a large effect on the relationship between air temperature and ice phenology. For Russia, there was a wide spread of sites (25-27 sites) longitudinally during the 1961-1990 period, but much fewer sites during the periods before (5-6 sites) and after (1 site). A better approach would be to examine the downscaled air temperature values for each site specifically. This would also remove potential problems associated with elevation. Further, a large number of correlations are being tested (e.g. >400 in panel b), which leads to the potential for false positives. A minor point, but presumably the bolding pattern refers to the level of significance of the trends. This would be important to point out in the caption. Perhaps I have misunderstood a few aspects of these analyses and in general more clarification of the statistical approach is needed.

We elected for this regionalised strategy because (1) the computational and human resources needed to analyse climate records for each individual site are much higher; and (2) we were interested in establishing broader regional climate drivers of ice phenology rather than developing correlations with local climate, which we would expect to be very strong. We have added this text to the Materials and Methods section.

The fact that we pick up some strong correlations between regional climate variables/indices and ice phenology provides a less obvious and more interesting finding than expected correlations with local climate – i.e. that broad regional climate exerts considerable influence over ice phenology. We have discussed the value and limitations of our approach in the Causes of Ice Phenology Change section, and we have also acknowledged the possibility of 'false positives' with so many correlations and advised caution when interpreting the findings.

We have also added text to clarify that the spatially-averaged regional time series for the three geographical regions encompassed only study sites with data for the full 1931-2005 time period. In this respect, the geographical area used to calculate the 'mean' climatic conditions did not need to change with changing time periods for sites.